# Exploring the limits of strong membership inference attacks on large language models

Jamie Hayes[1]*    Ilia Shumailov[1]    Christopher A. Choquette-Choo[1]
Matthew Jagielski[1]    Georgios Kaissis[1]    Milad Nasr[1]
Meenatchi Sundaram Muthu Selva Annamalai[2]    Niloofar Mireshghallah[3]    Igor Shilov[4]
Matthieu Meeus[4]    Yves-Alexandre de Montjoye[4]    Katherine Lee[1]
Franziska Boenisch[5]    Adam Dziedzic[5]    A. Feder Cooper[6,7]*

[1]Google DeepMind    [2]University College London    [3]University of Washington
[4]Imperial College London    [5]CISPA Helmholtz Center for Information Security
[6]Stanford University    [7]Microsoft Research

## Abstract

State-of-the-art membership inference attacks (MIAs) typically require training many reference models, making it difficult to scale these attacks to large pre-trained language models (LLMs). As a result, prior research has either relied on weaker attacks that avoid training references (e.g., fine-tuning attacks), or on stronger attacks applied to small models and datasets. However, weaker attacks have been shown to be brittle and insights from strong attacks in simplified settings do not translate to today's LLMs. These challenges prompt an important question: are the limitations observed in prior work due to attack design choices, or are MIAs fundamentally ineffective on LLMs? We address this question by scaling LiRA—one of the strongest MIAs—to GPT-2 architectures ranging from 10M to 1B parameters, training references on over 20B tokens from the C4 dataset. Our results advance the understanding of MIAs on LLMs in four key ways. While (1) strong MIAs can succeed on pretrained LLMs, (2) their effectiveness, remains limited (e.g., AUC<0.7) in practical settings. (3) Even when strong MIAs achieve better-than-random AUC, aggregate metrics can conceal substantial per-sample MIA decision instability: due to training randomness, many decisions are so unstable that they are statistically indistinguishable from a coin flip. Finally, (4) the relationship between MIA success and related LLM privacy metrics is not as straightforward as prior work has suggested.

## 1 Introduction

In a membership inference attack (MIA), an adversary aims to determine whether a specific data record was part of a model's training set [53, 63]. MIAs pose a significant privacy risk to ML models, but state-of-the-art attacks are often too computationally expensive to run at the scale of pre-trained large language models (LLMs). This is because strong MIAs require training multiple reference models to form membership predictions—and pre-training even one LLM is often prohibitively expensive in research settings. As a result, current work makes one of two compromises: running weaker attacks that avoid training reference models (e.g., attacks that fine-tune an LLM), or running strong attacks that train small reference models on small datasets. However, both exhibit notable limitations (Section 2). Weaker attacks are more practical, but they have been shown to be brittle—often performing no better than random guessing [18, 21, 43]. Stronger attacks, when run in simplified settings, fail to capture the complex dynamics of large-scale, pre-trained language models; as a result, their insights do not reliably generalize to modern LLMs [38].

Results from both of these approaches leave key questions unanswered about the effectiveness of MIAs on LLMs. In particular, *are the fidelity issues of weaker attacks due to omitting reference*

---

*Equal contribution; corresponding authors: `jamhay@google.com`, `afedercooper@gmail.com`

*models, or do they point to a deeper, more fundamental challenge with applying membership inference to large language models?* Current research has not offered an answer because, to date, there are no baselines of how the strongest MIAs perform on large-scale, pre-trained LLMs.

In this paper, we bridge this gap by running strong attacks at a scale significantly larger than previously explored. We pre-train over 4,000 GPT-2–like reference models, ranging from 10 million to 1 billion parameters [31], on subsets of the C4 dataset [47] that are *three orders of magnitude larger than those used in prior MIA studies*—over 50 million samples, compared to fewer than 100,000 in previous work [40]. We use these models to conduct a detailed investigation of the Likelihood Ratio Attack (LiRA) [5], one of the strongest MIAs in the literature. This substantial effort proves worthwhile, as we uncover four key insights that advance the state of the art in understanding the potency and reliability of membership inference attacks on large language models:

- **Strong membership inference attacks can succeed on pre-trained LLMs.** We are the first to execute strong attacks at this scale, and find that LiRA—in contrast to weaker fine-tuning attacks—can easily beat random ROC-AUC baselines (Section 3.1). Our results on Chinchilla-optimal models (trained for 1 epoch) exhibit a non-monotonic relationship between model size and MIA vulnerability: larger models are not necessarily more at risk (Section 3.2).

- **The overall success of strong MIAs is limited on pre-trained LLMs.** Even though we demonstrate that LiRA can succeed at LLM scale, we are only able to achieve impressive results (i.e., AUC≥0.7) when diverging from typical training conditions—specifically, by training for multiple epochs (Section 4.1) and varying training dataset sizes (Section 4.2).

- **Many correct per-sample MIA decisions for LLMs do not reflect reliable inference.** Even when an MIA achieves better-than-random AUC, the underlying per-sample binary decisions are highly sensitive to training randomness. We quantify this by measuring per-sample decision instability (Section 5.1) and find that, even at modest FPR, many per-sample decisions are statistically indistinguishable from a coin flip (Section 5.2).

- **The relationship between MIA success and related LLM privacy metrics is not straightforward.** We show that samples seen later in training tend to be more at risk (Section 6.1); however, this trend is complicated by sample length. We also study if there is any relationship between training data extraction and MIA, and observe no correlation with MIA success. This suggests that the two privacy attacks may capture different signals related to memorization (Section 6.2).

Our contributions serve as an extensive benchmark of strong MIAs, and also provide some initial answers to urgent open questions about the conditions under which MIAs exhibit a threat to privacy for LLMs. Our work also quantifies the performance gap between weaker (more feasible) and stronger attacks, establishing an upper bound for what weaker attacks could achieve in this setting.

## 2 Background and related work

**Membership inference attacks (MIAs)** assess empirical privacy and information-leakage risk by asking whether an adversary can tell if *a particular data point* $x$ was used to train a **target model** $h$. Given knowledge of the target's architecture and training setup, the attacker trains multiple **reference models** $f \in \Phi$ on different subsets drawn from the same underlying distribution as the target's training data. For each $x$, references are partitioned into those trained with $x$ ($\Phi_{\text{IN}}$, where $x$ is a **member**) and those trained without $x$ ($\Phi_{\text{OUT}}$, where $x$ is a **non-member**). For a given $x$ and model $g$ (the target $h$ or a reference $f \in \Phi$), the attacker queries the model and computes an **observation statistic** $s(g, x)$ from the model's output on $x$ (e.g., loss, logit). MIAs transform these statistics into a **membership score** $\Lambda(x)$ that is used to infer whether $x$ was in the target's training data [5, 49, 58, 62, 64].

Different attacks specify different ways of turning observation statistics into membership scores. For instance, for each query sample $x$, the **Likelihood Ratio Attack (LiRA)** collects two sets of reference statistics, $\{s(f, x) : f \in \Phi_{\text{IN}}(x)\}$ and $\{s(f, x) : f \in \Phi_{\text{OUT}}(x)\}$. These sets are treated as samples from two empirical distributions, to which density models ($p_{\text{IN}}$ and $p_{\text{OUT}}$) are fit. LiRA evaluates the target statistic $s(h, x)$ under the fitted densities to compute a likelihood ratio membership score $\Lambda(x)$ for $x$ [5]. Given a score $\Lambda(x)$, the attacker outputs a binary membership decision via a threshold rule $b(x) = \mathbf{1}\{\Lambda(x) \geq \tau\}$. In practice, $\tau$ is typically calibrated on non-members to satisfy a fixed false positive rate (FPR) $\eta$. Although membership inference is defined as a decision problem for a *single* sample $x$, attack performance is evaluated as an *average over many samples* (e.g., reporting TPR at fixed FPR). Success is typically reported with threshold-agnostic metrics like

ROC-AUC [53, 63] (Appendix A). To address this gap, we also run experiments that offer novel insights into sample-specific attack performance (Sections 5 & 6, Appendices E & F).

The number of reference models necessary for successful attacks varies across methods—from tens or hundreds for LiRA and Attack-R [62], to as few as 1 or 2 for RMIA [64]. While these attacks have been successfully applied to smaller settings, they are often considered impractical for contemporary language models due to the prohibitive computational cost of training even a single reference LLM. As a result, prior work attempts to approximate stronger, reference-model-based attacks in various ways.

**Small-scale, strong, reference-based attacks.** The first work to evaluate MIAs in smaller language models (RNNs) trained 10 references [54]. However, insights from such settings do not translate to today's LLMs [40], as the training dynamics differ significantly. Other work has used a single reference model to attack a small, pre-trained masked language models [42], but this approach reduces precision, as effective membership inference is difficult with fewer references.

**Larger-scale, weak, reference-free attacks.** To avoid the cost of training reference models, weaker attacks consider a range of observation statistics to infer membership, typically leveraging black-box access to the model. Yeom et al. [63] use model loss, Carlini et al. [4] use normalized model loss and `zlib` entropy, and Mattern et al. [37] compare the model loss to the loss achieved for neighboring samples. More recent work experiments with token probabilities [52, 66] and changes in loss based on prompting with different context [57, 61].

Other work attempts to derive membership signal from changing the model. For instance, prior work perturbs inputs or model parameters and observes resulting changes in target loss on the sample, or uses (parameter-efficient) fine-tuning on domain-specific datasets to detect privacy risks [8, 21, 28, 33, 41, 43, 46, 48]. However, fine-tuning attacks introduce *new* data to the problem setup, which may complicate the validity of using MIAs to detect benchmark contamination [17, 34, 35, 45] and to draw reliable conclusions about other sensitive data issues [10, 12, 13, 15, 19, 30, 39, 52, 60, 65]. A recent approach evaluates attacks on LLMs using post-hoc collected datasets. While prior work has reported high success rates on a variety of models and datasets (AUC≈0.8) [38, 52, 57, 61, 66], such evaluations rely on the model's training-date cutoff as a proxy for distinguishing between member and non-member data points [35]. These newer data introduce distribution shift, which can undermine the validity of the reported results [16, 18, 35, 40]. Further, when current MIAs are evaluated in a controlled privacy game like this, they often barely outperform random guessing [18, 40].

## 3 Examining strong MIAs in realistic settings for pre-trained LLMs

Altogether, the limitations of prior work raise the key question that motivates our work: *are the fidelity issues of weaker attacks due to omitting reference models, or do they point to a deeper, more fundamental challenge with applying membership inference to large language models?* This is a big question, so we break it down into smaller ones that we can test with specific experiments that reveal different information about the effectiveness of strong MIAs on pre-trained LLMs. To start, we determine which strong MIA method to use across our experiments. We evaluate two of the strongest attacks in the literature—LiRA [4] and RMIA [64]—in a variety of settings. For the experiments that follow, we use LiRA because we observed that it can achieve substantially higher ROC-AUC when attacking pre-trained LLMs. We compare LiRA and RMIA in Appendix B.

In this section, we investigate the relationship between the number of reference models and attack success (Section 3.1). Based on these results, we decide to use 128 reference models in all following experiments. Then, we test the effectiveness of strong attacks under realistic settings—settings that reflect how LLMs are actually trained. To do so, we run LiRA on models of various sizes, which we train according to Chinchilla-scaling laws [26] (Section 3.2). Together, these experiments inform our first key result: with respect to overall ROC-AUC, **strong membership inference attacks can succeed on pre-trained LLMs**. In the following sections, we expand upon these results to other training and attack conditions; we will refine our first key result by investigating the limits of strong MIA success rates (Section 4), and by digging beneath aggregate metrics like AUC to better understand attack performance with respect to individual samples (Sections 5 & 6).

**General setup.** For all experiments, we pre-train GPT-2 architectures of varying sizes—from 10M to 1B—on subsets of the C4 dataset [47] using the open-source NanoDO library [31]. The training datasets we use are 3 *orders of magnitude larger than those in prior MIA studies*: over 50M samples, compared to fewer than 100K samples in previous work [40]. We explore datasets of this size

because, while it is well established that MIA success depends on both model capacity and training dataset size [53, 62, 63], the nature of this relationship remains unexplored pre-trained-LLM scale. For each attack, we start with a fixed dataset of size $2N$ (e.g., 20M) drawn from C4, from which we randomly subsample (with different random seeds) reference training sets of size $N$ (e.g., 10M). So, for each reference $f$, half of the drawn samples are members and half are non-members. This yields the different member (IN) and non-member (OUT) distributions for each sample that we use to run LiRA. In our largest experimental setting, we use $2N \approx 100M$. Specific experimental configurations vary, so we introduce additional setup as needed. (See Section G for details.)

## 3.1 Warm-up: How many reference models should we use?

To determine the number of reference models to use for all of our experiments, we train 140M-parameter models on $\approx$7M samples, which equates to approximately 2.8B training tokens. (This is optimal for this model size, according to Chinchilla scaling laws with an over-training multiplier of 20 [26].) As shown in Figure 1, we test a range of reference models. (We plot the number of IN references; the total number of IN/OUT references is $2\times$ this number.) The plot shows multiple receiver operating characteristic (ROC) curves, indicating the observed true positive rate (TPR) for the fixed false positive rate (FPR) on a log-log scale. Area under the curve (AUC) is provided for each ROC. The dashed red line represents the baseline for which the attack is equivalent to random guessing (i.e.,

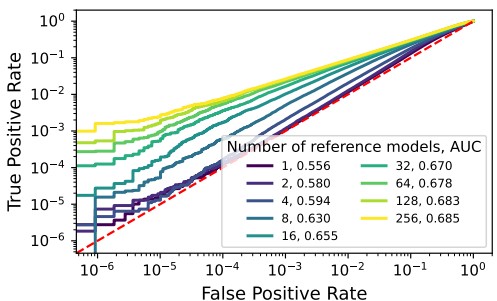

Figure 1: **LiRA with different references.** We attack a 140M model trained on $\approx$7M samples. As references increase, LiRA's performance improves (measured with ROC-AUC). However, there are diminishing returns: AUC is effectively unchanged from 128 to 256 IN references.

cannot distinguish between true and false positives so TPR=FPR; AUC=0.5). We report AUC as our primary metric, as it is otherwise challenging to visualize TPR over a wide range of fixed FPR. (For comparison, see Figure 2b, which shows a limited range of FPR, but does not surface threshold-agnostic AUC.) We also investigate the performance of different observation statistics (Section B.1), and choose to use model loss. Altogether, while LiRA clearly beats the random baseline, it is not remarkably successful in this setting: regardless of the number of references, it never achieves an AUC of 0.7. Even though success increases with more references, there are diminishing returns. From 1 to 8 IN references (2 to 16 references total), AUC has a relative increase of 13.3%; for the next 8× increase (from 8 to 64), AUC only increases 7.6%; and, doubling from 128 to 256 only yields a 0.2% improvement. We opt to use 128 total references (64 IN, 64 OUT) in most experiments below.

## 3.2 Training and attacking a compute-optimal model

In practice, models are typically trained based on observed scaling laws: for a given model size, the scaling law suggests the optimal number of tokens to use for training. To assess strong MIA in realistic conditions for pre-trained LLMs, we attack models of various sizes, setting the number of training samples to be optimal according to Chinchilla scaling [26]. Specifically, we set the number of training tokens to be 20× larger than the number of model parameters and we *only train for 1 epoch*—a common choice in large training runs [1, 56]. Specific training recipes and experimental details are in Appendices C and G, including the number of samples used to train each model size.

In Figure 2, we show two views of the results of attacking 10M-, 85M-, 302M-, 489M-, 604M- and 1018M-parameter models. These model sizes come from the default configurations available in NanoDO [31]. For readability, we exclude the results for the 140M model, as we investigate this architecture above. In Figure 1, the attack on the 140M model with 128 IN/OUT references has AUC=0.678, which puts its performance below the 85M and 302M models. Interestingly, we observe a non-monotonic relationship between model size and MIA vulnerability under these training conditions. In Figure 2a, the 85M model shows the highest AUC=0.699, followed by the 302M (AUC=0.689). The 489M model exhibits the lowest success (AUC=0.547).

Figure 2b provides a different view of the same results. By model size, each line compares the TPR for fixed settings of FPR. Our expectation was that each line would look approximately horizontal, as the training set size is being scaled proportionally (and optimally, according to Hoffmann et al. [26])

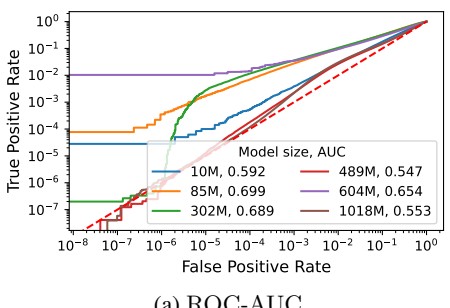
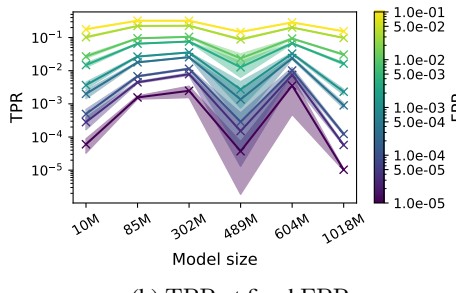

(a) ROC-AUC                                    (b) TPR at fixed FPR

Figure 2: **MIA vulnerability for compute-optimally trained models.** We show attacks on 6 models of different sizes under Chinchilla-optimal conditions for 1 epoch, using 128 references. (**a**) ROC curves demonstrate varying MIA susceptibility for 10M (AUC=0.592), 85M (AUC=0.699), 302M (AUC=0.689), 489M (AUC=0.547), 604M (AUC=0.654) and 1018M (AUC=0.553). The 85M and 302M models show the highest vulnerability, indicating that increasing model size does not uniformly decrease MIA risk in this setting. (**b**) How TPR for each fixed FPR varies by model size.

to model size. From 10M to 302M, there is a consistent pattern of the TPR increasing with model size; but then, at 489M, there is a significant drop in TPR. There are many reasons why this may have occurred. First, the most pronounced differences in TPR are at extremely small values. Even subtle differences in training runs may flip samples from correct to incorrect member predictions (Section 5), which, in the low TPR regime, can have a large effect on overall MIA success. Second, Chinchilla scaling [26] is not the only such law. Sardana et al. [50], Hu et al. [27], and Grattafiori et al. [23] all introduce other ways to optimally select the number of training tokens for a given model. In future work, we will investigate if these other token-size-selection methods stabilize TPR as model size grows.

As we discuss next (Section 4.2), repeating this experiment by training these architectures on the same fixed dataset size exhibits vastly different results. We additionally test other training configurations. In Appendix D, we alter the learning rate schedule and observe that there is a modest effect on attack performance. (See Appendix C, where, as a sanity check, we also confirm that larger models converge to lower loss values, reflecting their increased capacity to fit the training data.)

## 4  Varying compute budget and training dataset size

Even in the most successful (i.e., highest AUC) case, overall attack performance is not particularly impressive when running LiRA with a large number of references on compute-optimal models trained for 1 epoch. Similar to our experiments with LiRA and varied numbers of references (Figure 1), the maximum AUC we observe remains under 0.7 for all model sizes (Figure 2). This raises a natural follow-on question: if we free ourselves from the constraints of typical training settings, is it possible to improve success? *Can we identify an upper bound on how strong MIAs could perform on pre-trained LLMs?*

To address this question, we run attacks on models trained on different-sized (not always Chinchilla-optimal) datasets (Section 4.2) for more than 1 epoch (Section 4.1). Our experiments show that diverging from typical settings can indeed improve attack success. However, while these experiments are a useful sanity check, they do not suggest conclusions about the effectiveness of strong MIAs in general. Instead, there appears to be an upper bound on how well strong MIAs can perform on LLMs under practical conditions. In other words, these experiments inform our second main observation: **the success of strong MIAs is limited in typical LLM training settings.**

### 4.1  Effects of scaling the compute budget (i.e., training for more epochs)

In Figure 3a, we compare MIA AUC for the 44M model under different training configurations. We keep the total number of tokens surfaced to the model during training Chinchilla-optimal, but we alter *when* these tokens are surfaced. As a baseline, we train for 1 epoch on the entire dataset, and achieve AUC=0.620 with LiRA. (See Figure 3a, 1 of 1.) We then take *half* of the training dataset and train the same architecture for 2 epochs. In both settings the total number of training tokens is Chinchilla-optimal, however, in the latter, the model has processed each training sample twice rather than once. For the 2-epoch model, we observe a significant increase in MIA vulnerability: AUC=0.744, which is higher than when this model has only completed 1 epoch of training (AUC=0.628, 1 of 2) and than when the model is trained for 1 epoch on the entire dataset (AUC=0.620, 1 of 1). Increasing

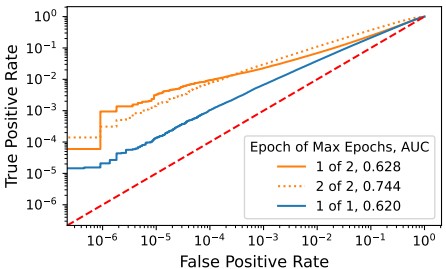
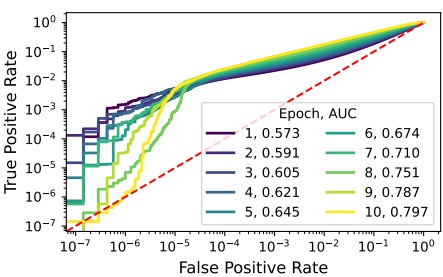

(a) 44M model, split dataset in half and train for 2 epochs, or train on the entire dataset for 1 epoch

(b) 140M model, training for 10 epochs

Figure 3: **Studying the effect of varying epochs.** (**a**) We compare attacking a 44M model trained on the whole Chinchilla-optimal dataset in 1 epoch (AUC=0.620 after 1 of 1 epoch) to training for 2 epochs on only half of the dataset (AUC=0.744 after 2 of 2 epochs). (**b**) We attack a 140M model trained on the whole Chinchilla-optimal dataset for 10 epochs. AUC increases with more epochs.

training epochs—even on a smaller dataset to maintain Chinchilla optimality—amplifies vulnerability to MIA, compared to training for fewer epochs on a larger dataset. However, there is no significant uplift in TPR at small fixed FPR between epochs 1 and 2 for the 2-epoch model. The MIA at the second epoch is less successful than the one after 1 epoch for small FPR. As above, this is perhaps due to subtle differences in runs having an impact at these small values (Sections 3.2 & 5).

To investigate this further, in Figure 3b, we show how AUC changes over the course of training the 140M model for 10 epochs. As expected, AUC increases with more epochs, starting from 0.573 and reaching 0.797 at the end of the tenth epoch.[2] As in Figure 3a, there is an FPR inflection point where TPR for later epochs is *smaller* than earlier epochs. In Appendix D, we also train the 140M model on fewer than the $\approx$ 7M Chinchilla-optimal samples, and (similar to Figure 3a) we observe a more dramatic increase in MIA vulnerability. Attacking a 140M model trained on $2^{19} \approx$ 500K samples exhibits both greater absolute MIA success and a faster relative increase in success in the first few epochs.

## 4.2   Effects of scaling the training dataset size

We next run two sets of experiments to study the role of training dataset size on MIAs—beyond training on the Chinchilla-optimal number of tokens. We train 140M models on datasets ranging from 50K to 10M samples (again for 1 epoch) and attack these models with LiRA. In Figure 4a, we show ROC curves for the different models. As we train models on smaller datasets, for a given FPR, TPR does not always increase. This suggests that TPR at fixed FPR is not necessarily positively correlated with decreasing the training set size. Rather, AUC is highest for moderately sized datasets (around 1M samples, AUC=0.753), and decreases for both very small and very large datasets (under AUC=0.7 for both). Indeed, the capacity of the model also has an effect on susceptibility to strong MIAs.

In Figure 4b, we train different model sizes with a fixed training set size of $2^{23} \approx$ 8.3M samples— significantly more tokens than is Chinchilla-optimal for several models (e.g., 10M, 44M). We plot the mean and standard deviation of TPR at fixed FPR, where we run the attack 16 times using different random seeds, which has the effect of dictating the batch order. For each model size, we train 16 sets of 128 reference models, and we also vary the target model over each attack. We include the associated ROC-AUC for each model size in Appendix D, which are consistent with the MIA prediction variability in Figure 4b. We observe a monotonic increase in TPR at different FPRs as model size increases. This is quite different from Figure 2b, where we scale the training set size with model size. As model capacity grows, vulnerability to MIA also grows if we keep the training set size constant. Further, there is significantly more variance in TPR for larger model sizes and at smaller fixed FPR.

## 5   Uncovering per-sample MIA membership decision instability

The high degree of variance that we observe in the prior section raises a natural question: how stable are the underlying per-sample membership decisions in strong MIAs? Next, we describe the setup we use to measure per-sample MIA decision instability (Section 5.1). In general, this is a reasonable

---

[2]At epoch 1, AUC=0.573, which differs from AUC=0.678 in Figure 2a (also 1 epoch). This is likely because of variance between runs (Section 5) and substantially different learning rates between the two setups.

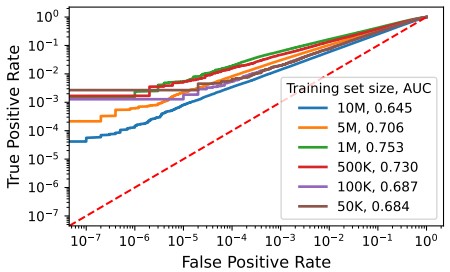
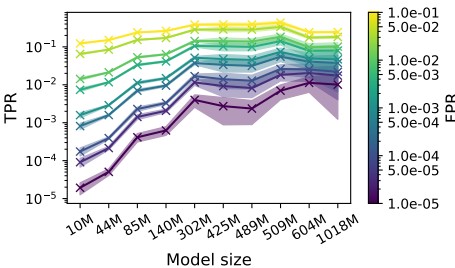

(a) 140M model × various dataset sizes

(b) Various model sizes × $2^{23}$ sample training set

Figure 4: **Varying sizes of training dataset and model** (1 **epoch**). (**a**) We attack 140M models trained on different-size datasets (50K to 10M samples). MIA success does not monotonically increase with dataset size. (**b**) We attack different-size models trained on a fixed dataset size ($\approx$8.3M samples), and plot how TPR varies at fixed FPR. MIA success monotonically increases with model size.

thing to do: it is standard to report attack success with *aggregate* metrics (AUC, TPR at fixed FPR), but the MIA security game is defined with respect to determining if *a particular sample* $\boldsymbol{x}$ was used in training (Section 2 & Appendix A). We show a selection of results for the 302M model (Section 5.2), which reveal our third key takeaway: aggregate metrics may imply that a strong MIA on an LLM performs better than random guessing; however, even at modest FPR, due to training randomness **a large fraction of underlying, individual membership decisions are highly unstable, resembling coin flips**. For these samples, strong MIAs are not producing reliable knowledge about membership.

## 5.1 Computing per-sample flip rate on calibrated membership decision rules

We fix the IN/OUT references and target training set, varying only the random seed used to train targets. By isolating variability from equally plausible targets, we can probe the stability of using counterfactual references to derive membership signal for a given sample $\boldsymbol{x}$. Substantial cross-target disagreement on $\boldsymbol{x}$'s membership would indicate the signal is not providing robust membership evidence for $\boldsymbol{x}$.

Formally, let $r \sim \mu$ denote a target trained on a *fixed* dataset with randomness induced by the seed controlling batch order. We train *one set* of references for all attacks on different $r \sim \mu$. Let $\Lambda_r(\boldsymbol{x}) \in \mathbb{R}$ be the $r$-specific LiRA score for $\boldsymbol{x}$. At a fixed FPR $\eta$, we calibrate a per-seed threshold $\tau_r(\eta)$ on non-members to form the binary decision rule $b_r^{(\eta)}(\boldsymbol{x}) = \mathbf{1}\{\Lambda_r(\boldsymbol{x}) \geq \tau_r(\eta)\}$ (Section 2). Per-seed calibration mirrors the standard threat model, in which an attacker runs the MIA on a *single* target [5]. The (population) **flip rate** [14] at $(\eta, \boldsymbol{x})$ is the pairwise decision disagreement probability under $\mu$:

$$\text{flip}_\eta(\boldsymbol{x}) \ := \ \Pr_{r,r' \stackrel{\text{i.i.d.}}{\sim} \mu} \big[ b_r^{(\eta)}(\boldsymbol{x}) \neq b_{r'}^{(\eta)}(\boldsymbol{x}) \big].$$

In practice, with $B \geq 2$ i.i.d. target replicas $r_1, \dots, r_B \sim \mu$, the canonical unbiased estimator is

$$\widehat{\text{flip}}_{\eta,B}(\boldsymbol{x}) \ = \ \binom{B}{2}^{-1} \sum_{1 \leq i < j \leq B} \mathbf{1}\{b_{r_i}^{(\eta)}(\boldsymbol{x}) \neq b_{r_j}^{(\eta)}(\boldsymbol{x})\} \ = \ \frac{2 B_0(\boldsymbol{x}) B_1(\boldsymbol{x})}{B(B-1)}, \tag{1}$$

where $B_1(\boldsymbol{x}) = \sum_{i=1}^B b_{r_i}^{(\eta)}(\boldsymbol{x})$ and $B_0(\boldsymbol{x}) = B - B_1(\boldsymbol{x})$ are the counts of member and non-member decisions for $\boldsymbol{x}$ at $\eta$ among the $B$ target replicas. In practice, $\widehat{\text{flip}}_{\eta,B}(\boldsymbol{x}) \in [0, \approx0.5]$; the finite-$B$ maximum exceeds $0.5$ and converges to $0.5$ as $B \to \infty$ (Appendix E.2). $\widehat{\text{flip}}_{\eta,B}(\boldsymbol{x}) \approx 0$ means the MIA decision for $\boldsymbol{x}$ is stable across equally plausible targets; $\widehat{\text{flip}}_{\eta,B}(\boldsymbol{x}) \approx 0.5$ means the decision is statistically indistinguishable from a coin flip, with roughly half of decisions calling $\boldsymbol{x}$ a member, and the other half calling $\boldsymbol{x}$ a non-member. Figure 5b provides an intuition for how this can occur. For a member $\boldsymbol{x}$ at FPR $\eta = 10^{-2}$, we plot the reference IN/OUT distributions and the median target statistic $s$ (across the $B = 127$ targets). The IN/OUT distributions overlap substantially, making membership for this $\boldsymbol{x}$ hard to disambiguate; accordingly, the 127 decisions are split down the middle, so $\widehat{\text{flip}}_{10^{-2},127}(\boldsymbol{x}) \approx 0.5$.

## 5.2 Many membership decisions are statistically indistinguishable from a coin flip

Flip rate (Equation 1) lets us peer beneath average metrics to assess the reliability of the MIA procedure's decisions for individual samples (Appendix E.2.5). For the 302M model, we train 128 IN/OUT references to use for all attacks, and 127 target replicas on the exact same $\approx$500K dataset with different seeds. While the population $\text{flip}_\eta(\boldsymbol{x}) = 0.5$ indicates coin-flip decisions for $\boldsymbol{x}$, with finite replicas $B$, we need to determine a defensible cutoff above which $\widehat{\text{flip}}_{\eta,B}$ signifies this behavior.

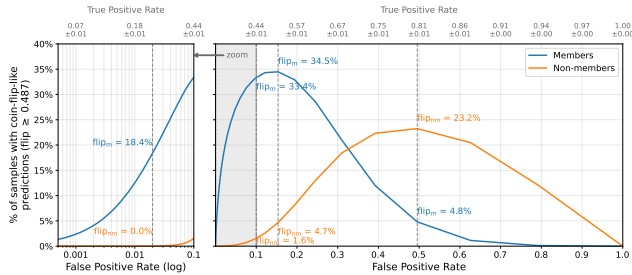
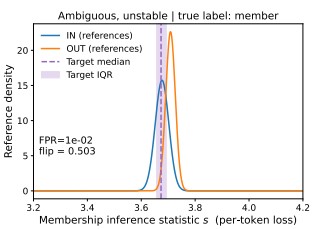

(a) Flip rate (Equation 1) by membership at varied fixed FPR, $B{=}127$    (b) Unstable member sample

Figure 5: **Visualizing decision instability.** We train $B{=}127$ targets for the 302M model on $2^{19}$ samples, and one set of 128 references to use for all attacks. LiRA achieves stable mean AUC=0.752 $\pm$ 0.007, yet many per-sample decisions behave like coin flips. (**left**) Share of coin-flip-like decisions across FPR (log-scale for small FPR; $\widehat{\mathrm{flip}}_{\eta,B}{\gtrsim}0.487$, the $\alpha{=}0.05$ cutoff, see Appendix E.2.4). Members exhibit more coin-flip-like decisions than non-members. (**right**) A representative unstable member: IN/OUT distributions overlap; the $B$ target decisions flip because $\boldsymbol{x}$'s seed-specific scores lie near the respective seed-specific thresholds (Appendix E.2.5).

To do so, we set up a two-sided exact binomial test for $\theta{:=}\Pr_{r\sim\mu}[b_r^{(\eta)}(\boldsymbol{x}){=}1]$, and use "coin flip" as shorthand for "fail to reject the null $H_0 : \theta{=}0.5$ at level $\alpha$." For $B{=}127$ target replicas and $\alpha{=}0.05$, this corresponds to the cutoff $\widehat{\mathrm{flip}}_{\eta,127}{\gtrsim}0.487$ (Appendix E.2.4).

**Aggregate attack success is high and stable.** A training set of $\approx$500K samples is significantly smaller than what is Chinchilla-optimal for the 302M model ($\approx$15.1M), so we expect higher overall MIA success (Section 4.2). Indeed, aggregate attack success is stable, and substantially outperforms random guessing (AUC=0.752$\pm$0.007). At fixed FPR, TPR is also stable (Figure 5a, mean TPR $\pm$ STD annotations). Nevertheless, models $r\sim\mu$ that yield similar accuracy can have very different decision rules, and therefore can disagree substantially on individual decisions [3] (Appendix E.2.3).

**At meaningful** FPR**, flip rate rises with** FPR **and model size, and is higher for members.** Figure 5a shows that, even at modest FPR, large numbers of membership decisions behave like coin flips. Across fixed FPR $\eta$, we plot the proportion of samples with statistically coin-flip-like decisions, i.e., $\widehat{\mathrm{flip}}_{\eta,127}{\gtrsim}0.487$ ($\alpha{=}0.05$); the samples that satisfy this filter resemble the sample in Figure 5b. At $\eta{=}0.02$, $\approx$18.4% of members have coin-flip decisions (at $\alpha{=}0.05$); if we relax the flip threshold to also include highly unstable $\widehat{\mathrm{flip}}_{0.02,127}{\geq}0.4$ decisions, this proportion becomes $\approx$39.8%. (By contrast, for non-members these proportions are $\approx$0.03% and $\approx$0.2%, respectively. This is unsurprising because decision thresholds are calibrated on non-members; see Appendix E.2.5.)

At lower FPR, as FPR increases, the proportion of samples with coin-flip-like decisions increases. Each seed's calibrated threshold $\tau_r(\eta)$ moves into score regions where IN/OUT overlap is more extensive, expanding the overall set of samples (especially members) whose scores lie near the decision boundary. As a result, small seed-induced score shifts (as well as across-seed variation in $\tau_r(\eta)$ itself, see Appendix E.2.5) flip decisions more often. This effect is more pronounced for the 302M model, compared to the 140M model (Appendices E.2.3 & E.2.5).

**This is *not* an attack, but a diagnostic of MIA signal robustness using counterfactuals.** We are able to assess per-sample MIA decision instability by training *many* different targets $r\sim\mu$, each of which is a plausible outcome of the same training process. However, this is clearly not the same procedure that an attacker runs: under the standard threat model (Section 2 & Appendix A), an attacker faces a *single* target. This matters, even though a true positive is an MIA success: a coin-flip-like MIA *decision* may be *correct*, but it does *not* show that the *MIA procedure* is informing a *reliable inferential claim* about membership for that sample. For such samples, flip rate shows that attack success is fragile: the claim for $\boldsymbol{x}$ is an artifact of seed-specific idiosyncrasies, rather than a reflection of stable signal obtained from running that specific MIA procedure (Appendix E.2.7). An alternative diagnostic could fix the target, train $B$ independent IN/OUT reference sets on the same IN/OUT draws (varying only random seed), and measure disagreement with respect to those sets. However, this is substantially more expensive. Our experiments varying the target similarly probe counterfactual robustness; we would expect to observe qualitatively similar instability under reference resampling (Appendix E.2.8).

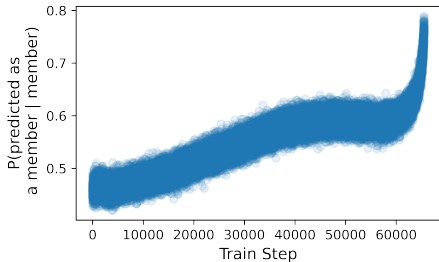
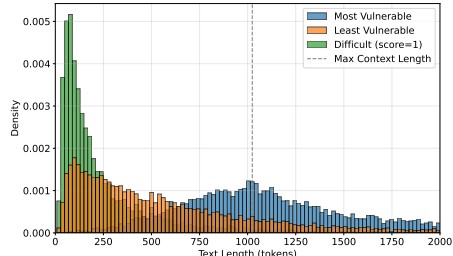

(a) Per-member success over training steps.

(b) Sample-length distributions by vulnerability.

Figure 6: **Sample vulnerability to MIA.** For the 140M model, (**a**) the evolution of sample vulnerability during training, shown by sample true positive probabilities $\Pr(\text{predicted as a member}|\text{member})$ at each step. (**b**) Distributions over sample lengths, according to MIA vulnerability for the $1,000$ samples that are least vulnerable, most vulnerable, and most difficult for MIA (i.e., with smallest, largest, and closest to $0.5$ $\Pr(\text{predicted as a member}|\text{member})$.) See Appendix E.1.

Overall, flip rate diagnostics show that training randomness plays a significant role in per-sample MIA decisions. Even at $\text{FPR}=10^{-3}$, we estimate for the 302M model that roughly $15.4\pm0.6\%$ of true positives behave like coin flips—i.e., $\widehat{\text{flip}}_{10^{-3},127}\gtrsim0.487$, at $\alpha=0.05$. If we expand to include highly unstable decisions ($\widehat{\text{flip}}_{10^{-3},127}\geq0.4$), these constitute $42.2\%\pm0.9\%$ of true positives (Appendix E.2.6).

# 6 Analyzing sample vulnerability to membership inference

The instability in membership predictions that we observe for individual samples suggests a natural follow-on question: when does strong MIA succeed? Which samples are actually vulnerable to MIA, and (how) does this vulnerability vary during training? We approach these questions by digging deeper into our strong attacks on 140M models—trained with a Chinchilla-optimal training set ($\approx$7M samples) for 1 epoch—with 128 references. Samples seen later in training tend to be more vulnerable; however, this trend is complicated by sample length (Section 6.1). While sample length has previously been linked to extraction risk [7, 44], we observe no correlation between MIA and standard extraction methodology (Section 6.2). Together, this analysis informs our fourth key takeaway: **the relationship between MIA vulnerability and related privacy metrics is not straightforward for LLMs.**

## 6.1 Identifying patterns in per-sample MIA vulnerability

We first investigate how sample MIA vulnerability evolves over the course of training. In Figure 6a, the scatter plot illustrates per-sample true positive probabilities by training step: we plot how the probability of a training sample being correctly predicted as a member changes as model training progresses, where the membership prediction for $\boldsymbol{x}$ is computed using the reference distributions, i.e., $\frac{p_{\text{IN}}(\cdot|\boldsymbol{x})}{p_{\text{IN}}(\cdot|\boldsymbol{x})+p_{\text{OUT}}(\cdot|\boldsymbol{x})}>0.5$ (Section 2 & Appendix A). Across samples in the batch at each step, there is considerable variance in the underlying sample true positive probabilities $\Pr(\text{predicted as a member}|\text{member})$: it can vary by more than $15\%$, having an effect on overall attack success. For much of training, the mean $\Pr(\text{predicted as a member}|\text{member})$ is close to $0.5$, indicating many samples are challenging for MIA to distinguish as either members or non-members.

The density of the points shifts upward toward the end of training (around step $60,000$). Samples in batches that are processed in later epochs tend to be more vulnerable, as indicated by the higher probability of being correctly identified as members. This result highlights that the recency of exposure influences a sample's vulnerability to membership inference. Put differently, samples introduced earlier in training are more likely to be "forgotten" [6]: they are less vulnerable to MIA. This is perhaps a partial reason for LiRA decision instability for targets trained on the same dataset, but with different random seeds that control batch order (Section 5). For some targets, a member $\boldsymbol{x}$ may be seen late in training and exhibit a high true positive probability; for others, the same $\boldsymbol{x}$ may appear early and be "forgotten." (i.e., result in false negatives).

While this appears to be the dominant trend, the details are more complicated. In Figure 6b, we plot the distribution over members according to length, and partition this distribution according to vulnerability. We consider members for which LiRA's predictions are confident but incorrect (i.e., predict non-member) to be least vulnerable, and members that LiRA correctly and confidently predicts as members to be most vulnerable. We also highlight members for which LiRA struggles to determine

membership status (true positive probabilities ≈0.5). Figure 6b suggests that vulnerable sequences tend to be longer. (See also Appendix F, which illustrates similar results for samples that have a higher proportion of `<unk>` tokens and higher average TF-IDF scores.) This result is consistent with those in Carlini et al. [7], which show that longer sequences tend to be more vulnerable to extraction attacks.

## 6.2 Comparing MIA vulnerability and extraction

Results such as those in Figure 6b are consistent with prior work on memorization and extraction [4]. In general, it is assumed that a successful membership inference attack and successful extraction of training data imply that some degree of memorization has occurred for the attacked ML model. For MIA, this is assumed because the success of such attacks hinges on the model's tendency to behave differently for data it has seen during training (members) compared to unseen data (non-members) (Section 2 & Appendix A). Prior work frequently ascribes this differential behavior to the model having memorized certain aspects of the training data.

We therefore investigate whether samples that are vulnerable to strong MIAs are also vulnerable to standard extraction attacks. In Figure 7, for the 1,000 samples identified as most vulnerable to strong MIA in the 140M Chinchilla-optimal model (Figure 1), we use the first 50 tokens of each sample (prefix) to see if the next 50 tokens (target suffix) is extractable. We use a sample's negative log-probability as a proxy for computing a probabilistic variant [25] of **discoverable extraction** [7]— the standard extraction metric in research and model release reports [2, 22, 24, 44, 55]. Discoverable extraction systematically underestimates extraction, relative to probabilistic extraction [15, 25]. We measure probabilistic extraction because we expect it to provide more reliable signal for memorization. A smaller negative log-probability implies that a sample is easier to extract [15].

After 1 epoch, LiRA is able to identify members with better-than-random AUC (Figure 1). Out of the 1,000 samples with the highest LiRA scores, 713 are indeed members. The largest suffix extraction probability is ≈0.0067—for the member sample in Figure 7 that has the (smallest) negative log probability of ≈5. Most samples—members and non-members alike—have negative log probabilities $> 100$, corresponding to probabilities on the order of $10^{-44}$ (measurements that do not register as successful extraction [15, 25]). Altogether, while much prior work draws a direct connection between MIA and extraction vulnerability [e.g., 4], our results suggest a more nuanced story: the success of a strong MIA on a given member does not necessarily imply that the LLM is more likely to generate that sample than would be expected under the data distribution [15, 25].

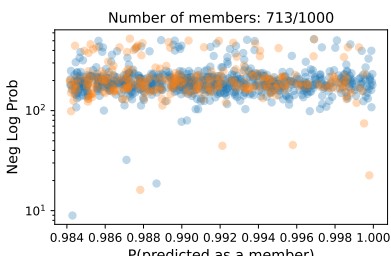

Figure 7: **Extraction for 140M.** Negative log-probability of the 50-token suffix (given the prior 50 tokens as prefix) for the 1,000 samples predicted most strongly as members.

## 7 Conclusion and future work

We perform dozens of experiments on thousands of GPT-2-like models (ranging from 10M–1B parameters) on enormous training datasets sampled from C4 (up to three orders of magnitude larger than those in prior work). In doing so, we address an urgent open question in ML privacy research: *are the fidelity issues of weaker attacks due to omitting reference models, or do they point to a deeper, more fundamental challenge with applying membership inference to large language models?* We uncover four novel groups of findings. While (1) strong MIAs can succeed on pre-trained LLMs (Section 3), (2) their success is limited (i.e., AUC<0.7) for LLMs trained using practical settings (Section 4). Even when attacks achieve above-chance AUC, (3) many per-sample membership decisions are very unstable; under training randomness, they are statistically indistinguishable from a coin flip (Section 5). Further, (4) the relationship between MIA vulnerability and related privacy metrics is not straightforward for LLMs (Section 6).

As the first work to perform large-scale strong MIAs on pre-trained LLMs, we are also the first to clarify the extent of actual privacy risk MIAs pose in this setting. By evaluating the effectiveness and limits of strong attacks, we are able to establish an upper bound on the accuracy that weaker, more feasible attacks can achieve. Together, our findings can guide others in more fruitful research directions to develop novel attacks and, hopefully, more effective defenses.

## Acknowledgments and Disclosure of Funding

Thank you to our anonymous reviewers, Nicholas Carlini, Zachary Charles, and Christopher De Sa for feedback on earlier versions of this work. A. Feder Cooper's contributions originated with a 2023–2024 student researcher position at Google Research. Franziska Boenisch has received funding from the European Research Council (ERC) under the European Union's Horizon Europe research and innovation programme (grant agreement No. 101220235)

IS is currently employed at a startup; CCC, MN, and KL are currently employed by OpenAI; MJ is currently employed by Anthropic; NM is currently employed by Meta.

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

# A  Membership inference attacks

**Security game, threat model, and notation.**   Membership inference is formalized as a security game between a challenger and an attacker (i.e., adversary).   Let $\mathcal{D}$ denote the underlying data-generating distribution over samples (and labels, if applicable). The challenger draws a finite training dataset $\mathbb{D} \sim \mathcal{D}^n$ and trains a target model $h$ on $\mathbb{D}$. A challenge record $\boldsymbol{x}$ is selected to be either a **member** ($\boldsymbol{x} \in \mathbb{D}$) or a **non-member** ($\boldsymbol{x} \notin \mathbb{D}$). The attacker is given query access to $h$ together with auxiliary resources and outputs a guess about $\boldsymbol{x}$'s membership; success means accuracy exceeding random guessing.

The strong attacks we study—LiRA and RMIA (Section 3.1 and Appendix B)—assume the attacker can (i) query $h$ on arbitrary inputs to obtain per-sample outputs (losses, logits, or confidence scores), and (ii) train **reference models** $f \in \Phi$ by replicating the target's training recipe on datasets drawn from the same population $\mathcal{D}$ that generated $\mathbb{D}$ (in practice, from a large proxy corpus approximating $\mathcal{D}$). For a fixed query sample $\boldsymbol{x}$, each reference's training dataset either *includes* $\boldsymbol{x}$ (IN) or *excludes* $\boldsymbol{x}$ (OUT), yielding a per-$\boldsymbol{x}$ partition:

$$\Phi_{\mathrm{IN}}(\boldsymbol{x}) \subseteq \Phi, \qquad \Phi_{\mathrm{OUT}}(\boldsymbol{x}) \subseteq \Phi, \qquad \Phi_{\mathrm{IN}}(\boldsymbol{x}) \cap \Phi_{\mathrm{OUT}}(\boldsymbol{x}) = \varnothing.$$

This is the **online** setting; the **offline** setting assumes access only to $\Phi_{\mathrm{OUT}}(\boldsymbol{x})$. Neither attack requires access to the target's parameters or $\mathbb{D}$; only queries to $h$ and attacker-trained references are needed. In research settings, one often controls both the target and references, which allows evaluation across many $\boldsymbol{x}$ with known membership. It is common (though not required) to choose $\Phi$ so that $|\Phi_{\mathrm{IN}}(\boldsymbol{x})| \approx |\Phi_{\mathrm{OUT}}(\boldsymbol{x})|$ for stability. We do so in this work.

**Observation statistics and membership scores.**   For any model $g$ and query sample $\boldsymbol{x}$, let

$$s(g, \boldsymbol{x}) \in \mathbb{R}$$

denote a fixed scalar **observation statistic** from $g$ on $\boldsymbol{x}$ (e.g., loss, negative $\log$-likelihood, or a monotone transform of confidence such as a logit). A **membership inference attack (MIA)** maps the available statistics for $\boldsymbol{x}$ (from $h$, and when used, from $\Phi$) to a real-valued **membership score** $\Lambda(\boldsymbol{x}) \in \mathbb{R}$, with larger values indicating stronger evidence that $\boldsymbol{x}$ is a member.

**Baseline (reference-free) loss attack [63].**   Using only the target's statistic,

$$\Lambda_{\mathrm{Loss}}(\boldsymbol{x}) \ = \ - s(h, \boldsymbol{x}),$$

so larger $\Lambda_{\mathrm{Loss}}(\boldsymbol{x})$ implies lower loss on $\boldsymbol{x}$. Any strictly monotone transform preserves ranking and therefore ROC-AUC. No reference models are used in this baseline approach. Stronger attacks use reference models to yield improved membership signal.

**Likelihood Ratio Attack (LiRA) [5].**   LiRA uses references to model per-sample IN/OUT distributions over the chosen statistic $s$. For a fixed $\boldsymbol{x}$, the attacker forms

$$\{s(f, \boldsymbol{x}) : f \in \Phi_{\mathrm{IN}}(\boldsymbol{x})\} \quad \text{and} \quad \{s(f, \boldsymbol{x}) : f \in \Phi_{\mathrm{OUT}}(\boldsymbol{x})\},$$

fits univariate models (typically Gaussians) to obtain densities $p_{\mathrm{IN}}(\cdot \mid \boldsymbol{x})$ and $p_{\mathrm{OUT}}(\cdot \mid \boldsymbol{x})$, and evaluates the target's statistic $s(h, \boldsymbol{x})$ under these densities to form a likelihood ratio:

$$\Lambda_{\mathrm{LiRA}}(\boldsymbol{x}) \ = \ \frac{p_{\mathrm{IN}}\big(s(h, \boldsymbol{x}) \,\big|\, \boldsymbol{x}\big)}{p_{\mathrm{OUT}}\big(s(h, \boldsymbol{x}) \,\big|\, \boldsymbol{x}\big)}. \tag{2}$$

The online variant uses both IN and OUT; the offline variant performs a one-sided test using only OUT. Working with $\log \Lambda(\boldsymbol{x})$ is common for numerical stability; since this is monotone, ROC-AUC is unchanged.

**Robust Membership Inference Attack (RMIA) [64].**   RMIA also compares the target model's statistic on the sample $\boldsymbol{x}$ to outputs for $\boldsymbol{x}$ from a set of reference models $\Phi$, but uses a different construction based on a *pairwise* likelihood ratio. This ratio is normalized by a reference population $\mathbb{Z}$ (e.g., a calibration set drawn from $\mathcal{D}$ or a held-out proxy). Define

$$\alpha(\boldsymbol{x}) \ = \ \frac{s(h, \boldsymbol{x})}{\mathbb{E}_{f \in \Phi} \, s(f, \boldsymbol{x})}. \tag{3}$$

The expectation in the denominator is approximated empirically over the trained references. To improve robustness, RMIA contextualizes this ratio relative to population $\mathbb{Z}$. For each $\boldsymbol{z} \in \mathbb{Z}$:

$$\alpha(\boldsymbol{z}) \;=\; \frac{s(h, \boldsymbol{z})}{\mathbb{E}_{f \in \Phi}\, s(f, \boldsymbol{z})}, \qquad L(\boldsymbol{x}, \boldsymbol{z}) \;=\; \frac{\alpha(\boldsymbol{x})}{\alpha(\boldsymbol{z})}. \tag{4}$$

The computed membership score aggregates the pairwise tests at a threshold $\gamma > 0$:

$$\Lambda_{\text{RMIA}}(\boldsymbol{x}) \;=\; \frac{1}{|\mathbb{Z}|} \sum_{\boldsymbol{z} \in \mathbb{Z}} \mathbf{1}\big[L(\boldsymbol{x}, \boldsymbol{z}) \geq \gamma\big]. \tag{5}$$

We focus on online (two-sided) variants of these attacks that use both IN and OUT references, as opposed to offline variants that only use OUT references.

**Decision rules and calibration.** Given a real-valued score $\Lambda(\boldsymbol{x})$ (e.g., $\Lambda_{\text{Loss}}$, $\Lambda_{\text{LiRA}}$, or $\Lambda_{\text{RMIA}}$), the attacker outputs a binary decision about the membership of $\boldsymbol{x}$ via

$$b(\boldsymbol{x}) \;=\; \mathbf{1}\{\Lambda(\boldsymbol{x}) \geq \tau\}.$$

To operate at a fixed false positive rate (FPR) $\eta$, it is convenient to write

$$b^{(\eta)}(\boldsymbol{x}) \;=\; \mathbf{1}\{\Lambda(\boldsymbol{x}) \geq \tau(\eta)\},$$

where $\tau(\eta)$ is calibrated for the target $h$ using non-members (i.e., samples not in $h$'s training subset $\mathbb{D}$). (We will sometimes refer to the training set as $\mathbb{D}_{\text{IN}}$, when we want to refer to the set of non-members as $\mathbb{D}_{\text{OUT}}$.)

**Calibration to non-members at fixed** FPR. Fix a target $h$ and an FPR $\eta \in [0, 1]$, and assume larger scores are indicate stronger evidence that $\boldsymbol{x}$ is a member. Let the non-member (OUT) set be $\mathbb{D}_{\text{OUT}}$ with size $N_{\text{OUT}} = |\mathbb{D}_{\text{OUT}}|$. (The attacker can draw i.i.d. samples from the population distribution $\mathcal{D}$, or use auxiliary data from the same source, independently of the training set, to form $\mathbb{D}_{\text{OUT}}$.) Write the scores as $\{\Lambda(\boldsymbol{x}) : \boldsymbol{x} \in \mathbb{D}_{\text{OUT}}\}$. The empirical CDF of OUT scores is

$$\widehat{F}_{\text{OUT}}(t) \;=\; \frac{1}{N_{\text{OUT}}} \sum_{\boldsymbol{x} \in \mathbb{D}_{\text{OUT}}} \mathbf{1}\{\Lambda(\boldsymbol{x}) \leq t\}.$$

We choose the right-continuous empirical $(1 - \eta)$-quantile

$$\tau(\eta) \;=\; \inf\{t : \widehat{F}_{\text{OUT}}(t^-) \;\geq\; 1 - \eta\}.$$

Equivalently, if $\Lambda_{(1)} \leq \cdots \leq \Lambda_{(N_{\text{OUT}})}$ are the sorted OUT scores, let $k = \lceil (1 - \eta)\, N_{\text{OUT}} \rceil$, $\bar{k} = \max\{j : \Lambda_{(j)} = \Lambda_{(k)}\}$, and set $\tau(\eta) = \Lambda_{(\bar{k}+1)} (\Lambda_{(N_{\text{OUT}}+1)} = +\infty)$.

We then apply the calibrated binary MIA decision rule

$$b^{(\eta)}(\boldsymbol{x}) \;=\; \mathbf{1}\{\Lambda(\boldsymbol{x}) \geq \tau(\eta)\}.$$

By construction, this guarantees (finite-sample, with ties handled conservatively) that

$$\widehat{\text{FPR}}(\eta) \;=\; \frac{1}{N_{\text{OUT}}} \sum_{\boldsymbol{x} \in \mathbb{D}_{\text{OUT}}} \mathbf{1}\{\Lambda(\boldsymbol{x}) \geq \tau(\eta)\} \;=\; 1 - \widehat{F}_{\text{OUT}}\big(\tau(\eta)^-\big) \;\leq\; \eta.$$

This is because taking the right-continuous quantile ensures that any mass tied at $\tau(\eta)$ is counted on the $\leq$ side of the CDF. Therefore, the realized FPR on OUT never exceeds $\eta$ (and may be smaller in the presence of ties).

**Common performance metrics.** Because MIAs are typically compared across operating points, it is typical to report ROC curves and AUC (threshold-agnostic), and—when reporting TPR at a fixed FPR—to set $\tau$ to achieve the target FPR. For RMIA, the internal pairwise threshold $\gamma$ controls the per-comparison likelihood ratio test, while the final decision threshold $\tau$ controls the operating point. Calibration may be global (single $\tau$) or conditional (e.g., per class/bucket). All monotone transforms of the score $\Lambda$ leave ROC-AUC invariant, while operating-point metrics (e.g., TPR at fixed FPR) depend on calibration.

**Practical note.** Calibrating FPR without knowledge of ground-truth membership can be challenging [65]. In our experiments, we control training and evaluation, so membership labels are known; this enables exact calibration and measurement at desired operating points.

# B Comparing membership inference attacks and observation statistics

At the beginning of this project, we considered two candidates for strong membership inference attacks to use in our experiments: the Likelihood Ratio Attack (LiRA) [5] and the Robust Membership Inference Attack (RMIA) [64]. Both attacks involve training reference models (Section 2) that enable the computation of likelihood ratios (which result in stronger attacks), though they differ in important ways. LiRA [5] estimates membership by comparing the loss of a sample $x$ in a target model to empirical loss distributions from reference models trained with and without $x$. In contrast, RMIA [64] performs and aggregates statistical pairwise likelihood ratio tests between $x$ and population samples $z$, using both reference models and $z$ to estimate how the inclusion of $x$ versus $z$ affects the probability of generating the observed model $\theta$ (Appendix A).

By leveraging signal from both models and population samples, Zarifzadeh et al. [64] observe that RMIA can outperform LiRA using fewer reference models. However, no prior work has compared these methods in the pre-trained LLM setting and with large numbers of reference models, leaving open the question of which attack fares better under these conditions.

In this appendix, we investigate this question for the first time, and our results clearly indicate that LiRA outperforms RMIA for a large number of reference models in the online setting (Appendix A). We observe limited cases where RMIA can outperform LiRA if the population dataset is large enough and the attack is performed for certain small numbers of reference models. However, we caution generalizing about comparative performance. LiRA seems to perform better with 1 or 2 IN references, while RMIA performs better with 4–16, and then LiRA once again outperforms RMIA for >16 IN references.

Overall, attacks with larger numbers of references perform better, as measured by ROC-AUC. Since our aim is to test the strongest attacks possible—to investigate an upper bound on strong MIA performance—this makes LiRA the best choice for our experiments. For those with smaller compute budgets that still wish to run strong attacks using ≈16 IN references, in some circumstances, RMIA may be a better choice.

Following from our discussion of the threat model for membership inference, and how it is implemented with slight variations for LiRA and RMIA (Appendix A), we next discuss our experiments comparing the performance of these two attacks. We first show how different choices for the observation statistic impact attack performance (Appendix B.1). This provides more detail about the choices we make in our overall experimental setup throughout the paper (introduced in Section 3). Then, we show our full results that compare the performance of LiRA and RMIA using different numbers of reference models (Appendix B.2), which lead us to choose LiRA for the experiments that follow.

For all experiments comparing LiRA and RMIA, we train 140M-parameter models on ≈7M samples, which equates to approximately 2.8B training tokens (i.e., what is optimal for this model size, according to Chinchilla scaling laws [26] with an over-training multiplier of 20).

## B.1 Different observation statistics

In our initial experiments in Section 3, we compare LiRA [5] and RMIA [64] to decide which strong attack to use. We also investigated the efficacy of different observation statistics for membership inference. We tested model loss and model logits (averaged over the entire sequence). For example, in Figure 8, we plot the ROC curve for using LiRA to attack a 140M model trained on ≈7M samples with 128 references. The plot shows the true positive rate (TPR) against the false positive rate (FPR) on a log-log scale, with one ROC curve each for logit and loss statistics. For the logit curve, ROC-AUC=0.576, while the loss curve has a higher ROC-AUC=0.678. This indicates that, in this setup, using loss as the observation statistic results in a more effective attack compared to using logits. Based on results like this, throughout this paper, we opt to use loss as observation statistic $s$.

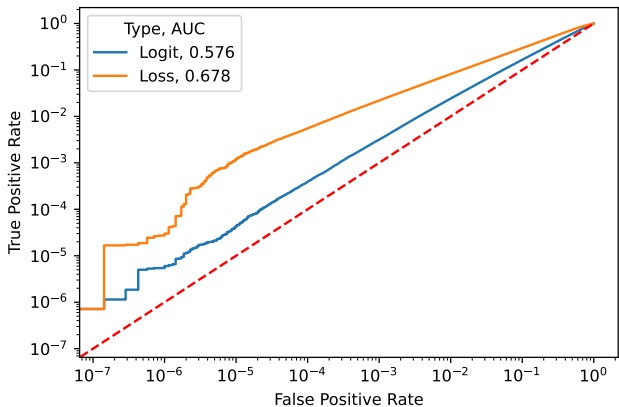

Figure 8: **Influence of observation statistic type on MIA Performance.** For the 140M model, we plot ROC curves to compare the efficacy of using model logits (AUC=0.576) and model loss (AUC=0.678) as observation statistics for membership inference with LiRA. In this setting, loss provides a stronger signal for distinguishing members from non-members.

## B.2 MIA attack performance for different numbers of reference models

Figure 9 compares LiRA and RMIA, showing ROC curves and ROC-AUC for different numbers of reference models. Figure 10 provides an alternate view of the same results, plotting ROC-AUC for both attacks as a function of reference models. LiRA's performance generally dominates RMIA's. LiRA continues to improve as we increase the number of reference models, while RMIA's effectiveness plateaus. However, with 4-16 IN references, RMIA surpasses the performance of LiRA. It essentially matches LiRA using 16 IN references. That is, with 4 references, LiRA exhibits ROC-AUC=0.594, which under-performs RMIA's corresponding ROC-AUC=0.643; but LiRA's ROC-AUC increases to 0.678 with 64 IN references, which outperforms RMIA's ROC-AUC=0.658.

Also note that RMIA exhibits a distinct diagonal pattern at low FPR (Figure 11). While RMIA aims to be a strong attack that is effective in low-compute settings, we find that a large population $\mathbb{Z}$ is necessary to obtain meaningful FPR at very low FPR thresholds. In particular, for a minimally acceptable $\text{FPR}_{\min}$, RMIA requires a population size $|\mathbb{Z}|$ that is $\frac{1}{\text{FPR}_{\min}}$. In practice, this is quite expensive, as RMIA's membership score is computed via pairwise comparisons with these $|\mathbb{Z}|$ reference points (i.e., there are $\mathcal{O}(|\mathbb{Z}|)$ pairwise likelihood ratio tests for target record $\boldsymbol{x}$, see Appendix A). In these initial experiments we only used $|\mathbb{Z}|$=10,000 samples. We measure performance of RMIA on larger population sizes below in Appendix B.2.1.

Overall, as noted in Section 3, while both attacks clearly beat the random baseline of ROC-AUC=0.5, neither is remarkably successful in this setting: regardless of the number of reference models, neither attack achieves that meets or exceeds ROC-AUC=0.7.

### B.2.1 Further experiments on RMIA

We now further investigate RMIA, decoupling its different components. We investigate removing the dependence on the population $\mathbb{Z}$, population sizes other than $|\mathbb{Z}|$=10,000, and varying threshold $\gamma$.

**Eliminating dependence on population $\mathbb{Z}$.** First, we consider the simplest form of RMIA (*simple*), eliminating its dependence on a population $\mathbb{Z}$ and using $\alpha(\boldsymbol{x})$ directly (Equation 3). Figure 11 shows the ROC curves for all three MIAs attacking one target model with 10M parameters, trained for 1 epoch on a training set size of $2^{19}$ samples. We use 128 reference models and consider $2 \times 2^{19}$=$2^{20}$ target records $\boldsymbol{x}$ with (as elsewhere) balanced membership/non-membership to analyse MIA. We find all three attacks reach similar ROC-AUC values.

We also gauge MIA performance by evaluating the TPR at low, fixed FPR. To understand the values RMIA reaches for TPR at low FPR, an important subtlety arises from the entropy of the score distribution. Attacks that produce very coarse membership scores inherently limit achievable

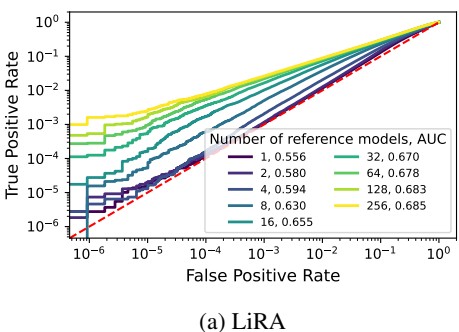

(a) LiRA

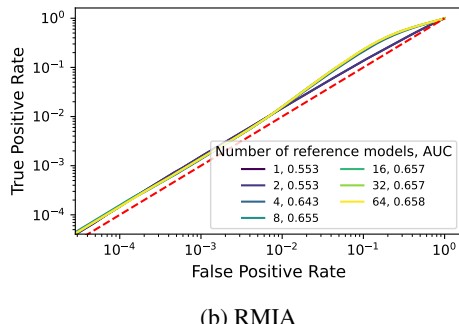

(b) RMIA

Figure 9: **Comparing LiRA and RMIA.** We attack a 140M-parameter model, with the target and references trained on $\approx$7M samples. ROC curves illustrate the effectiveness of (**a**) LiRA [5] and (**b**) RMIA [64] for different numbers of reference models. As we increase the number of references, LiRA's performance surpasses RMIA's, measured by ROC-AUC. These plots show the number of IN references. (There are $2\times$ as many references in total, accounting for OUT.)

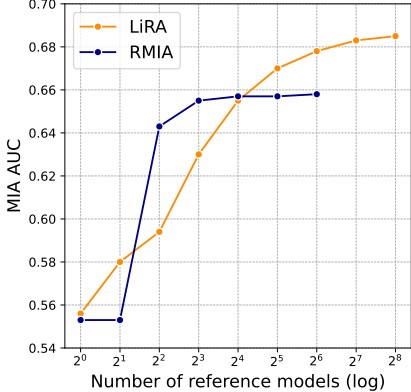

Figure 10: **Comparing LiRA and RMIA.** As an alternative view of Figure 9, we plot the ROC-AUC achieved by both attack methodologies for an increasing number of reference models. As the number of references increases, LiRA's performance continues to improve, while RMIA's gains saturate. Overall, LiRA is the stronger attack. This plot also only shows the number of IN references on the $x$-axis (there are the same number of OUT).

TPR at very low FPR. For example, as RMIA compares $\alpha(\boldsymbol{x})$ to $\alpha(\boldsymbol{z})$ for all $\boldsymbol{z} \in \mathbb{Z}$ to compute its membership score $\Lambda_{\mathrm{RMIA}}(\boldsymbol{x})$ (Equation 5), there are maximally $|\mathbb{Z}|$ unique values $\Lambda_{\mathrm{RMIA}}(\boldsymbol{x})$ can take for all $\boldsymbol{x}$. This limits the score's entropy and the possibility of achieving a meaningful TPR at very low FPR. This explains the diagonal pattern for RMIA in Figure 11, where $|\mathbb{Z}|$=10,000. By contrast, both LiRA and RMIA (simple) provide a membership score that is not limited in entropy, leading to more meaningful values for TPR at lower FPR.

**Increasing the population size** $|\mathbb{Z}|$**.** We next test further increasing the size of the population $\mathbb{Z}$ when computing RMIA. For the same setup as Figure 11, Figure 12 shows how MIA performance varies with the size of $\mathbb{Z}$. We observe very similar values for RMIA (simple) and RMIA ROC-AUC for all population sizes that we test. When examining TPR at low FPR, we find that increasing $|\mathbb{Z}|$ improves the MIA performance. Indeed, the increased entropy in $\Lambda_{\mathrm{RMIA}}(\boldsymbol{x})$ now allows the attack to reach meaningful values of TPR for FPR as low as $10^{-6}$. Notably, for all values of $|\mathbb{Z}|$ we consider, LiRA still outperforms RMIA at low FPR, while the $|\mathbb{Z}|$ likelihood comparisons in RMIA for every target record $\boldsymbol{x}$ also incur additional computational cost.

**Varying threshold** $\gamma$**.** Finally, we evaluate RMIA under varying threshold $\gamma$. As $\gamma$ increases, in Equation 4, it becomes less likely that $\alpha(\boldsymbol{x})$ sufficiently exceeds $\alpha(\boldsymbol{z})$ for many $\boldsymbol{z} \in \mathbb{Z}$ to count toward the score—i.e., that $\alpha(\boldsymbol{x})/\alpha(\boldsymbol{z}) \geq \gamma$ (Equation 5).

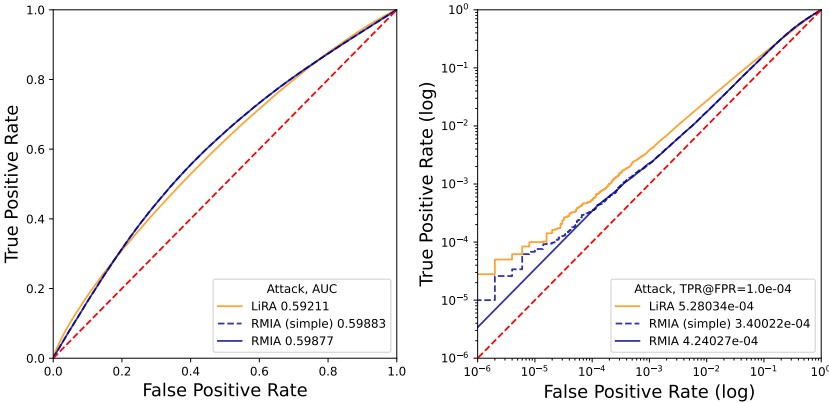

Figure 11: **Comparing LiRA, RMIA (simple) and RMIA.** Attacking a 10M-parameter model trained for 1 epoch with a training set size of $2^{19}$ samples.

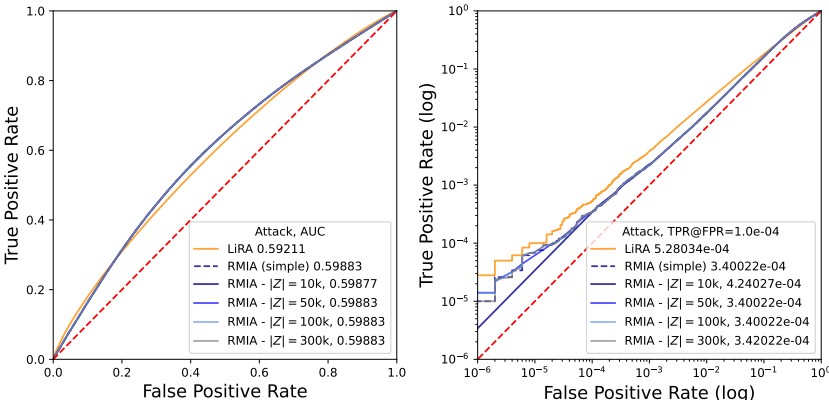

Figure 12: **Performance of RMIA for different population sizes** $|\mathbb{Z}|$**.** We attack a 10M-parameter model trained for 1 epoch with a training set size of $2^{19}$ samples.

Again for the same setup, Figure 13 shows how RMIA performs for varying values of $\gamma$, considering both $|\mathbb{Z}|$=10,000 (Figure 13a) and $|\mathbb{Z}|$=300,000 (Figure 13b). While MIA ROC-AUC remains relatively stable as $\gamma$ increases, the TPR at low FPR varies. For $|\mathbb{Z}|$=10,000, the TPR at FPR=$10^{-4}$ decreases for increasing values of $\gamma$, reaching 0 for $\gamma{\geq}1.1$. This is due to the reduced granularity of RMIA's membership score: for larger $\gamma$, fewer $\boldsymbol{z}$ satisfy $\alpha(\boldsymbol{x})/\alpha(\boldsymbol{z}) \geq \gamma$; this constrains the entropy of the RMIA score, making it harder to reach meaningful values of TPR at low FPR. A larger reference population ($|\mathbb{Z}|$=300,000) mitigates this issue, allowing meaningful TPR even at low FPR for similar $\gamma$ values.

Taking these three sets of results together, we find LiRA to outperform RMIA when a sufficiently large number of reference models is available, especially in the low-FPR regime. Since our aim is to study the strongest attacks, we adopt LiRA as the primary attack throughout our experiments.

### B.3 MIA performance in the offline setting

As stated in Section 2 and Appendix A, the literature distinguishes between online and offline settings for reference-based MIAs [5, 64]. In the online setting, the attacker has access to reference models trained on data including the target sample $\boldsymbol{x}$ ($\Phi_{\text{IN}}$) and excluding it($\Phi_{\text{OUT}}$). In the offline setting, the attacker only has access to $\Phi_{\text{OUT}}$. Throughout this work, we consider the strongest attacker, and thus report all results in the online setting.

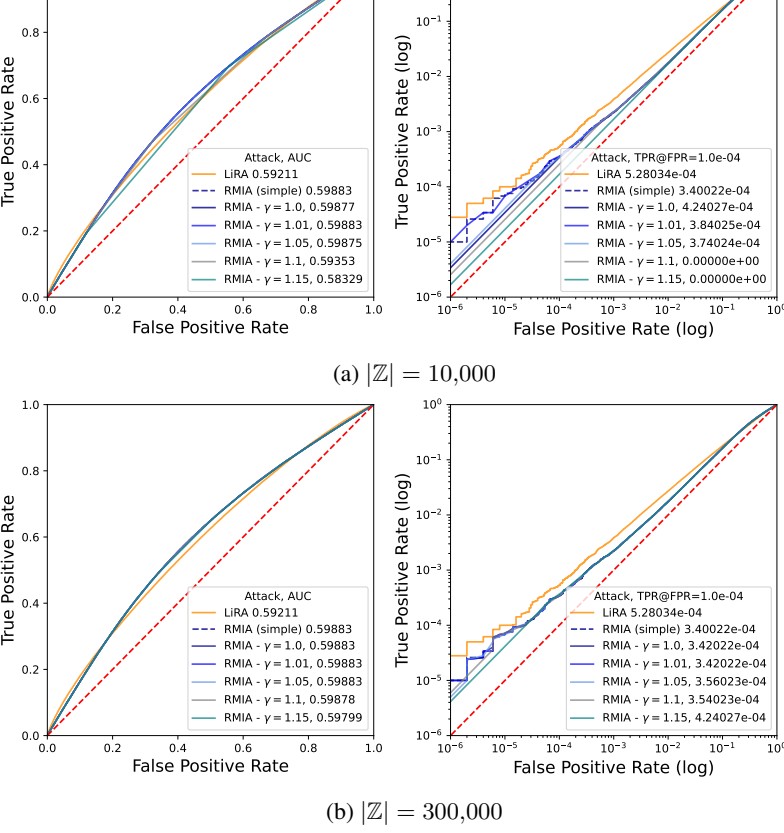

(a) $|\mathbb{Z}| = 10{,}000$

(b) $|\mathbb{Z}| = 300{,}000$

Figure 13: **Performance of RMIA for varied** $\gamma$**.** We attack a 10M-parameter model trained for 1 epoch with a training set size of $2^{19}$ samples, varying the threshold $\gamma$ used to compute $\Lambda_{\text{RMIA}}$.

For completeness, we instantiate MIAs in the offline setting in the same experimental setup as considered above for our additional RMIA tests (Appendix B.2.1). We test the offline versions for both LiRA and RMIA, as originally proposed in Carlini et al. [5] and Zarifzadeh et al. [64], respectively.

For LiRA, without $\Phi_{\text{IN}}$, we are unable to approximate the probability $p_{\text{IN}}\big(s(h, \boldsymbol{x})\big)$ (Equation 2), and so just consider the one-sided hypothesis test instead of the likelihood ratio:

$$\Lambda_{\text{LiRA,offline}}(\boldsymbol{x}) = 1 - p_{\text{OUT}}\big(s(h, \boldsymbol{x})\big).$$

For RMIA, we now compute the denominator in $\alpha(\boldsymbol{x})$ (Equation 3) by taking the expectation over the reference models that are available to the attacker, i.e.:

$$\alpha_{\text{offline}}(\boldsymbol{x}) \;=\; \frac{s(h, \boldsymbol{x})}{\mathbb{E}_{f \in \Phi_{\text{OUT}}} s(f, \boldsymbol{x})}.$$

Note that Zarifzadeh et al. [64] propose to further adjust the denominator by using a variable $a$ (their Appendix B.2.2) to better approximate the $\mathbb{E}_{f \in \Phi} s(f, \boldsymbol{x})$, when only references $\Phi_{\text{OUT}}$ are available. We set $a{=}1$ and just compute the empirical mean across all reference models in $\Phi_{\text{OUT}}$ to approximate the expectation in the denominator. We then compute $\alpha_{\text{offline}}(\boldsymbol{z})$ and use membership inference score

$$\Lambda_{\text{RMIA,offline}}(\boldsymbol{x}) \;=\; \frac{1}{|\mathbb{Z}|} \sum_{\boldsymbol{z} \in \mathbb{Z}} \mathbf{1}\left[L_{\text{offline}}(\boldsymbol{x}, \boldsymbol{z}) \geq \gamma\right], \quad \text{where} \quad L_{\text{offline}}(\boldsymbol{x}, \boldsymbol{z}) = \frac{\alpha_{\text{offline}}(\boldsymbol{x})}{\alpha_{\text{offline}}(\boldsymbol{z})}.$$

Figure 14 compares the MIA performance between the online and offline setting, for LiRA, RMIA (simple) (which does not use the reference population $\mathbb{Z}$, Appendix B.2.1), and RMIA; we set $\gamma{=}1$ and $|\mathbb{Z}|{=}300{,}000$. We again attack a 10M-parameter model trained for 1 epoch, using a training set size of $2^{19}$ samples. We use 128 reference models for the online setting and 64 in the offline setting (on average, per target sample).

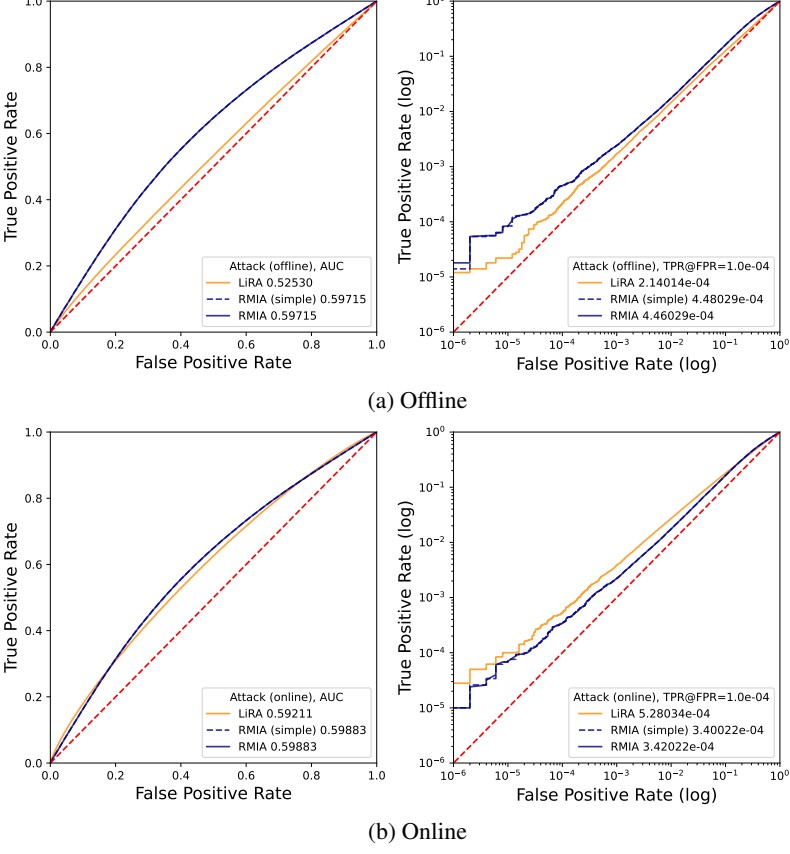

(a) Offline

(b) Online

Figure 14: **MIA performance in the offline and online setting.** We attack a 10M-parameter model trained for 1 epoch with a training set size of $2^{19}$ samples, considering 128 references in the online setting and only the corresponding models $\Phi_{\text{OUT}}$ in the offline setting (on average 64 references per $x$).

We find that, in this configuration and with this number of reference models, offline RMIA outperforms offline LiRA, in terms of both ROC-AUC and TPR at low fixed FPR. This suggests that RMIA's offline variant more accurately captures membership signal compared to the one-sided hypothesis test used in offline LiRA. In contrast, in the online setting, LiRA and RMIA achieve similar ROC-AUC, with LiRA performing better than RMIA in the low-FPR regime.

## C More experiments on Chinchilla-optimal models

In this appendix, we provide additional details on our experiments involving LiRA attacks on Chinchilla-optimal [26] models of different sizes in Section 3.2: 10M, 44M, 85M, 140M, 489M, and 1018M. We summarize training hyperparameters in Section G.

**Observing changes in loss during training.** In Figure 15a, we show the decrease in validation loss over a single epoch. The $x$-axis represents the fraction of the training epoch completed (from 0.0 to 1.0), and the $y$-axis shows the corresponding loss. As expected, all models exhibit a characteristic decrease in loss as training progresses. Larger models (namely, 489M and 1018M) demonstrate faster convergence to lower loss values, reflecting their increased capacity to fit the training data. They also maintain a lower loss throughout the epoch compared to smaller models (10M–140M).

**Investigating the role of learning rate schedule.** In the Chinchilla-optimal setting, we also investigate the role of hyperparameters on MIA performance. In Figure 15b, we show ROC curves that compare the MIA vulnerability (with LiRA) of 140M-parameter models (trained on ≈7M records, with 128 reference models), where we vary the learning rate schedule: Linear (AUC=0.676), Cosine (no global norm clipping, AUC=0.660), Cosine (no weight decay, AUC=0.673), and standard

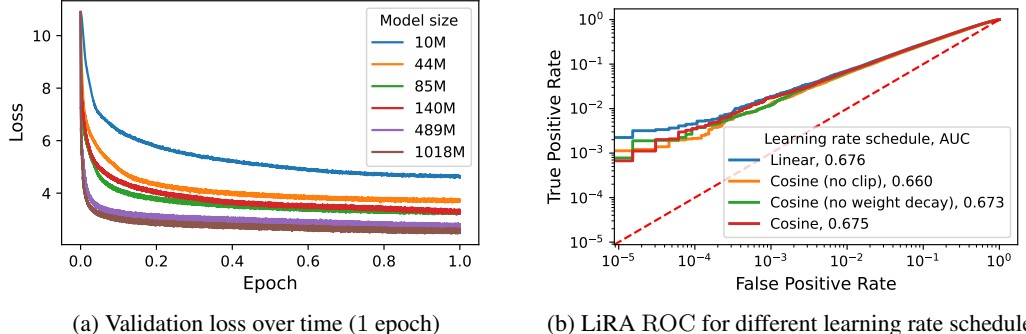

(a) Validation loss over time (1 epoch)  (b) LiRA ROC for different learning rate schedules

Figure 15: **Investigating training dynamics hyperparameters.** (**a**) Validation throughout the 1 training epoch for our experiments involving Chinchilla-optimal trained models of various sizes. (**b**) The effect of learning rate schedule on LiRA's attack success for 140M models using 128 references.

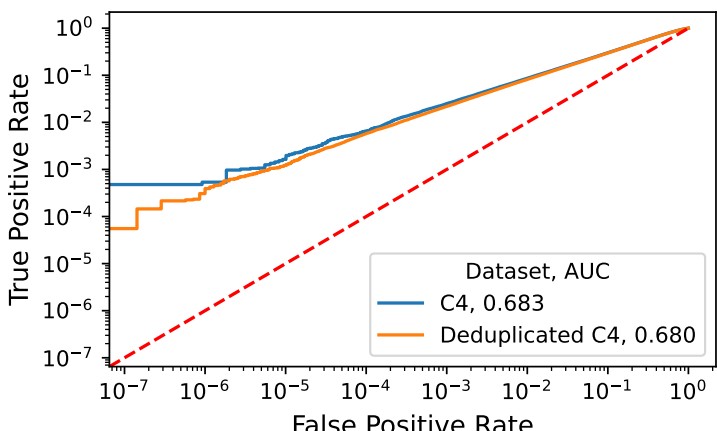

Figure 16: **The role of duplicates on MIA vulnerability.** We observe no significant differences (particularly as FPR increases) between models trained on C4 and de-duplicated C4.

Cosine (AUC=0.675). As with all of our ROC plots, the TPR is plotted against the FPR on a log-log scale. The ROC-AUC values for each curve are relatively close. This indicates that, while there are some minor differences in attack performance, the choice of learning rate schedule among those tested does not lead to drastically different MIA outcomes.

## D  Additional experiments exploring the limits of LiRA

In this appendix, we provide additional experiments that explore the limits of LiRA when there are duplicate samples in the training data, and (complementing results in Section 4) when there are varying numbers of training epochs and varied dataset size.

**Investigating the role of duplicate training samples.**  Given the relationship between MIA and memorization, and that prior work observes an important relationship between memorization and training-data duplication [29], we test the relationship between MIA vulnerability and the presence of duplicate training samples. In Figure 16, we test the Chinchilla-optimally trained 140M model on C4 and a de-duplicated version of C4. We de-duplicate C4 according to methodology described in Lee et al. [29], where we remove sequences that share a common prefix of at least some threshold length. This reduced the C4 dataset size from 364,613,570 to 350,475,345 samples.

We observe that the presence of duplicates has a negligible impact on AUC: it is 0.683 for C4, and 0.680 for de-duplicated C4. In other words, at least in terms of average attack success, the presence of duplicates does not seem to have a significant impact. However, further work is needed to assess

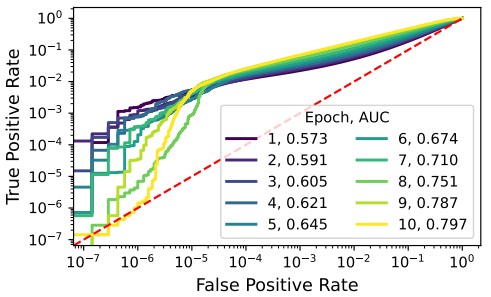
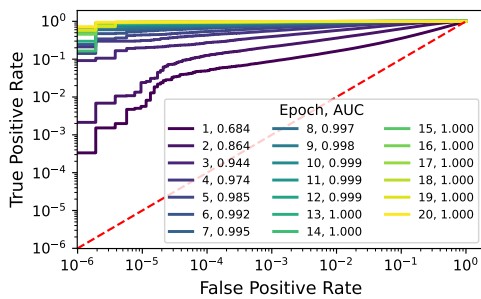

(a) 140M model, ≈7M samples, 10 epochs.    (b) 140M model, ≈500$K$, 20 epochs.

Figure 17: **Over-training and MIA.** ROC curves demonstrate that MIA success significantly increases as models are trained for more epochs. (**a**) The 140M model shows AUC rising from 0.573 (1 epoch) to 0.797 (10 epochs). (**b**) Attacking a 140M model trained on a smaller dataset shows a rapid escalation in AUC, from 0.604 (1 epoch) to near-perfect membership inference (AUC=1) by 13-20 epochs, highlighting that overfitting from prolonged training severely heightens privacy risks.

how attack success changes with more stringent de-duplication, since our de-duplication procedure only removed 10M samples from the dataset.

**Varying training epochs and dataset size.** In Figure 17, we reduce the training set size from ≈7M (Figure 17a) to $2^{19}$≈500K samples (Figure 17b) on the 140M model and train for 10 (Figure 17a) and 20 epochs (Figure 17b). Both figures show ROC curves that illustrate how MIA vulnerability changes with an increasing number of training epochs. The goal of these experiments is to investigate if MIA becomes better with more training epochs, and if so, how attack performance improves over epochs as a function of training dataset size.

For the 140M model trained on ≈7M samples for 10 epochs, the AUC increases with more epochs, starting from 0.573 at 1 epoch and reaching 0.797 at 10 epochs. For the 140M model trained on ≈500K samples for 20 epochs, we observe a more dramatic increase in MIA vulnerability. The AUC starts at 0.604 for 1 epoch, rapidly increases to 0.864 by 2 epochs, 0.944 by 3 epochs, and approaches perfect MIA (AUC close to 1.000) after 13 epochs. Of course, both of these experiments are effectively sanity checks. We intentionally over-train in both, and use a relatively small training dataset size in the second.

**Full results for various-sized Chinchilla-trained models and fixed training set size.** We provide full results for attacking Chinchilla-optimal models of various sizes for 1 epoch (Figure 2b), and attacking various model sizes trained on a fixed dataset of ≈8.3M samples for 1 epoch (Figure 4b). Both of these figures in the main paper show how TPR varies at fixed FPR in line plots. Here, in Figures 18 and 19, we give individual ROC curves for experimental results summarized in each of those figures, respectively. For each subplot, each line indicates a different target model that we attack. As discussed previously, some larger models appear to have more variance in their ROC curves over different experimental runs. In Figure 19i, we see that although AUC is similar over different target models, there is catastrophic attack failure for one model at small FPRs.

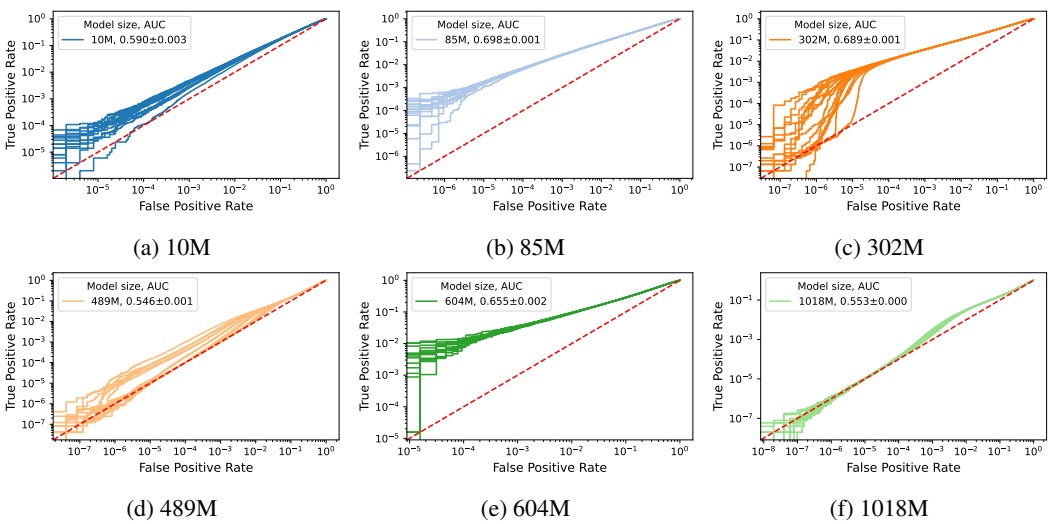

Figure 18: **ROC curves and AUC for Figure 2b.** We attack different model sizes trained on the Chinchilla-optimal number of tokens. In each subplot, each line indicates a different attacked target.

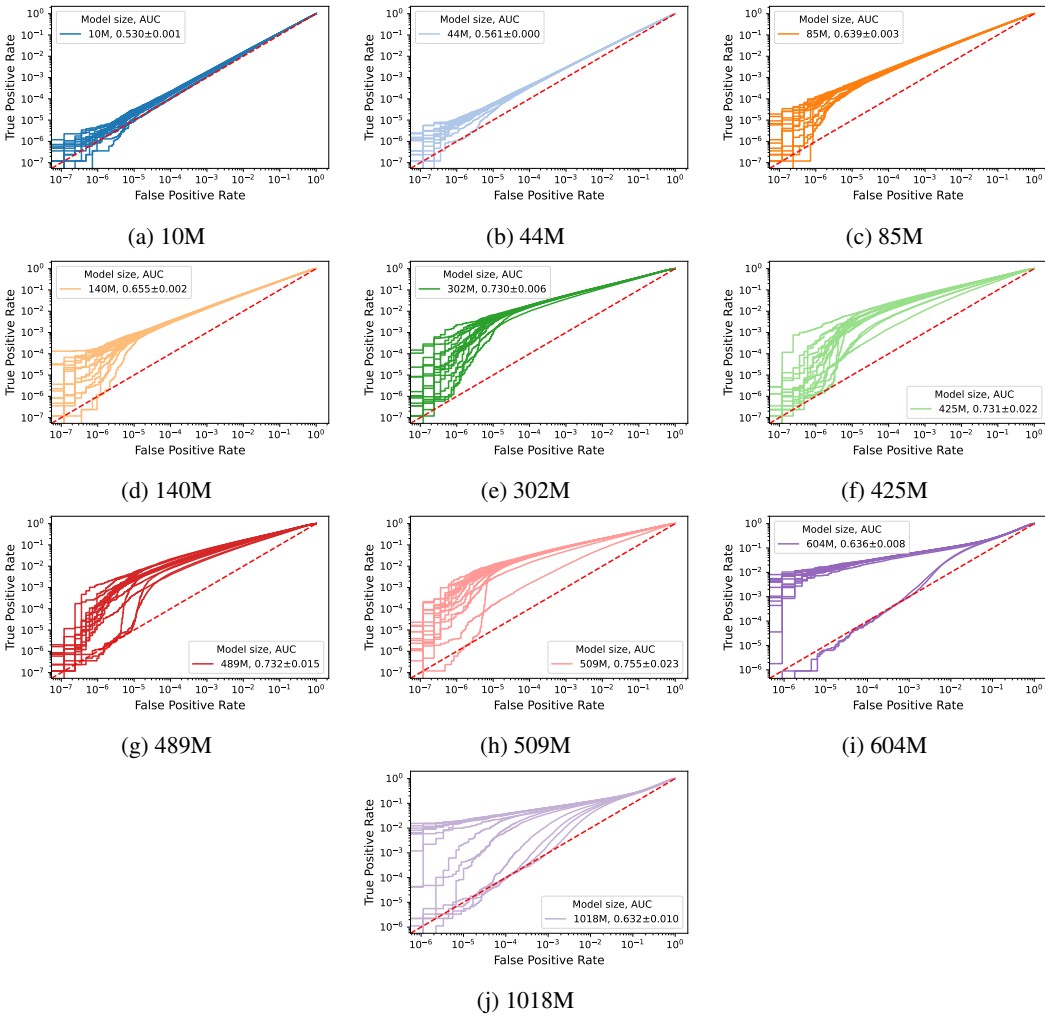

Figure 19: **ROC curves and AUC for Figure 4b.** We attack different model sizes trained on the same number of samples (≈8.3M). In each subplot, each line indicates a different attacked target.

**Varying reference models for all Chinchilla-optimally trained model sizes** In Figure 20, we replicate the experiments in Figure 18, but we vary the number of references. Each row in the figure is for a different-sized model. Each column uses a different number of total references (IN plotted) to perform the attack. We attack 8 targets trained on different training data subsamples in each plot.

Unsurprisingly, MIA improves as we use more references. This mirrors our findings in Figure 9a. The key point of these figures is to show the general pattern of where the ROC curve is relative to the reference line $y=x$. We also show that there is variance (in the insets) across attack runs for the same model size. These are not to be taken as detailed results that should be closely examined. (This is why the plots are not very large.) We investigate instability in Section 5 and Appendix E.

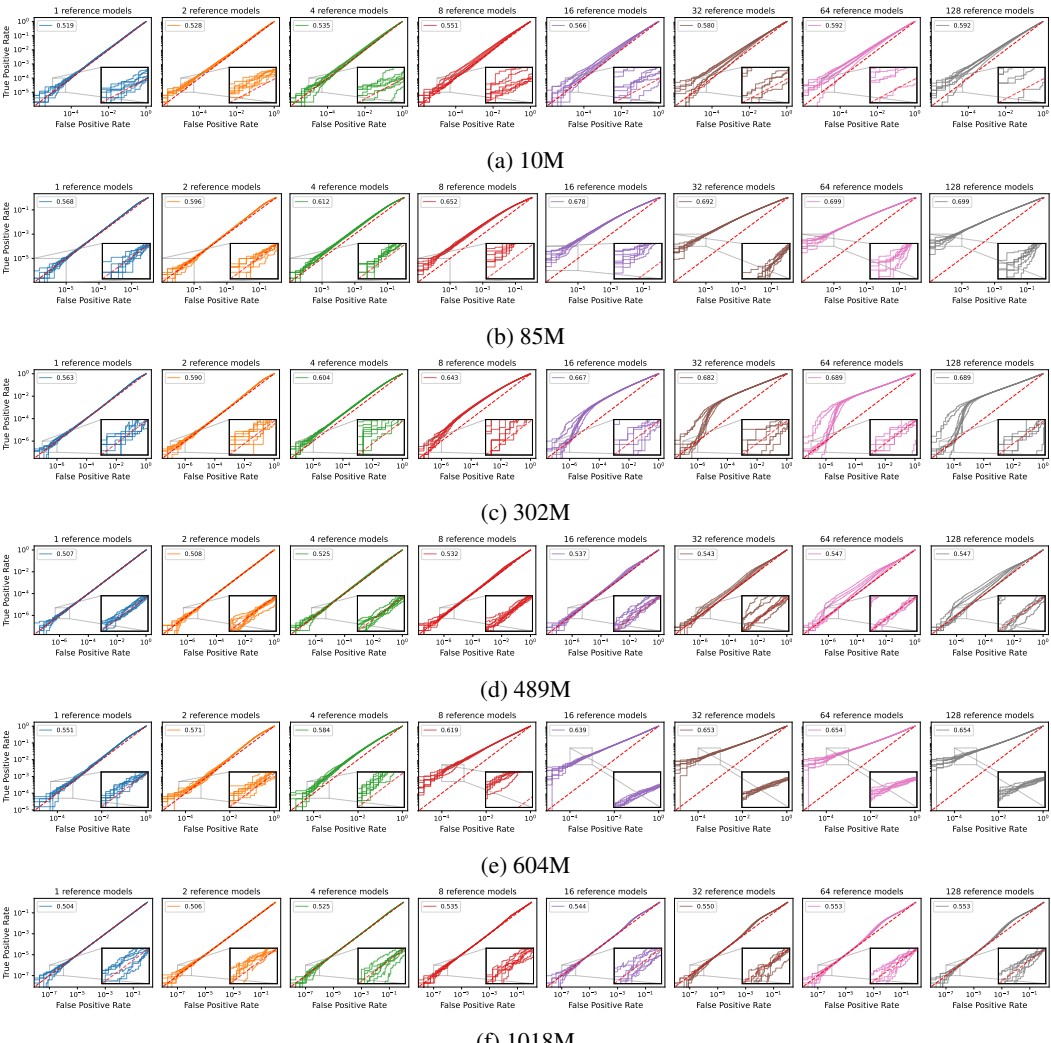

(a) 10M

(b) 85M

(c) 302M

(d) 489M

(e) 604M

(f) 1018M

Figure 20: **Extended ROC curves and AUC for Figure 2b.** For each subplot, each line indicates a different target model that we attack. Each row is a different model size. Each column represents using LiRA with a different number of total reference models. Each subplot also records the average AUC across attacks on different targets.

# E Investigating instability in per-sample membership decisions

As noted in Section 5, we observe substantial *per-sample* instability in membership decisions. We also notice significant variability in ROC-AUC across attacks in Figure 20. However, because standard attack metrics such as ROC-AUC are *aggregates* over samples and decision thresholds; they report metrics according to average FPR/TPR over many samples. As such, they can mask this instance-level variability. We visualize and quantify individual-sample instability, and connect our analysis to prior work in other areas of statistics and machine learning.

In Appendix E.1, we provide extended results for Section 6.1 on variation in per-sample true positive probabilities, and then in Appendix E.2 we include more results and discussion on flip rate (Section 5).

## E.1 Variation in per-sample true positive probabilities

For the 140M model, we plot the mean and standard deviation of the per-sample true positive probabilities, $\Pr(\text{predicted as member}|\text{member})$ for $2^{24}$=16,777,216 samples. For each sample, we compute variance across 64 target models (for which the sample is a member); overall, this experiment trained 128 models (140M size) on different random splits of the $2^{24}$ samples. We compute $\Pr$ predicted as a member|member, using $\frac{p_{\text{IN}}(\cdot|\boldsymbol{x})}{p_{\text{IN}}(\cdot|\boldsymbol{x})+p_{\text{OUT}}(\cdot|\boldsymbol{x})} > 0.5$ to determine if the sample is predicted as a member (Section 6). We loop over each model, selecting it as the target model and the remainder as reference models used for LiRA. Since each sample had a probability of 0.5 for inclusion in the training set, for each sample, we have on average 64 target models where the sample was in training and 64 for which it was not.

In Figure 21, we provide three plots that give different views of the same data. Figure 21a plots the true positive probability for each member. We sort members by the mean value of their true positive probability (i.e., the mean of $\Pr(\text{predicted as member}|\text{member})$ over 64 target models), so member ID corresponds to this ordering. We also show the variance over the 64 target models by plotting the standard deviation.

Together, Figures 21b and 21c provide an alternate view of Figure 21a. Figure 21b plots the histogram of the mean $\Pr(\text{predicted as member}|\text{member})$ for members across their respective 64 target models. The average across these mean true positive probabilities for each member is 0.543. However, note the distribution of per-sample means: while the across-sample average of the per-sample means is 0.543, a substantial mass of members exhibits mean $\Pr(\text{predicted as member}|\text{member})>0.6$. The spread is large: the average per-sample standard deviation is 0.143, with many members exceeding a standard deviation of 0.2.

Overall, variance is significant. The individual member true positive probabilities for each target are, when considered together, highly unstable. This variance can help explain why attack ROC-AUC is perhaps lower than one might have hoped; there is considerable variance in the underlying sample binary decisions. Altogether, this provides additional nuance concerning the extent of (alternatively, the limits of) attack robustness.

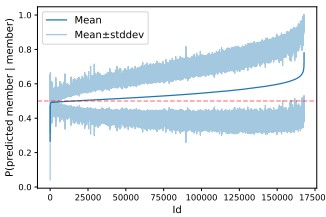 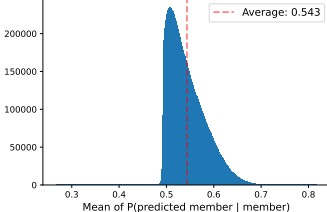 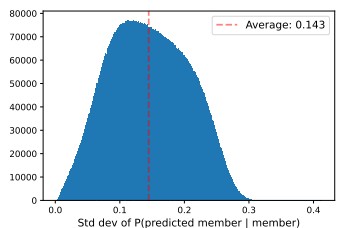

(a) Per-sample true positive probabilities (mean $\pm$ standard deviation), ordered from smallest to largest.

(b) Histogram of average per-sample true positive probabilities from Figure 21a.

(c) Histogram of standard deviation of per-sample true positive probabilities from Figure 21b.

Figure 21: **Instability of per-sample true positive probabilities.** For each of $2^{24}$ samples $\boldsymbol{x}$, we compute the mean and standard deviation of $\Pr(\text{predicted as member} \mid \text{member})$ across $B$=64 target models. (**a**) After sorting samples by their mean, the mean and one standard deviation band. A histogram of (**b**) these per-sample means and (**c**) of the corresponding standard deviations.

## E.2 Analyzing per-sample membership decision instability

This appendix deepens our analysis of per-sample MIA binary decision instability. We formalize **flip rate** [14]—the metric we use to measure instability at the per-sample decision level—and its unbiased empirical estimator (Appendix E.2.1). We then explain how we measure flip rate in the MIA setting used in our experiments, and why the metric is informative for strong MIAs (Appendix E.2.2). We connect our results to prior work on model/predictive multiplicity [3] (Appendix E.2.3).

Then, we derive an exact acceptance band (at level $\alpha$) for deciding when a sample's binary membership decisions are statistically indistinguishable from a coin flip; for a finite number of targets $B$ and acceptance level $\alpha$, we obtain the resulting flip cutoff $t_\alpha(B)$ that we deem the minimum required for $\boldsymbol{x}$'s MIA decisions to be called "statistically indistinguishable from a coin flip at level $\alpha$." (Appendix E.2.4). Using these tools, we present extended empirical results for two model sizes, 140M and 302M (Appendix E.2.5). We estimate how much of standard attack performance (ROC-AUC) can be attributed to coin-flip-like decisions as opposed to reliable inference (Appendix E.2.6). We also discuss additional ways to interpret the purpose and results of these experiments (Appendix E.2.7). Finally, we discuss a more intuitive, but significantly more expensive, alternative approach for computing per-sample MIA decision instability, which, in principle, an attacker could compute (Appendix E.2.8). We expect this procedure to surface qualitatively similar instability that we observe with the setup we test.

For reference, the acceptance-band cutoff values used in the figures/tables are $t_{0.05}(125) \approx 0.490$ and $t_{0.05}(127) \approx 0.487$ for the 140M and 302M models, respectively.

**Key points.** This is a long appendix, so we summarize key points here. For a fixed sample $\boldsymbol{x}$ and FPR $\eta$, we care whether the *binary* membership decision produced by LiRA is *reliable* (stable across equally plausible targets) or *statistically indistinguishable from a coin flip* with respect to target training randomness. We compute flip rate with respect to the seed-induced distribution $\mu$. Different seeds reflect realistic training randomness (e.g., batch order); aggregate metrics are stable, indicating that $\mu$ is not pathological/degenerate.

High flip rate (near 0.5) means the decision for $\boldsymbol{x}$ is effectively a coin flip across plausible targets, so a true positive on a particular target is not evidence of *reproducible* inference for $\boldsymbol{x}$; it is a lucky draw. Aggregate ranking performance (e.g., AUC) can still be $> 0.5$, but that is a different claim about *averages*. We call the MIA decision for $\boldsymbol{x}$ "indistinguishable from a coin flip at level $\alpha$" if, under the exact two-sided binomial test with $B$ votes $K \sim \mathrm{Binomial}(B, \theta)$, we fail to reject $H_0 : \theta = 0.5$, where $\theta := \mathrm{Pr}_{r \sim \mu}\big[b_r^{(\eta)}(\boldsymbol{x}) = 1\big]$. This yields a concrete cutoff $t_\alpha(B)$ on $\widehat{\mathrm{flip}}_{\eta,B}(\boldsymbol{x})$ via the equal-tails acceptance region under $H_0$; samples with $\widehat{\mathrm{flip}}_{\eta,B}(\boldsymbol{x}) \geq t_\alpha(B)$ are deemed indistinguishable from a coin flip. (See Appendix E.2.4 for the derivation; the values we use in practice are noted above.) "Indistinguishable from a coin flip at level $\alpha$" is a standard, finite-sample exact test with a clear, observable cutoff $t_\alpha(B)$.

Another way to understand these results is to see that, if $\theta = \mathrm{Pr}_{r \sim \mu}[b_r^{(\eta)}(\boldsymbol{x}) = 1] \approx 0.5$, then

$$\Pr_{r,r' \sim \mu}\big[b_r^{(\eta)}(\boldsymbol{x}) = b_{r'}^{(\eta)}(\boldsymbol{x})\big] = 1 - \mathrm{flip}_\eta(\boldsymbol{x}) \approx 0.5.$$

Retraining the same pipeline on the same data would reproduce the *same* decision for $\boldsymbol{x}$ only about half the time. This is the operational meaning of an "indistinguishable from a coin flip" per-sample MIA decision. Importantly, this claim concerning flip rate is about MIA decisions, not the underlying scores. Even if LiRA scores for $\boldsymbol{x}$ carry some signal, decision instability can be high when the calibrated threshold $\tau_r(\eta)$ varies across seeds; AUC may remain non-trivial while flip rate is high. Our claim is specifically about the reliability of *per-sample decisions*. (We further discuss interpretation details in Appendix E.2.7.) We apply this per-sample test and report descriptive counts across many $\boldsymbol{x}$; the inferential claim is *per sample* ("indistinguishable from a coin flip at level $\alpha$"), which is appropriate for the setup of the MIA security game.

The classifier threshold $\tau_r(\eta)$ is calibrated on non-members for each target (trained with seed $r$), anchoring non-member decisions to their own distribution while leaving members more exposed to seed-induced score and $\tau$ variation, especially where IN/OUT overlap. This is a feature of the real attack protocol, not an evaluation artifact. With finite $B$, some truly non-coin-flip-like $\boldsymbol{x}$ could be labeled coin-flip-like by chance. The exact binomial test controls Type-I error at level $\alpha$; the

acceptance band and $t_\alpha(B)$ make the rule explicit. In this decomposition (e.g., contributions to TPR and AUC), we filter only the "coin-flip-like" band (not all highly unstable cases like $[0.4, t_\alpha(B))$), so reported performance is a conservative *upper bound* on reliable inference.

### E.2.1 Measuring instability of individual membership decisions with flip rate

To complement our measurements of typical metrics from work on MIA, we adopt a metric from Cooper et al. [14] for measuring per-sample MIA decision instability. We first review this metric, then specify our MIA-calibrated version and its unbiased estimator.

**Self-consistency across a distribution of models.** Let $g \sim \nu \equiv \nu_{\mathcal{A},\mathcal{D}}$ denote a model drawn from the distribution induced by training algorithm $\mathcal{A}$ (a function of a random seed) with training data from distribution $\mathcal{D}$. For a binary decision rule $b_g(\boldsymbol{x}) \in \{0, 1\}$ (e.g., $b_g(\boldsymbol{x}) = \mathbf{1}\{g(\boldsymbol{x}) \geq \tau\}$), Cooper et al. [14] define the **self-consistency** at $\boldsymbol{x}$ as the pairwise agreement probability under two i.i.d. draws:

$$\mathrm{SC}(\boldsymbol{x}) \;:=\; \Pr_{g,g' \overset{\mathrm{i.i.d.}}{\sim} \nu} \big[b_g(\boldsymbol{x}) = b_{g'}(\boldsymbol{x})\big]. \tag{6}$$

For such binary decisions, $\mathrm{SC}(\boldsymbol{x}) \in [0.5, 1]$: values near 1 indicate stability among binary decisions for $\boldsymbol{x}$ in spite of randomness in the training process; values near $0.5$ indicate that the decision for $\boldsymbol{x}$ using this training process is statistically indistinguishable from a coin flip [14]. A standard U-statistic yields an unbiased estimator: $\mathbb{E}[\widehat{\mathrm{SC}}(\boldsymbol{x})] = \mathrm{SC}(\boldsymbol{x})$. Note that SC is defined for any $\boldsymbol{x}$. Cooper et al. [14] estimate it for samples in a held-out test.

**Flip rate on calibrated MIA decision rules.** In our setting, we fix the dataset $\mathbb{D} \sim \mathcal{D}$ and vary only the training seed, which affects batch order during training. We adapt SC from $\mu_{\mathcal{A},\mathcal{D}}$ to the MIA decisions under $\mu_{\mathcal{A},\mathbb{D}}$ calibrated at a fixed FPR.

Let $r \sim \mu \equiv \mu_{\mathcal{A},\mathbb{D}}$ denote a target model drawn from the seed-induced distribution with the (fixed) training dataset $\mathbb{D}$. Let $\Lambda_r(\boldsymbol{x}) \in \mathbb{R}$ be the attack score (e.g., LiRA posterior, Equation 2) for sample $\boldsymbol{x}$. For a desired false-positive rate $\eta \in [0, 1]$, define the per-seed calibrated threshold $\tau_r(\eta)$ (e.g., the $(1-\eta)$-quantile of $\Lambda_r$ on non-members for that seed), and the calibrated membership decision

$$b_r^{(\eta)}(\boldsymbol{x}) \;=\; \mathbf{1}\{\Lambda_r(\boldsymbol{x}) \geq \tau_r(\eta)\} \tag{7}$$

(as in Section 2 and Appendix A). Unlike Cooper et al. [14], we focus on *dis*agreement between cross-seed MIA decisions for $\boldsymbol{x}$ rather than agreement. The (population) **flip rate** at $\boldsymbol{x}$ under $\mu$ and FPR $\eta$ is

$$\mathrm{flip}_\eta(\boldsymbol{x}) \;:=\; \Pr_{r,r' \overset{\mathrm{i.i.d.}}{\sim} \mu} \big[b_r^{(\eta)}(\boldsymbol{x}) \neq b_{r'}^{(\eta)}(\boldsymbol{x})\big] \;=\; 1 - \mathrm{SC}_\eta(\boldsymbol{x}), \tag{8}$$

which lies in $[0, 0.5]$ at the population level, with 0 indicating that decision for $\boldsymbol{x}$ does not flip/ is stable across target replicas and $0.5$ indicating that the decision for $\boldsymbol{x}$ behaves like a coin flip. (The operator point $\eta$ is left implicit in the use of SC in Cooper et al. [14], as the authors always set $\tau = 0.5$ in practice.)

Note that we deliberately calibrate per seed, as this mirrors how MIAs are actually run in practice: a single target is calibrated at its chosen FPR. Here, we vary the target (via seed) to expose instability across plausible targets $r \sim \mu$ using the same training recipe.

**Unbiased estimator (order-2 U-statistic) and closed form.** In practice, we estimate the population flip rate (Equation 8) for a concrete number of target replicas $B$ trained with different random seeds that control batch order. Given $B \geq 2$ i.i.d. target replicas $r_1, \ldots, r_B \sim \mu$ with calibrated rules $b_{r_i}^{(\eta)}$, the canonical unbiased estimator of $\mathrm{flip}_\eta(\boldsymbol{x})$ is

$$\widehat{\mathrm{flip}}_{\eta,B}(\boldsymbol{x}) \;=\; \binom{B}{2}^{-1} \sum_{1 \leq i < j \leq B} \mathbf{1}\{b_{r_i}^{(\eta)}(\boldsymbol{x}) \neq b_{r_j}^{(\eta)}(\boldsymbol{x})\}, \qquad \mathbb{E}\big[\widehat{\mathrm{flip}}_{\eta,B}(\boldsymbol{x})\big] = \mathrm{flip}_\eta(\boldsymbol{x}). \tag{9}$$

Let $B_1(\boldsymbol{x}) = \sum_{i=1}^B b_{r_i}^{(\eta)}(\boldsymbol{x})$ and $B_0(\boldsymbol{x}) = B - B_1(\boldsymbol{x})$ be the numbers of "member" and "non-member" binary decisions among the $B$ replicas for $\boldsymbol{x}$. Then, Equation 9 has the closed form

$$\widehat{\mathrm{flip}}_{\eta,B}(\boldsymbol{x}) \;=\; \frac{2\, B_0(\boldsymbol{x})\, B_1(\boldsymbol{x})}{B\,(B-1)}. \tag{10}$$

Maximizing $B_0(\boldsymbol{x})B_1(\boldsymbol{x})$ under $B_0(\boldsymbol{x})+B_1(\boldsymbol{x})=B$ yields the finite-$B$ upper bound, since

$$\widehat{\text{flip}}_{\eta,B}(\boldsymbol{x}) \;\leq\; \frac{2\,\lfloor B/2\rfloor\,\lceil B/2\rceil}{B\,(B-1)} \;=\; \begin{cases} \dfrac{B}{2(B-1)} = \dfrac{1}{2} + \dfrac{1}{2(B-1)}, & B \text{ even},\\[2ex] \dfrac{B+1}{2B} = \dfrac{1}{2} + \dfrac{1}{2B}, & B \text{ odd}, \end{cases}$$

which exceeds $0.5$ and converges to $0.5$ as $B \to \infty$ (e.g., $B{=}125 \Rightarrow 0.504$).

To see why, note that $B_0(\boldsymbol{x})B_1(\boldsymbol{x})$ is maximized by the most balanced vote split (i.e., $B_0(\boldsymbol{x})B_1(\boldsymbol{x}) \leq \lfloor B/2\rfloor\lceil B/2\rceil$). If $B$ is even, i.e., $B{=}2k$, the maximum occurs at $B_0(\boldsymbol{x}) = B_1(\boldsymbol{x}) = \lfloor B/2\rfloor = k$, so

$$B_0(\boldsymbol{x})B_1(\boldsymbol{x}) = k^2 = \frac{B^2}{4} \implies \text{flip}_{\max} = \frac{2\cdot(B^2/4)}{B(B-1)} = \frac{B}{2(B-1)} = \frac{1}{2} + \frac{1}{2(B-1)}.$$

If $B$ is odd, $B = 2k+1$, the maximum occurs at $B_0(\boldsymbol{x})B_1(\boldsymbol{x}) = \lfloor B/2\rfloor\lceil B/2\rceil = k(k+1)$, so

$$B_0(\boldsymbol{x})B_1(\boldsymbol{x}) = k(k{+}1) = \frac{B^2-1}{4} \implies \text{flip}_{\max} = \frac{2\cdot((B^2-1)/4)}{B(B-1)} = \frac{B^2-1}{2B(B-1)} = \frac{1}{2} + \frac{1}{2B}.$$

Of course, this means that at low $B$, flip rate can have values that are quite far away from $0.5$. For example, when $B{=}2$, the $\text{flip}_{\max}{=}1$. Nevertheless, this is the right choice of metric, as it is unbiased. In our experiments, we ensure that the flip rate is easily interpretable by plotting results where the minimum $B{=}125$, such that $\text{flip}_{\max}{\approx}0.504$. We discuss this further in Appendix E.2.4.

**Why the U-statistic is the right estimator (unbiasedness).**   Fix a sample $\boldsymbol{x}$ and an FPR $\eta$. Write $b_r^{(\eta)}(\boldsymbol{x}) \in \{0,1\}$ for the calibrated decision of target $r \sim \mu_{\mathcal{A},\mathbb{D}}$. Let $b$ denote a generic draw of $b_r^{(\eta)}(\boldsymbol{x})$, and set

$$\theta := \Pr\big[b = 1\big] \in [0,1].$$

Draw $B \geq 2$ i.i.d. replicas $b_1,\ldots,b_B \overset{\text{i.i.d.}}{\sim} \text{Bernoulli}(\theta)$. The population flip rate at $(\boldsymbol{x},\eta)$ (the pairwise disagreement probability for two independent draws) is

$$\begin{aligned} \text{flip}_\eta(\boldsymbol{x}) &= \Pr[b \neq b'] = \Pr[b{=}1, b'{=}0] + \Pr[b{=}0, b'{=}1]\\ &= \theta(1{-}\theta) + (1{-}\theta)\theta = 2\theta(1{-}\theta). \end{aligned} \tag{11}$$

Because $(\theta - \frac{1}{2})^2 \geq 0 \iff \theta(1-\theta) \leq \frac{1}{4}$, we have $\text{flip}_\eta(\boldsymbol{x}) = 2\theta(1-\theta) \leq \frac{1}{2}$, i.e., the population flip rate never exceeds $0.5$.

For a concrete $B$, the empirical estimator (order-2 U-statistic) averages the pairwise indicator over all unordered pairs:

$$\widehat{\text{flip}}_{\eta,B}(\boldsymbol{x}) = \binom{B}{2}^{-1} \sum_{1\leq i<j\leq B} \mathbf{1}\{b_i \neq b_j\},$$

as in Equation 9. By linearity of expectation and independence,

$$\mathbb{E}\big[\widehat{\text{flip}}_{\eta,B}(\boldsymbol{x})\big] = \binom{B}{2}^{-1} \sum_{1\leq i<j\leq B} \mathbb{E}\big[\mathbf{1}\{b_i \neq b_j\}\big].$$

For any fixed pair $(i,j)$ with $i \neq j$,

$$\mathbb{E}\big[\mathbf{1}\{b_i \neq b_j\}\big] = \Pr[b_i \neq b_j] = \Pr[b_i{=}1, b_j{=}0] + \Pr[b_i{=}0, b_j{=}1].$$

Because $b_i, b_j$ are independent Bernoulli$(\theta)$,

$$\Pr[b_i{=}1, b_j{=}0] = \Pr[b_i{=}1]\Pr[b_j{=}0] = \theta(1-\theta), \quad \Pr[b_i{=}0, b_j{=}1] = (1-\theta)\theta,$$

so $\Pr[b_i \neq b_j] = 2\theta(1-\theta)$. Therefore every term in the sum equals $2\theta(1-\theta)$, so

$$\mathbb{E}\big[\widehat{\text{flip}}_{\eta,B}(\boldsymbol{x})\big] = \binom{B}{2}^{-1} \sum_{1\leq i<j\leq B} 2\theta(1-\theta) = 2\theta(1-\theta) = \text{flip}_\eta(\boldsymbol{x}),$$

so $\widehat{\text{flip}}_{\eta,B}$ is exactly unbiased for all $B \geq 2$.

**Showing unbiasedness via the vote fraction.** For our discussion below and in Appendix E.2.4, it is useful to see the same result via another argument. As above in our discussion of flip rate (Equation 1), Let

$$B_1(\boldsymbol{x}) \;=\; \sum_{i=1}^{B} b_i^{(\eta)}(\boldsymbol{x}), \qquad B_0(\boldsymbol{x}) \;=\; B - B_1(\boldsymbol{x}).$$

By construction, $B_1(\boldsymbol{x})$ is the sum of $B$ i.i.d. Bernoulli($\theta$) draws, so

$$B_1(\boldsymbol{x}) \sim \mathrm{Binomial}(B, \theta).$$

Define the **vote fraction** $v(\boldsymbol{x}) := B_1(\boldsymbol{x})/B$. Therefore, we can write

$$B_1(\boldsymbol{x}) = B v(\boldsymbol{x})$$
$$B_0(\boldsymbol{x}) = B(1 - v(\boldsymbol{x})).$$

The number of disagreeing unordered pairs is $B_1(\boldsymbol{x}) B_0(\boldsymbol{x})$ (choose one "member" vote and one "non-member" vote), so

$$\widehat{\mathrm{flip}}_{\eta,B}(\boldsymbol{x}) \;=\; \frac{B_1(\boldsymbol{x}) B_0(\boldsymbol{x})}{\binom{B}{2}} = B_1(\boldsymbol{x}) B_0(\boldsymbol{x}) \frac{(B-2)!2!}{B!} = \frac{2}{B(B-1)} B^2 v(\boldsymbol{x})(1 - v(\boldsymbol{x}))$$

$$= \frac{2B}{B-1} v(\boldsymbol{x})\big(1 - v(\boldsymbol{x})\big). \tag{12}$$

Since $B_1(\boldsymbol{x}) \sim \mathrm{Binomial}(B, \theta)$,

$$\mathbb{E}\big[v(\boldsymbol{x})\big] = \frac{\mathbb{E}[B_1(\boldsymbol{x})]}{B} = \frac{B\theta}{B} = \theta, \quad \text{and} \tag{13}$$

$$\mathrm{Var}\big[v(\boldsymbol{x})\big] = \frac{\theta(1-\theta)}{B}, \tag{14}$$

because

$$\mathrm{Var}\big[v(\boldsymbol{x})\big] = \mathrm{Var}\bigg[\frac{B_1(\boldsymbol{x})}{B}\bigg] = \frac{\mathrm{Var}[B_1(\boldsymbol{x})]}{B^2},$$

by the scaling law for variance:

$$\mathrm{Var}[aX] = \mathbb{E}\big[(aX - \mathbb{E}[aX])^2\big] = \mathbb{E}\big[(a(X - \mathbb{E}[X]))^2\big] = a^2 \, \mathbb{E}\big[(X - \mathbb{E}[X])^2\big] = a^2 \mathrm{Var}[X].$$

Next,

$$\mathrm{Var}[B_1(\boldsymbol{x})] = \mathrm{Var}\bigg[\sum_{r=1}^{B} b_r^{(\eta)}(\boldsymbol{x})\bigg]$$

$$= \sum_{r=1}^{B} \mathrm{Var}\big[b_r^{(\eta)}(\boldsymbol{x})\big] + 2\sum_{r<j} \mathrm{Cov}\big[b_r^{(\eta)}(\boldsymbol{x}), b_j^{(\eta)}(\boldsymbol{x})\big]$$

$$= \sum_{r=1}^{B} \mathrm{Var}\big[b_r^{(\eta)}(\boldsymbol{x})\big] \qquad \text{(independence: } \mathrm{Cov}(\cdot, \cdot) = 0 \text{ for } r \neq j\text{)}.$$

Because $b_r^{(\eta)}(\boldsymbol{x})$ is a Bernoulli variable with success probability $\theta$, $\mathrm{Var}[b_r^{(\eta)}(\boldsymbol{x})] = \theta(1-\theta)$. Therefore

$$\mathrm{Var}[B_1(\boldsymbol{x})] = \sum_{r=1}^{B} \theta(1-\theta) = B\theta(1-\theta),$$

and so

$$\mathrm{Var}\big[v(\boldsymbol{x})\big] = \frac{\mathrm{Var}[B_1(\boldsymbol{x})]}{B^2} = \frac{B\theta(1-\theta)}{B^2} = \frac{\theta(1-\theta)}{B},$$

as claimed in Equation 14. Finally, combining Equations 13 and 14 with the definition of variance,

$$\mathbb{E}\big[v(\boldsymbol{x})^2\big] \;=\; \mathrm{Var}[v(\boldsymbol{x})] + \mathbb{E}[v(\boldsymbol{x})]^2 \;=\; \frac{\theta(1-\theta)}{B} + \theta^2. \tag{15}$$

Therefore, by Equations 13 and 15,

$$\mathbb{E}\big[v(\boldsymbol{x})(1 - v(\boldsymbol{x}))\big] = \mathbb{E}[v(\boldsymbol{x})] - \mathbb{E}[v(\boldsymbol{x})^2] = \theta - \left(\frac{\theta(1 - \theta)}{B} + \theta^2\right)$$

$$= (\theta - \theta^2) - \frac{\theta(1 - \theta)}{B}$$

$$= \theta(1 - \theta) - \frac{1}{B} \cdot \theta(1 - \theta)$$

$$= \theta(1 - \theta)\left(1 - \frac{1}{B}\right).$$

Plugging into Equation 12 gives

$$\mathbb{E}\big[\widehat{\mathrm{flip}}_{\eta,B}(\boldsymbol{x})\big] = \frac{2B}{B - 1}\,\mathbb{E}\big[v(\boldsymbol{x})(1 - v(\boldsymbol{x}))\big] = \frac{2B}{B - 1}\,\theta(1 - \theta)\left(1 - \frac{1}{B}\right)$$

$$= \frac{2B\theta(1 - \theta)}{B - 1} - \frac{2\theta(1 - \theta)}{B - 1}$$

$$= \frac{2\theta(1 - \theta)(B - 1)}{B - 1}$$

$$= 2\theta(1 - \theta) = \mathrm{flip}_{\eta}(\boldsymbol{x}),$$

by Equation 11. Therefore, the U-statistic is unbiased for all $B \geq 2$.

**Why a quadratic surrogate for "lack of margin" is biased.**   As we discuss in Appendix E.2.4, interpreting empirical estimates of the flip rate can be a bit counter-intuitive. Empirical estimates that are very close to $0.5$ may actually reflect a vote split that seems a bit far from $\lfloor B/2 \rfloor / \lceil B/2 \rceil$. In other words, concrete splits for a given $B$ might "feel" somewhat far from a $50/50$ split even if $\widehat{\mathrm{flip}}_{\eta,B} \approx 0.5$. As a result, it might seem natural to derive a metric that captures coin-flip behavior by showing how far the vote fraction (Equation 14) is from a completely split vote, rather than estimating the flip rate.

That is, consider that the raw margin from a completely split vote is $v(\boldsymbol{x}) - \frac{1}{2}$. (Note that, if $v(\boldsymbol{x}) = 0.5$, then the raw margin is $0$; if $v(\boldsymbol{x}) = 1$, then the raw margin is $0.5$; if $v(\boldsymbol{x}) = 0$, then the raw margin is $-0.5$; and similarly, for any intermediate vote fraction.) Scaling so the range becomes $[0, 1]$ and taking absolute value so that there are no negative values gives

$$m(\boldsymbol{x}) := \big|2v(\boldsymbol{x}) - 1\big| \in [0, 1].$$

Therefore, $m(\boldsymbol{x}) = 0$ at a perfect split and $m(\boldsymbol{x}) = 1$ at unanimity. But of course, $m(\boldsymbol{x})$ is neither smooth nor concave. We show two convenient identities (by completing the square) that relate the margin and the quadratic in $v(\boldsymbol{x})$, so that we can have a smooth, concave alternative:

$$v(\boldsymbol{x})\big(1 - v(\boldsymbol{x})\big) = \tfrac{1}{4} - \big(v(\boldsymbol{x}) - \tfrac{1}{2}\big)^2 = \tfrac{1}{4} - \tfrac{1}{4}\big(2v(\boldsymbol{x}) - 1\big)^2 = \tfrac{1}{4}\big(1 - m(\boldsymbol{x})^2\big), \qquad (16)$$

$$2\,v(\boldsymbol{x})\big(1 - v(\boldsymbol{x})\big) = \tfrac{1}{2} - 2\big(v(\boldsymbol{x}) - \tfrac{1}{2}\big)^2 = \tfrac{1}{2} - \tfrac{1}{2}\,m(\boldsymbol{x})^2.$$

So $2v(\boldsymbol{x})(1 - v(\boldsymbol{x}))$ is a smooth, concave, symmetric surrogate for "lack of margin" (maximal at $v(\boldsymbol{x}) = \frac{1}{2}$, decreasing as the margin grows).

While this alternative seems to behave "nicely" in practice (i.e., is at most $\frac{1}{2}$, unlike $\widehat{\mathrm{flip}}_{\eta,B}$), it is biased (downward) for finite $B$. That is,

$$\mathbb{E}\big[2v(\boldsymbol{x})\big(1 - v(\boldsymbol{x})\big)\big] = 2\big(\mathbb{E}[v(\boldsymbol{x})] - \mathbb{E}[v(\boldsymbol{x})^2]\big)$$

$$= 2\Big(\mathbb{E}[v(\boldsymbol{x})] - \big(\mathrm{Var}[v(\boldsymbol{x})] + \mathbb{E}[v(\boldsymbol{x})]^2\big)\Big) \qquad \text{(variance identity)}$$

$$= 2\Big(\theta - \big(\mathrm{Var}[v(\boldsymbol{x})] + \theta^2\big)\Big) \qquad \text{(by Equation 13)}$$

$$= 2\Big(\theta - \big(\tfrac{\theta(1-\theta)}{B} + \theta^2\big)\Big) \qquad \text{(by Equation 14)}$$

$$= 2\Big(\theta - \theta^2 - \tfrac{\theta(1-\theta)}{B}\Big)$$

$$= 2\,\theta(1 - \theta)\Big(1 - \tfrac{1}{B}\Big).$$

The population flip rate is $2\theta(1-\theta)$ (Equation 11), so

$$\mathbb{E}\left[2v(\boldsymbol{x})\big(1-v(\boldsymbol{x})\big)\right] = \mathrm{flip}_\eta(\boldsymbol{x}) \cdot \left(1 - \tfrac{1}{B}\right),$$

i.e., the quadratic surrogate metric for showing a "lack of margin" (i.e., coin-flip-like behavior of MIA decisions for $\boldsymbol{x}$) is downward biased by $\frac{1}{B}$ (i.e., is $\frac{1}{B}$ below the population flip rate) for any finite $B \geq 2$, and becomes unbiased only as $B \to \infty$. By contrast, the U-statistic $\widehat{\mathrm{flip}}_{\eta,B}(\boldsymbol{x})$ (Equation 1) is exactly unbiased at every $B \geq 2$. This is why we report the U-statistic, i.e., the pairwise decision disagreement probability (which we informally call the flip rate).

### E.2.2 Measuring flip rate for MIA

In Cooper et al. [14], the authors train $B$ models using bootstrap replicates drawn from a dataset $\mathbb{D}$. They split $\mathbb{D}$ into train and test sets, train $B$ models on bootstrap subsamples of the train set, and, for each held-out test sample, compute an unbiased estimate of self-consistency from the $B$ binary decisions.

Here, we measure flip rate in a setup that mirrors strong MIA. We fix a dataset $\mathbb{D}$ of size $2N{=}2^{20}$ (so $N{=}2^{19}$) and train each target model on the *same* $N$-sized subset—i.e., the set of members (size $N$) and non-members (size $N$) is identical across targets. When training targets, we change *only* the random seed that determines batch order during training. Changing the batch order induces randomness in the training process. Together with unavoidable hardware non-determinism, this yields the variability we observe across target models [11].

For LiRA, we fix a reference set of 128 independently trained models on different $N$-sized subsamples, and we use these same references for *every* target to compute per-sample IN and OUT reference distributions, $p_{\mathrm{IN}}(\cdot \mid \boldsymbol{x})$ and $p_{\mathrm{OUT}}(\cdot \mid \boldsymbol{x})$. At a chosen FPR $\eta \in [0,1]$, each target model $r$ calibrates its *own* threshold $\tau_r(\eta)$ on that target's non-member scores (i.e., we perform per-seed calibration), and then applies the calibrated decision rule in Equation 7. We then compute the flip rate for a sample $\boldsymbol{x}$ over the ensemble $\{r_i\}_{i=1}^B$ via Equation 9, thereby isolating the effect of target-training randomness while holding references fixed. We run such experiments on two model sizes: 140M and 302M (Appendix E.2.5).

For reference, we highlight some key points about calibration that will come up repeatedly in the rest of this appendix.

---

**Calibration asymmetry and its consequences**

**What we calibrate.** For each target $r$ and fixed FPR $\eta$, the decision threshold is $\tau_r(\eta) = \inf\{\, t : \widehat{F}_{\mathrm{OUT},r}(t^-) \geq 1 - \eta \,\}$, i.e., the empirical $(1-\eta)$-quantile of that seed's *non-member* scores. This guarantees the *non-member* tail is controlled at level $\eta$ for that seed (with the usual tie convention; see Appendix A).

**Why asymmetry arises.** Because $\tau_r(\eta)$ is re-estimated on non-members for each seed, it "tracks" seed-to-seed shifts in *non-member* score distributions by construction. Members, however, are not used for calibration, so many member scores lie closer to (and straddle) the moving boundary across seeds (Figures 25, 26, & 27).

**Empirical effect.** In regions where IN/OUT scores overlap (Figures 22 & 23), small seed-induced shifts in either the score or the boundary can flip member decisions; consequently, members exhibit substantially higher flip rate than non-members at the same $\eta$, and the gap widens at larger $\eta$ (up to a point) and with increased model size (Figures 28 & 29).

**Implication.** This calibration asymmetry explains why aggregate metrics (e.g., mean TPR at fixed FPR, see Tables 1 & 2) can look stable (Figures 24 & 30), while many *member* decisions are individually unstable (Tables 3, 4, 5, & 6). It also motivates our hypothesis-test cutoff $t_\alpha(B)$ for flagging statistically coin-flip-like per-sample decisions (Appendix E.2.4).

---

**Interpreting flip rate for MIA.** Each target model is a plausible outcome of this training process. Any of them would be a reasonable choice for running LiRA, as they are i.i.d. draws from the same seed-induced distribution. Measuring flip rate across targets therefore quantifies how resilient LiRA's per-sample decision is to randomness in target training.

If a sample's binary membership decisions are *stable* (low flip), LiRA's decision for that sample is *robust* to target-training randomness and more likely to reflect persistent signal, rather than

seed-specific idiosyncrasies. Conversely, if binary decisions are *unstable* (flip near its population maximum $0.5$), the per-sample decision is effectively *arbitrary* with respect to seed choice—even when aggregate performance metrics (e.g., TPR at fixed FPR, or AUC>0.5) look stable and reasonably high-performance. In this case, per-sample membership decisions are so influenced by randomness in the training process that we cannot draw a reliable conclusion about membership. Put differently, measuring per-sample instability lets us peer beneath high-level, average metrics—e.g., for a fixed FPR, mean TPR over all members across plausible targets $r \sim \mu$—to assess what strong MIAs can (and cannot) say reliably about individual samples.

### E.2.3   Connections to prior work on model and predictive multiplicity

This analysis connects to broader literature in statistics and machine learning outside membership inference. Notably, Leo Breiman's seminal work on the **Rashomon effect** emphasized that, for a given dataset, there often exists a *multiplicity* of distinct decision rules with essentially the same overall accuracy [3]. The Rashomon set—the set of models within a small tolerance of the optimal risk—can be surprisingly large [20, 51]. More recent work on predictive multiplicity also shows that training processes can produce models with effectively indistinguishable overall test accuracy that nonetheless disagree widely at the per-sample level [14, 36, 59].

To the best of our knowledge, this connection has not been made in the MIA setting. Our setup differs in that we fix $\mathbb{D}$ and vary only algorithmic randomness (via seed controlling batch order for target replicas); we then observe targets with similar overall accuracy but substantial per-sample churn, quantified by flip rate (Appendix E.2.5). (We make no claims about the optimality of the resulting MIA rules.) The key result of these experiments is that average attack performance can remain stable, while individual membership decisions vary across seeds—a phenomenon that bears directly on the reliability and validity of membership claims about specific samples (as the problem is set up in the membership inference security game).

### E.2.4   Reasoning about the minimum empirical flip rate that reflects coin-flip MIA decisions

As noted in Appendix E.2.1, the population flip rate $\mathrm{flip}_\eta(\boldsymbol{x}) \in [0, 0.5]$ (Equation 8): $0$ reflects MIA decisions that are completely stable for $\boldsymbol{x}$ (i.e., do not flip) and $0.5$ reflects coin-flip decisions for $\boldsymbol{x}$. In practice, we estimate the population flip rate with the U-statistic for flip rate using a concrete number of target replicas $B$, namely $\widehat{\mathrm{flip}}_{\eta,B}(\boldsymbol{x})$ (Equation 9). This empirical estimate also has a minimum of $0$, reflecting completely stable binary decisions, but its maximum (reflecting maximal disagreement) slightly exceeds $0.5$ and converges to $0.5$ as $B \to \infty$. This raises an important question: for concrete $B$ in practice, which measurements of $\widehat{\mathrm{flip}}_{\eta,B}(\boldsymbol{x})$ reflect that the MIA decisions for $\boldsymbol{x}$ behave like a coin flip? That is, we need to determine a reasonable cutoff for $\widehat{\mathrm{flip}}_{\eta,B}(\boldsymbol{x})$, indicating that the decisions for $\boldsymbol{x}$ are statistically indistinguishable from a coin flip.

A principled way to determine this cutoff is to set up a hypothesis test at level $\alpha$: we call the MIA decision for a sample $\boldsymbol{x}$ "indistinguishable from a coin flip at level $\alpha$" if a two-sided exact binomial test fails to reject. We do this for our experiments in Section 5 and Appendix E.2.5. For the experiment with the 140M model ($B{=}125$), we call the MIA decision for $\boldsymbol{x}$ indistinguishable from a coin flip at $\alpha{=}0.05$ if the MIA decisions for $\boldsymbol{x}$ exhibit $\widehat{\mathrm{flip}}_{\eta,B}(\boldsymbol{x}) \gtrsim 0.490$; for the 302M model ($B{=}127$), we call the MIA decision for $\boldsymbol{x}$ indistinguishable from a coin flip at $\alpha{=}0.05$ if the MIA decisions for $\boldsymbol{x}$ exhibit $\widehat{\mathrm{flip}}_{\eta,B}(\boldsymbol{x}) \gtrsim 0.487$ (Figure 5a). In the end, all this requires is finding the minimal number of member votes $k$ at which the CDF $F(k)$ of the binomial $\mathrm{Binomial}(B, 0.5) \geq \frac{\alpha}{2}$, i.e.,

$$k_{\mathrm{L}} = \min\{k : F(k) \geq \alpha/2\}, \tag{17}$$

and computing the coin-flip cutoff as $\geq \widehat{\mathrm{flip}}_{\eta,B}$ with $B_1(\boldsymbol{x}) = k_{\mathrm{L}}$ and $B_0(\boldsymbol{x}) = B - k_{\mathrm{L}}$.

In this appendix, for the reader interested in a refresher, we walk through how we set up this exact test. We describe the hypothesis test at level $\alpha$, how this results in an acceptance region (in terms of the number of member votes), and how we convert that region into a minimum empirical $\widehat{\mathrm{flip}}_{\eta,B}(\boldsymbol{x})$ that we can defensibly interpret as behaving like a coin flip. (This depends on the the vote fraction $v(\boldsymbol{x})$ discussion from above.)

**Setting up a hypothesis test.**   We call the MIA decision for a sample $\boldsymbol{x}$ indistinguishable from a coin flip if the probability of predicting "member" equals the probability of predicting "non-member".

Let $r_1, \ldots, r_B \overset{\text{i.i.d.}}{\sim} \mu$ denote $B$ target replicas (varying only by seed) from the training pipeline. As throughout, let

$$B_1(\boldsymbol{x}) \;=\; \sum_{i=1}^{B} b_{r_i}^{(\eta)}(\boldsymbol{x}),$$

and note that, for fixed $(\boldsymbol{x}, \eta)$, the indicators $b_{r_i}^{(\eta)}(\boldsymbol{x})$ are i.i.d. Bernoulli with success probability

$$\theta \;\coloneqq\; \Pr_{r \sim \mu}\left[ b_r^{(\eta)}(\boldsymbol{x}) = 1 \right].$$

Therefore, $B_1(\boldsymbol{x}) \sim \text{Binomial}(B, \theta)$. Coin-flip behavior corresponds to $\theta = 0.5$.

We set up the null hypothesis

$$H_0 : \;\; \theta = 0.5 \quad \text{(two-sided exact binomial test at level } \alpha\text{)}. \tag{18}$$

If we fail to reject $H_0$, then we do not have sufficient evidence to say that the MIA decision for $\boldsymbol{x}$ is *not* a coin flip, and so we deem the decision indistinguishable from a coin flip. The significance level $\alpha$ means that, if $H_0$ is true (i.e., the decision is indistinguishable from a coin flip), the probability that we incorrectly reject $H_0$ (i.e., say that the decision is not a coin flip) is at most $\alpha$. Smaller $\alpha$ imposes a stricter standard for rejecting $H_0$ (stronger evidence is required). We will later show that, for $B$ replicas, "fail to reject" is equivalent to

$$\widehat{\text{flip}}_{\eta, B}(\boldsymbol{x}) \;\geq\; t_\alpha(B),$$

with $t_\alpha(B)$ computed from the binomial acceptance region under $H_0$ (Equation 23).

**Deriving the two-sided exact $p$-value at level $\alpha$.** For $B$ replicas and operating point $\eta$, each replica $r$ outputs a binary membership decision $b_r^{(\eta)}(\boldsymbol{x}) \in \{0, 1\}$. Going forward, we denote the member-vote count

$$K \;=\; B_1(\boldsymbol{x}) \;=\; \sum_{i=1}^{B} b_{r_i}^{(\eta)}(\boldsymbol{x}).$$

Under the null hypothesis $H_0$ in Equation 18, each target replica's MIA decision behaves like a fair coin, so

$$K \;\sim\; \text{Binomial}(B, 0.5).$$

Intuitively, the further $K$ is from the center $B/2$ (i.e., a split vote, indicating coin-flip behavior), the stronger the evidence against $H_0$.

More formally, let the binomial PMF and CDF under $H_0$ be, respectively,

$$\Pr(K = i) \;=\; \binom{B}{i} 2^{-B}, \qquad F(k) \;=\; \Pr(K \leq k) \;=\; \sum_{i=0}^{k} \binom{B}{i} 2^{-B}. \tag{19}$$

Because $\binom{B}{i} = \binom{B}{B-i}$,

$$\Pr(K = i) \;=\; \Pr\left(K = B - i\right) \quad \text{for all } i,$$

and so the distribution is symmetric about $B/2$.

We can reason about the tails of this distribution in terms of a concrete vote $k$ and the CDF, i.e.,

$$F(k) = \Pr(K \leq k) = \sum_{i=0}^{k} \Pr(K = i) = \sum_{i=0}^{k} \Pr\left(K = B - i\right) = \sum_{j=B-k}^{B} \Pr(K = j)$$

$$= \Pr\left(K \geq B - k\right). \tag{20}$$

Thus the left tail at $k$ equals the right tail at $B - k$. This follows from a change of variable, setting $j = B - i$ (when $i = 0$, $j = B$ and when $i = k$, $j = B - k$, so the index runs in reverse and $\sum_{i=0}^{k} \Pr(K = B - i) = \sum_{j=B-k}^{B} \Pr(K = j)$).

And so,

$$\Pr(K \geq k) = 1 - \Pr(K \leq k - 1) = 1 - F(k - 1)$$
$$= 1 - \Pr\big(K \geq B - (k - 1)\big) \quad \text{(by Equation 20 with } k \mapsto k - 1)$$
$$= \Pr\big(K \leq B - k\big). \tag{21}$$

Intuitively, a two-sided $p$-value measures how surprising the actual observed count $K=k$ is under the null hypothesis (Equation 18): it sums the probabilities of outcomes *at least as far from the center* $B/2$ *as $k$*, in both tails. Because the binomial for $0.5$ is symmetric and unimodal about $B/2$, "equally or more extreme" corresponds to the union of the left tail up to $k$ and the symmetric right tail from $B-k$ upward (or the mirror statement when $k$ is on the right). So, to derive a $p$-value at $k$, there are two cases to consider.

*Case 1*: $k \leq \lfloor B/2 \rfloor$

Here, the left tail is $K \leq k$ and the right tail is $K \geq B - k$. Therefore,

$$p\text{-value}(k) = \Pr(K \leq k) + \Pr(K \geq B - k)$$
$$= 2 \cdot \Pr(K \leq k) \quad \text{(by tail symmetry, Equation 20)}$$
$$= 2 \cdot F(k) \quad \text{(by definition of the CDF, Equation 19)}.$$

*Case 2*: $k \geq \lceil B/2 \rceil$ Here, the right tail is $K \geq k$ and the left tail is $K \leq B - k$. Therefore,

$$p\text{-value}(k) = \Pr(K \geq k) + \Pr(K \leq B - k)$$
$$= 2 \cdot \Pr(K \geq k) \quad \text{(by tail symmetry, Equation 21)}$$
$$= 2 \cdot \big(1 - F(k - 1)\big).$$

From this, we derive the standard form of the two-sided exact $p$-value. That is, because the binomial is symmetric and unimodal about $B/2$, this can be written as

$$p\text{-value}(k) = 2\min\Big\{ \Pr(K \leq k),\ \Pr(K \geq k) \Big\},$$

which means we double the smaller tail in order to capture both-sided extremeness. Alternatively, using $\Pr(K \geq k) = 1 - \Pr(K \leq k - 1) = 1 - F(k - 1)$ (Equation 21), this becomes

$$p\text{-value}(k) = \begin{cases} 2\,F(k), & k \leq \lfloor B/2 \rfloor, \\ 2\,\big(1 - F(k - 1)\big), & k \geq \lceil B/2 \rceil. \end{cases}$$

Finally, to handle discreteness at the exact center (even $B$ and $k = B/2$, which counts the mass at $k$ twice), we cap $p$-values at $1$:

$$p\text{-value}(k) = \min\Big\{ 1,\ 2\min\big(F(k),\ 1 - F(k - 1)\big)\Big\}. \tag{22}$$

These equal-tail formulas handle discreteness conservatively: the acceptance region is defined so that $p$-value$(k) \geq \alpha$ inside the region and $p$-value$(k) < \alpha$ outside, with $\frac{\alpha}{2}$ in each tail. Because the binomial is discrete, the equal-tail construction is slightly conservative. We follow the convention "reject if $p < \alpha$" and fail to reject if $p \geq \alpha$, so boundary points with $p$-value $= \alpha$ remain inside the acceptance region.

Using Equation 22, the acceptance region for member votes $k$ at level $\alpha$ is constructed by finding

$$k_{\mathrm{L}} = \min\{k \in \{0, \ldots, \lfloor B/2 \rfloor\} :\ F(k) \geq \alpha/2\},$$

and then setting

$$\mathcal{A}_\alpha = \{k_{\mathrm{L}},\, k_{\mathrm{L}}+1,\, \ldots,\, B - k_{\mathrm{L}}\},$$

so that $K \in \mathcal{A}_\alpha \iff p\text{-value}(K) \geq \alpha$ (fail to reject). By symmetry, the upper endpoint is $B - k_{\mathrm{L}}$ and so the acceptance region is a symmetric band around $B/2$ (Equation 20).

Equivalently, the critical (rejection) region is

$$\{\, K \leq k_{\mathrm{L}} - 1 \,\} \cup \{\, K \geq B - k_{\mathrm{L}} + 1 \,\},$$

so $K \in \mathcal{A}_\alpha \iff p\text{-value}(K) \geq \alpha$ (fail to reject), and $K \notin \mathcal{A}_\alpha \iff p\text{-value}(K) < \alpha$ (reject).

And so, for a fixed $B$ and given $k$, we can check if $F(k) \geq \alpha/2$ simply by computing $F(k) = \sum_{i=0}^{k} \binom{B}{i} 2^{-B} \geq \alpha/2$, as in Equation 19. For $\alpha=0.05$ and $B=127$, $k_{\mathrm{L}}=52$ (and $B - k_{\mathrm{L}}=75$) with $F(52) \approx 0.02524$; for $\alpha=0.05$ and $B = 125$, it is also the case that $k_{\mathrm{L}}=52$ with $F(52) \approx 0.03661$ (but $B - k_{\mathrm{L}}=73$).

**From the acceptance band to a concrete flip cutoff.** For fixed $B$ and FPR $\eta$, the empirical flip at $(\boldsymbol{x}, \eta)$ as a function of the member-vote count $K$ is

$$\phi_B(K) := \frac{2\,K\,(B-K)}{B(B-1)},$$

where $\widehat{\mathrm{flip}}_{\eta,B}(\boldsymbol{x}) = \phi_B(K)$ (to preserve notation/ defining $\widehat{\mathrm{flip}}_{\eta,B}$ on $\boldsymbol{x}$ and continue using $K$, as in the rest of this section). Since this is just a rewrite of the empirical flip rate at $K$,

$$\phi_B(K) = \phi_B(B-K) \quad \text{(symmetry about } B/2\text{).}$$

This function is symmetric in $K$ about $B/2$ and unimodal. A one-step discrete difference shows it is strictly increasing on the left half:

$$\Delta(K) := \phi_B(K+1) - \phi_B(K) = \frac{2\,[\,B - 2K - 1\,]}{B(B-1)} > 0 \quad \text{for } K < \frac{B-1}{2}.$$

Moreover, $\Delta(K) = 0$ at $K = \frac{B-1}{2}$ and $\Delta(K) < 0$ for $K > \frac{B-1}{2}$, so $\phi_B$ increases up to the center and then decreases (unimodal).

Therefore, on the symmetric acceptance band

$$\mathcal{A}_\alpha = \{\, K : k_{\mathrm{L}} \leq K \leq B - k_{\mathrm{L}} \,\},$$

the minimum flip occurs at the endpoints $K = k_{\mathrm{L}}$ or $K = B - k_{\mathrm{L}}$, and both give the same value by symmetry. And so, the **empirical flip cutoff** at level $\alpha$ is

$$t_\alpha(B) := \frac{2\,k_{\mathrm{L}}\,(B - k_{\mathrm{L}})}{B(B-1)}. \tag{23}$$

Equivalently, similar to Equation 12, in vote-fraction form with $v_{\mathrm{L}} = k_{\mathrm{L}}/B$,

$$t_\alpha(B) = \frac{2B}{B-1}\,v_{\mathrm{L}}\,(1 - v_{\mathrm{L}}).$$

We declare the MIA decision for sample $\boldsymbol{x}$ *indistinguishable from a coin flip at level* $\alpha$ if

$$\widehat{\mathrm{flip}}_{\eta,B}(\boldsymbol{x}) \geq t_\alpha(B).$$

Because the binomial is discrete, equal-tail tests are slightly conservative. We follow the convention "reject if $p < \alpha$" and "fail to reject if $p \geq \alpha$," so boundary points with $p$-value $= \alpha$ lie inside the acceptance region. This yields the monotone flip rule $\widehat{\mathrm{flip}}_{\eta,B}(\boldsymbol{x}) \geq t_\alpha(B)$.

For the experiments in Section 5 and Appendix E.2.5, we set $\alpha{=}0.05$. For the 302M model we have $B{=}127$ target replicas and for the 140M model we have $B{=}125$ target replicas, respectively:

- $B{=}127$: $k_{\mathrm{L}}{=}52$ (so the acceptance band is $K \in [52,\,75]$) and

$$t_{0.05}(127) = \frac{2 \cdot 52 \cdot 75}{127 \cdot 126} \approx 0.487.$$

- $B{=}125$: $k_{\mathrm{L}}{=}52$ (acceptance band $K \in [52,\,73]$) and

$$t_{0.05}(125) = \frac{2 \cdot 52 \cdot 73}{125 \cdot 124} \approx 0.490.$$

(For $B{=}125$, our threshold is more conservative in part because the discrete CDF lands at $k_{\mathrm{L}}{=}52$, but a normal continuity-corrected approximation puts it closer to 51.6.)

### E.2.5 Extended results on flip rate

We provide results for two model architectures: 140M and 302M. We use the same training dataset size for both: overall $2N{=}2^{20}$, so models are trained on $N{=}2^{19}{=}524{,}288{\approx}500\mathrm{K}$ samples each. Note that for both architectures, this training dataset size is significantly smaller than what is Chinchilla optimal ($\approx$7M for the 140M model and $\approx$15.1M for the 302M model). As a result, we expect attack success to be higher (as measured by ROC-AUC) compared to Chinchilla-optimal trained and attacked models (Sections 3.2 & 4.2). For each model, we train one set of 128 reference models (with

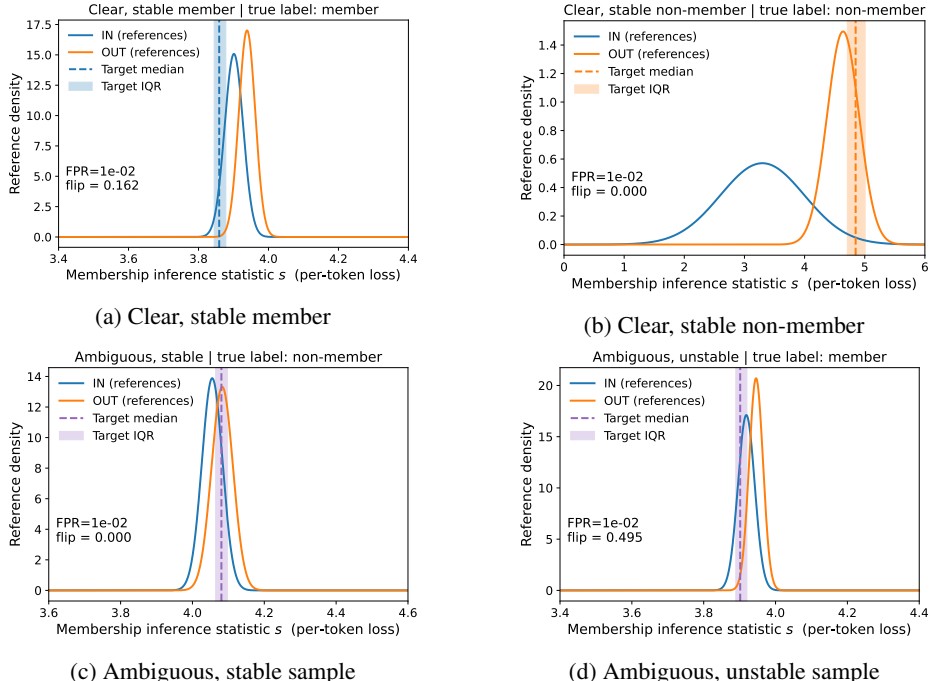

(a) Clear, stable member

(b) Clear, stable non-member

(c) Ambiguous, stable sample

(d) Ambiguous, unstable sample

Figure 22: **Different sample "archetypes" for the 140M target models.** We plot the per-sample $x$'s reference distributions (IN and OUT), median target statistic $s$ (and IQR) for $x$ across the 125 targets at FPR$=10^{-2}$ for four different $x$: (**a**) clear, stable member; (**b**) clear, stable non-member; (**c**) ambiguous, stable sample; and, (**d**) ambiguous, unstable sample. We annotate each plot with the sample's true label and empirical flip rate. For this architecture, we also provide snippets for the text of each sample in the main text.

0.5 probability that each sample is included as a member, so that member and non-member classes are balanced). To measure flip rate, we then train many target models on the *exact same* training dataset (i.e., the member and non-member samples are the same for all targets). The only difference across models is the random seed, which controls the batch order in which samples are surfaced to the training algorithm.

We intended to train 128 target replicas for each architecture; however, some runs crashed (and we ran out of time to re-run them), so in all we have 125 targets for the 140M model and 127 for the 302M model. As noted in Appendix E.2.4, the minimum values that we consider indistinguishable from a coin flip for $\widehat{\mathrm{flip}}_{\eta,B}$ are $t_{0.05}(125)\approx0.490$ for the 140M model and $t_{0.05}(127)\approx0.487$ for the 302M model.

**An intuition for per-sample flip rate.** Flip rate captures a sample $x$'s membership inference instability, computed across a set of target models where the only difference is the random seed that controls batch order. For a given $x$, it captures how much cross-decision disagreement there is—how much the MIA decisions for $x$ flip between both classes for equally plausible targets $r\sim\mu$.

To give a sense of how this can happen, we provide plots at the sample-level that show where target membership observation statistics for a given $x$ fall in relation to $x$'s IN and OUT reference distributions, $p_{\mathrm{IN}}(\cdot|x)$ and $p_{\mathrm{OUT}}(\cdot|x)$—fitted from the statistics obtained for $x$ using the reference sets $\Phi_{\mathrm{IN}}$ and $\Phi_{\mathrm{OUT}}$, respectively. We plot four "archetypes" that capture different patterns in sample-specific MIA decision behavior, in relation to reference distributions: (a) clear, stable member; (b) clear, stable non-member; (c) ambiguous, stable sample; and, (d) ambiguous, unstable sample. We identify these archetypes at FPR$=10^{-2}$. In Figure 22, we plot all four archetypes for the 140M architecture. In Figure 23, we plot archetypes (b)–(d), as we are unable to find clear, stable members at FPR$=10^{-2}$. Even for the 140M model, we have to relax the flip rate in our search filter to allow for $\widehat{\mathrm{flip}}_{10^{-2},125}\leq0.2$ to identify a "stable" member (when arguably, such a flip rate is not particularly stable). We are unable to satisfy this relaxed filter for the 302M architecture.

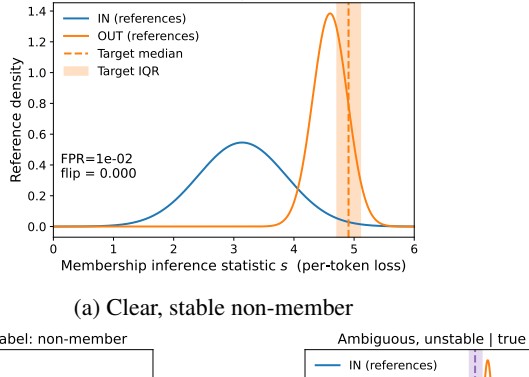

(a) Clear, stable non-member

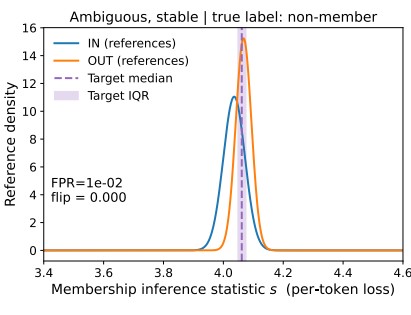

(b) Ambiguous, stable sample

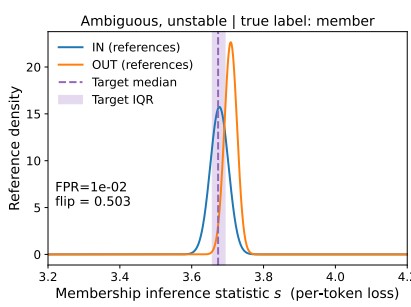

(c) Ambiguous, unstable sample

Figure 23: **Different sample "archetypes" for the 302M target models.** We plot the per-sample $x$'s reference distributions (IN and OUT), median target statistic $s$ (and IQR) for $x$ across the 127 targets at FPR$=10^{-2}$ for four different $x$: (**a**) clear, stable non-member; (**b**) ambiguous, stable sample; and, (**c**) ambiguous, unstable sample. We annotate each plot with the sample's true label and empirical flip rate. We are unable to identify a clear, moderately stable ($\widehat{\text{flip}}_{10^{-2},127} \leq 0.2$) member sample.

Note that, for both model sizes, the IN and OUT reference distributions overlap considerably for member samples. This overlap is a reasonable explanation for MIA decision instability: if LiRA has difficulty between establishing signal between members and non-members, then this will understandably impact the reliability of MIA decisions. Across targets trained on different random seeds, this can also manifest as the MIA decision flipping from one class to the other. In contrast, we identify cases for non-member samples where there is clear separation of IN and OUT reference distributions (Figures 22b & 23a).

For the 140M archetypes, we include short snippets of the text for each sample:

140M*: clear, stable member.* "Whether it's your first time looking for a Personal Trainer and you are just starting out, or you are a veteran who has been around a long-time, SINA Fitness can help you reach your fitness goals. Our Trainers are experienced, friendly and very energetic. We will help you set your fitness and lifestyle goals and most importantly help you achieve them. . . . "

140M*: clear, stable non-member.* "Ä Release Notes: AI War is an entirely unique large-scale RTS with aspects of TBS, tower defense, and grand strategy. It features single or cooperative play with as many as 8 humans against a pair of powerful, intelligent AIs. These AIs are driven by an AI Progress stat that players contribute to through aggressive actions such as taking control of planets and destroying key units, forcing tough decisions regarding which targets are worth capturing or destroying. . . . "

140M*: ambiguous, stable sample.* "The Gingrich commentary came hours after The Wall Street Journal reported that Mueller empaneled a grand jury.

"The Mueller threat has probably been the most deadly, he has the power of the law, he has the ability to indict people, the ability to negotiate and let some people off if they'll testify against other people," said Gingrich, also a Fox News contributor. . . . "

140M*: ambiguous, unstable sample.* "Winner of the Junior Australian Open 2015 Tereza Mihalikova (20), who is going to participate at EMPIRE Women's Indoor 2019 tournament, had spent the entire 2018 season under the guidance of tennis coach Martin Hromec. At the end of the year, the well-

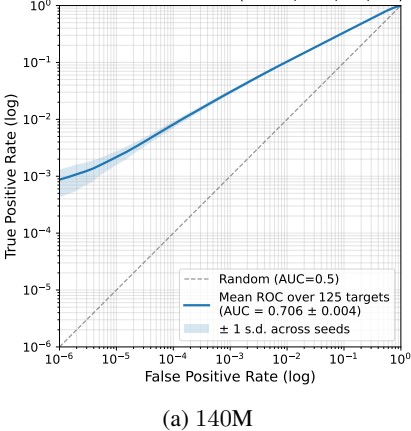
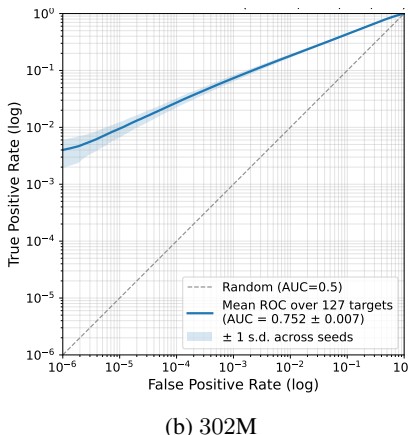

(a) 140M                                                 (b) 302M

Figure 24: **Averaged ROC curves and AUC across targets.** We plot the mean ROC across targets ($B$=125 for the 140M architecture; $B$=127 for the 302M architecture), and $\pm1$ STD across seeds. Both models are trained on substantially fewer samples than is Chinchilla optimal ($\approx$500K, compared to $\approx$7M and $\approx$15.1M, respectively). Mean ROC-AUC is higher than for Chinchilla-optimal models (as in Section 4.2). For the 140M model (**left**), ROC-AUC=0.706 $\pm$ 0.004; for the 302M model (**right**), ROC-AUC=0.752 $\pm$ 0.007. These are not typical attack ROC curves, as they average over results for multiple targets. In the standard MIA threat model, the attacker only has access to a single target. These plots give a sense of the stability of overall attack performance, as computed over equally plausible targets $r \sim \mu$ where the only difference in targets is the seed controlling batch order.

known fitness coach Jozef Ivanko, strengthened the team. Ivanko worked with Top 10 players in WTA ranking already. . . . ”

**Aggregate attack performance for MIA flip-rate experiments.** While our main focus here is to measure per-sample instability, as a point of comparison, we also include measurements about attack averages. For both model sizes, we include mean (cross-seed) ROC-AUC metrics and curves (Figure 24) and associated tables that show average (cross-seed) accuracy and error rates by class at fixed FPR (Tables 1 & 2).

We again emphasize that the models we attack in these experiments were *not* trained on the Chinchilla-optimal number of tokens ($\approx$7M and $\approx$15.1M samples, for 140M and 302M models, respectively; see Section 3.2 & Appendix C). Both sets of experiments involved training models on only $\approx$500K samples. As a result, we expect (and do observe) attack performance (in terms of ROC-AUC) to be higher than in the Chinchilla-optimal setting (Section 4.2 & Appendix D). For the 140M architecture, we observe average AUC=0.706 $\pm$ 0.004 across the 125 targets (Figure 24a); for the 302M architecture, we observe average AUC=0.752 $\pm$ 0.007 across the 127 targets (Figure 24b). In both cases, AUC is stable across targets (as indicated by the low standard deviation). This same pattern of stability in overall attack metrics is also clear in Tables 1 and 2: accuracy and error exhibit low standard deviation, with respect to these rates being aggregated across all samples (conditioned by class) and averaged across targets trained with different seeds.

For an alternate view of these results, we also include direct comparisons of attack performance (as measured by average TPR $\pm$ standard deviation at fixed FPR) and variability in the underlying decision rule (with respect to threshold $\tau$) across targets. In Figure 25, we provide these comparisons for both the 140M and 302M model sizes. Of course, as is also surfaced by ROC curves (Figure 24) at very low fixed FPR, the TPR is also low. Here, we also show how this naturally results in a very high decision threshold $\tau$, which also exhibits low variability. As we increase FPR, TPR also increases and remains stable, with respect to low standard deviation. However, the underlying decision rules for the targets can vary considerably; the underlying targets can have very different $\tau$. This result is consistent with prior work on model and predictive multiplicity (Appendix E.2.3): models with similar overall accuracy can have very different underlying decision rules. As we address further below in this appendix and in Section 5, even when overall accuracy is similar, the different decision rules can result in very different/disagreeing membership MIA decisions for the same samples.

Table 1: **140M-parameter model error rate metrics.** We report accuracy-related metrics as a function of fixed FPR. Entries are rates (not percentages), as elsewhere in this paper. We report mean $\pm$ STD where applicable. Since we fix FPR, there is no STD to report. Since $1 -$ FPR = TNR, similarly, there is no STD to report. ACC $= \frac{\text{TP+TN}}{N}$, with $2N$=1,048,576. Typically reported log-scale FPR rows are highlighted in gray.

| FPR | ACC All | FNR Members | FPR Non-members | TNR Non-members | TPR Members |
|---|---|---|---|---|---|
| $10^{-5}$ | $0.501 \pm 0.0$ | $0.998 \pm 0.001$ | $0.0$ | $1.0$ | $0.002 \pm 0.001$ |
| $10^{-4}$ | $0.504 \pm 0.001$ | $0.992 \pm 0.001$ | $0.0$ | $1.0$ | $0.008 \pm 0.001$ |
| $10^{-3}$ | $0.515 \pm 0.001$ | $0.97 \pm 0.002$ | $0.001$ | $0.999$ | $0.03 \pm 0.002$ |
| $10^{-2}$ | $0.547 \pm 0.002$ | $0.896 \pm 0.005$ | $0.01$ | $0.99$ | $0.104 \pm 0.005$ |
| $0.02$ | $0.564 \pm 0.003$ | $0.852 \pm 0.005$ | $0.02$ | $0.98$ | $0.148 \pm 0.005$ |
| $0.05$ | $0.593 \pm 0.003$ | $0.764 \pm 0.006$ | $0.05$ | $0.95$ | $0.236 \pm 0.006$ |
| $10^{-1}$ | $0.618 \pm 0.003$ | $0.664 \pm 0.006$ | $0.1$ | $0.9$ | $0.336 \pm 0.006$ |
| $0.2$ | $0.64 \pm 0.003$ | $0.52 \pm 0.006$ | $0.2$ | $0.8$ | $0.48 \pm 0.006$ |
| $0.5$ | $0.633 \pm 0.002$ | $0.234 \pm 0.004$ | $0.5$ | $0.5$ | $0.766 \pm 0.004$ |
| $0.75$ | $0.582 \pm 0.001$ | $0.085 \pm 0.002$ | $0.75$ | $0.25$ | $0.915 \pm 0.002$ |
| $10^{0}$ | $0.5 \pm 0.0$ | $0.0 \pm 0.0$ | $1.0$ | $0.0$ | $1.0 \pm 0.0$ |

Table 2: **302M-parameter model error rate metrics.** We report accuracy-related metrics as a function of fixed FPR. Entries are rates (not percentages), as elsewhere in this paper. We report mean $\pm$ STD where applicable. Since we fix FPR, there is no STD to report. Since $1 -$ FPR = TNR, similarly, there is no STD to report. ACC $= \frac{\text{TP+TN}}{N}$, with $2N$=1,048,576. Typically reported log-scale FPR rows are highlighted in gray.

| FPR | ACC All | FNR Members | FPR Non-members | TNR Non-members | TPR Members |
|---|---|---|---|---|---|
| $10^{-5}$ | $0.505 \pm 0.001$ | $0.991 \pm 0.003$ | $0.0$ | $1.0$ | $0.009 \pm 0.003$ |
| $10^{-4}$ | $0.514 \pm 0.003$ | $0.973 \pm 0.005$ | $0.0$ | $1.0$ | $0.027 \pm 0.005$ |
| $10^{-3}$ | $0.536 \pm 0.005$ | $0.927 \pm 0.009$ | $0.001$ | $0.999$ | $0.073 \pm 0.009$ |
| $10^{-2}$ | $0.585 \pm 0.007$ | $0.819 \pm 0.013$ | $0.01$ | $0.99$ | $0.181 \pm 0.013$ |
| $0.02$ | $0.608 \pm 0.007$ | $0.765 \pm 0.014$ | $0.02$ | $0.98$ | $0.235 \pm 0.014$ |
| $0.05$ | $0.642 \pm 0.007$ | $0.667 \pm 0.014$ | $0.05$ | $0.95$ | $0.333 \pm 0.014$ |
| $10^{-1}$ | $0.668 \pm 0.007$ | $0.565 \pm 0.014$ | $0.1$ | $0.9$ | $0.435 \pm 0.014$ |
| $0.2$ | $0.685 \pm 0.006$ | $0.43 \pm 0.013$ | $0.2$ | $0.8$ | $0.57 \pm 0.013$ |
| $0.5$ | $0.655 \pm 0.004$ | $0.191 \pm 0.008$ | $0.5$ | $0.5$ | $0.809 \pm 0.008$ |
| $0.75$ | $0.588 \pm 0.002$ | $0.075 \pm 0.004$ | $0.75$ | $0.25$ | $0.925 \pm 0.004$ |
| $10^{0}$ | $0.5 \pm 0.0$ | $0.0 \pm 0.0$ | $1.0$ | $0.0$ | $1.0 \pm 0.0$ |

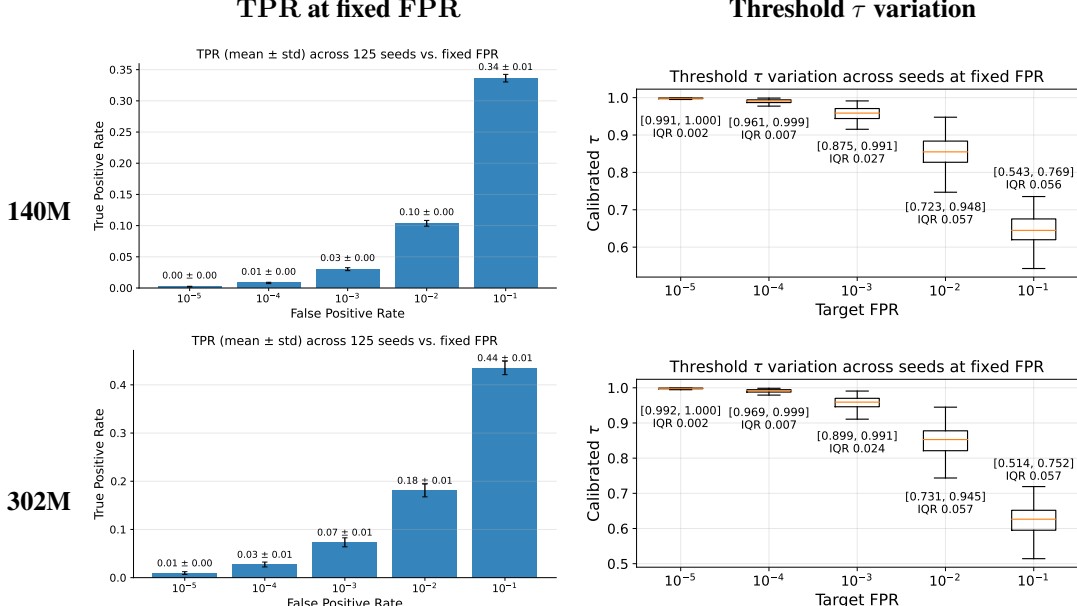

Figure 25: **Comparing attack performance and decision thresholds at fixed FPR.** Row shows results for model architectures: 140M and 302M The left column shows mean $\pm$ standard deviation of the attack TPR at different fixed FPR, computed across $B$ targets. The right column shows the decision threshold $\tau$ range (and IQR) at fixed FPR across $B$ targets, where the decision threshold for each target is calibrated with respect to non-member samples (Section 2 & Appendix A). As is also surfaced by ROC curves (Figure 24) at very low fixed FPR, the TPR is also low. Here, we also show how this naturally results in a very high decision threshold $\tau$, which consequently exhibits low variability. As we increase FPR, TPR also increases and remains stable, with respect to low standard deviation. However, the underlying decision rules for the targets start to vary considerably; the underlying targets can have very different $\tau$, which (as we address in this appendix and in Section 5) can result in very different/disagreeing membership decisions for the same sample $\boldsymbol{x}$.

**Sample flip rate variation at a single fixed** FPR. We provide three complementary views (by class) at fixed FPR to characterize per-sample instability: **(left)** complementary CDFs (CCDFs) of flip rate; **(middle)** flip rate vs. mean absolute distance to the calibrated decision boundary; and **(right)** flip rate vs. mean LiRA posterior. We show these results for the 140M and 302M models in Figures 26 and 27, respectively, each for FPR $\in \{10^{-5}, 10^{-4}, 10^{-3}, 10^{-2}, 10^{-1}\}$.

For the middle and right columns, the $x$-axis uses equal-count (quantile) bins so that each plotted point aggregates (essentially) the same number of samples; points are therefore directly comparable across the curves. Together with Tables 3 and 4, these plots support our main takeaway in Section 5: *aggregate* attack metrics (e.g., mean TPR at fixed FPR, ROC-AUC) can look stable while many *individual* membership decisions are indistinguishable from a coin flip.

We organize observations by theme:

- **Flip rate rises with FPR.** As is clear from the complementary CDFs for flip rate (left column), flip rate rises with FPR. This is because, as FPR grows, the per-seed calibrated threshold $\tau_r(\eta)$ moves down (right-tail quantile of the non-member distribution), into regions where IN/OUT distribution overlap is more extensive. This boundary shift puts $\tau_r(\eta)$ in score regions where many member sample posteriors lie, and also increases the proportion of samples whose seed-specific scores lie near the boundary. As a result, small seed-induced score shifts (as well as across-seed variation in $\tau_r(\eta)$ itself, see Figure 25) flip the decision more often (i.e., increase per-sample MIA decision disagreement). (More non-members will also be labeled as members, by construction; so, too will members.)

  The effect is modest at very low FPR, where $\tau_r(\eta)$ sits deep in the extreme tail. But it becomes more pronounced as we increase FPR (i.e., as the boundary moves toward denser parts of the score distribution). The CCDFs (left column) and distance plots (middle column) both show this pattern: at FPR$=10^{-1}$, $\approx 70\%$ of *members* for the 302M targets have $\widehat{\text{flip}}_{10^{-1},127} \geq 0.4$ vs. $\approx 7\%$ of non-members; for the 140M targets the corresponding figures for $\widehat{\text{flip}}_{10^{-1},125} \geq 0.4$ are $\approx 49\%$ vs. $\approx 8\%$ (see Table 4 and Figure 27, left; Table 3 and Figure 26, left).

- **Mean absolute distance to the boundary is a direct proxy for instability.** For each sample $\boldsymbol{x}$, we define per-seed distance to the decision boundary and a cross-seed measure of closeness to the boundary, regardless of side:

$$d_r(\boldsymbol{x}) = \Lambda_r(\boldsymbol{x}) - \tau_r(\eta); \quad |\overline{d}|(\boldsymbol{x}) = \tfrac{1}{B} \sum_r |d_r(\boldsymbol{x})|.$$

  For associated plots (middle column), quantile bins with small $|\overline{d}|$ put the sample close to the decision boundary, resulting in high flip rate (i.e., many MIA decisions disagree). Quantile bins with larger $|\overline{d}|$ are more reliably on one side of the decision boundary, which results in a lower flip rate (i.e., more decisions concentrate). At FPR$=10^{-1}$ for both model sizes, member flip rate is persistently high across a wide range of $|\overline{d}|$—evidence that IN/OUT overlap is substantial in the region where $\tau_r(\eta)$ lies for many seeds (middle column). In general, members exhibit markedly higher flip rate than non-members at the same $|\overline{d}|$, with the differences in the two becoming wider at higher FPR.

- **Flip vs. mean posterior is non-monotone at high** FPR. For the plots in the right column, we define the mean posterior across targets as

$$\overline{\Lambda}(\boldsymbol{x}) = \tfrac{1}{B} \sum_r \Lambda_r(\boldsymbol{x}).$$

  Flip rate increases as $\overline{\Lambda}(\boldsymbol{x})$ approaches $\tau_r(\eta)$ (more seeds straddle the boundary) and then *declines* once $\overline{\Lambda}(\boldsymbol{x})$ is well above the boundary for most seeds (decisions re-concentrate on "member"). This non-monotonicity is most visible at FPR$=10^{-1}$ (right column). (We similarly see this in the middle column, which directly plots distances to the boundary.)

- **Members flip much more than non-members, and the gap widens with model size.** Two forces seem to drive this. First, there is *structural asymmetry across classes from calibration* (See box, Appendix E.2.2). Thresholds are calibrated on non-members for each seed-specific target (Section 2 & Appendix A), so $\tau_r(\eta)$ tracks seed-to-seed shifts in the non-member distribution by design, and many non-members remain far below $\tau_r(\eta)$ for modest FPR. In contrast, the threshold is not anchored to the member score distribution. Often, their scores straddle the moving decision boundary, so, small seed-induced shifts (either in the score or the decision

boundary, see Figure 25) can flip the decision. This effect is more pronounced for higher settings of FPR, which push the threshold into a higher density region of the member score distribution. Second, there is *greater across-seed score variability for members*. Intuitively, training randomness primarily perturbs samples seen in training (as opposed to those that are not). Empirically, IN/OUT reference distributions for many members overlap substantially, while some non-members exhibit clearer separation (Figures 22 & 23). Both of these effects are stronger for the larger model 302M (compare Table 4 vs. Table 3).

- **Effect of model size.** In general, the observations above show that flip rate instability is worse for the larger (302M) model. Members flip much more than non-members, and the gap widens with model size. These results are also consistent with model-multiplicity-related results for higher capacity models: those with similar aggregate accuracy can exhibit more disagreement at the individual sample level [14].

**Flip rate over varied fixed** FPR. We summarize across operating points in Figures 28 and 29. Each contains five sub-plots that show, for a given flip rate range, the class-conditional proportion of samples in each range as a function of FPR (with the corresponding mean TPR $\pm$ STD annotated above the panels). We use the disjoint ranges $[0, 0.1)$ (very stable), $[0.1, 0.25)$ (low/mid stable), $[0.25, 0.4)$ (mid/high unstable), $[0.4, t_\alpha(B))$ (very unstable), $[t_\alpha(B), \widehat{\text{flip}}_{\eta,B}^{(\max)}]$, where for $B{=}125$ we take $t_{0.05}(125){\approx}0.490$ and $\widehat{\text{flip}}_{\eta,125}^{(\max)}{=}0.504$, and for $B{=}127$ we take $t_{0.05}(127){\approx}0.487$ and $\widehat{\text{flip}}_{\eta,127}^{(\max)}{\approx}0.50394$.

Our values for $t_{0.05}(125)$ and $t_{0.05}(127)$ are obtained from the exact two-sided binomial acceptance region (Appendix E.2.4). That is, we compute $t_\alpha(B) = \frac{2\,k_\mathrm{L}\,(B-k_\mathrm{L})}{B(B-1)}$ with $k_\mathrm{L}$ chosen as the smallest integer such that $F(k_\mathrm{L}) \geq \alpha/2$ for $K \sim \text{Binomial}(B, 1/2)$. $\widehat{\text{flip}}_{\eta,B}^{(\max)} = \frac{2\lfloor B/2 \rfloor \lceil B/2 \rceil}{B(B-1)}$ (Appendix E.2.1).

These summary curves reinforce the fixed-FPR views above: flip rate increases with FPR, members flip far more than non-members at all reasonable FPR (i.e., FPR$\lesssim$0.2), and the 302M model shows larger gaps between members and non-members as well as mass in the statistically indistinguishable-from-a-coin-flip range.

**Flip CCDF**          **Flip vs. mean |distance|**          **Flip vs. mean posterior**

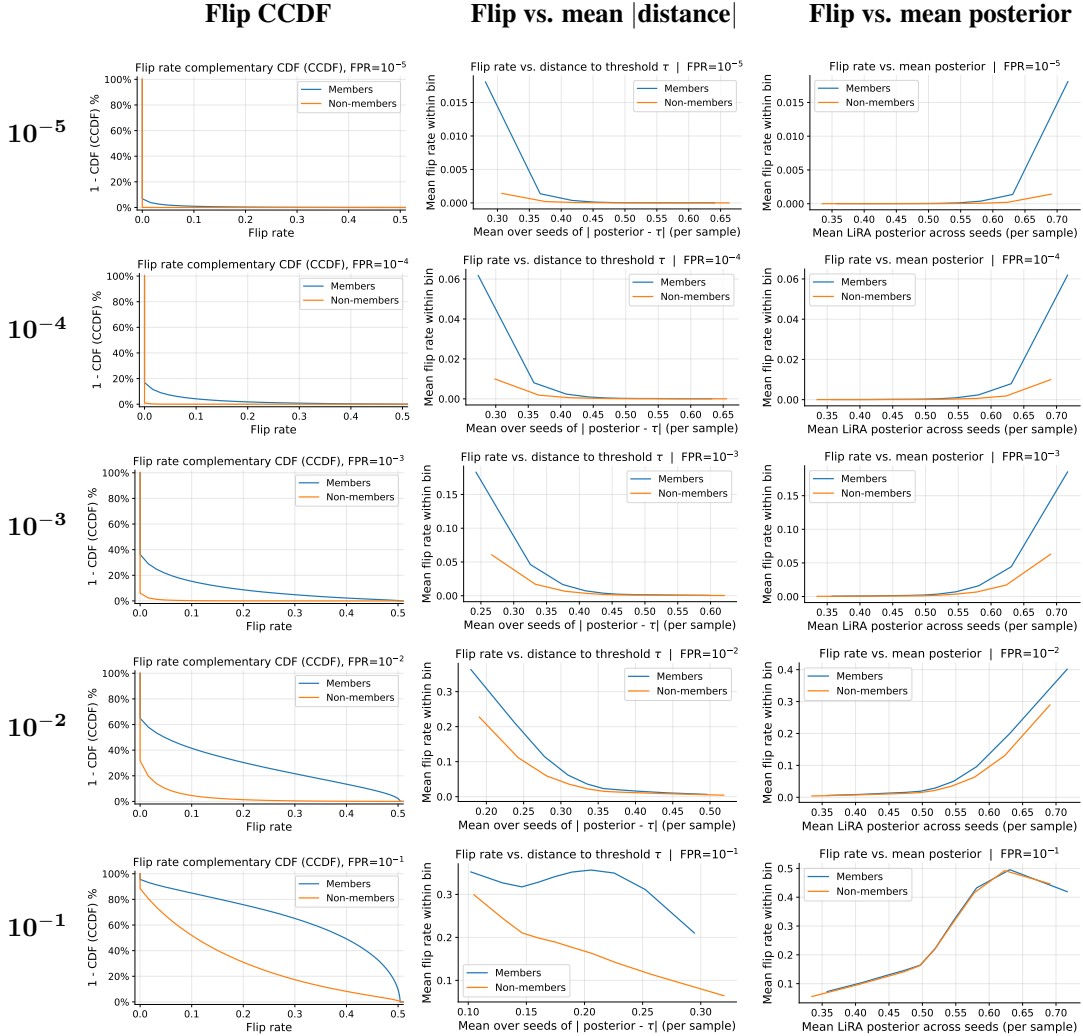

Figure 26: **Detailed flip rate results for the 140M model.** For the 140M model, the number of target replicas $B{=}125$. (**rows**) For different fixed FPR $\in \{10^{-5}, 10^{-4}, 10^{-3}, 10^{-2}, 10^{-1}\}$, we provide (**columns**) three different views on flip rate across member and non-member samples. (**left**) We plot the empirical complementary CDF (CCDF) for flip rate, conditioned on membership status. Higher curves indicate more instability. As FPR increases, the differences in flip rate across classes become more pronounced. While flip rate is minimal at low FPR, it is substantial—particularly for members—at higher FPR. For example, at FPR$=10^{-1}$ approximately $50\%$ of member samples exhibit $\widehat{\text{flip}}_{10^{-1},125} {\geq} 0.4$, compared to approximately $10\%$ of non-members. (**middle**) We plot flip rate as a function of the mean of the magnitude of the distance (per sample) from the posterior to the decision threshold $\tau$. Further to the left means closer to $\tau$. This shows instability in terms of the distance to the threshold (regardless of direction of that distance). For higher FPR, the flip rate for members is much higher than for non-members for the same mean absolute distance to $\tau$. (**right**) We plot flip rate as a function of the mean posterior. For higher FPR, the member and non-member flip rates are more similar as a function of the mean LiRA posterior. For the last two columns, we use quantile (i.e., equal-count) bucketing on the $x$-axis so that each plotted point is based on (essentially) the same number of samples. That way, points on the curves are directly comparable.

**Flip CCDF**          **Flip vs. mean |distance|**          **Flip vs. mean posterior**

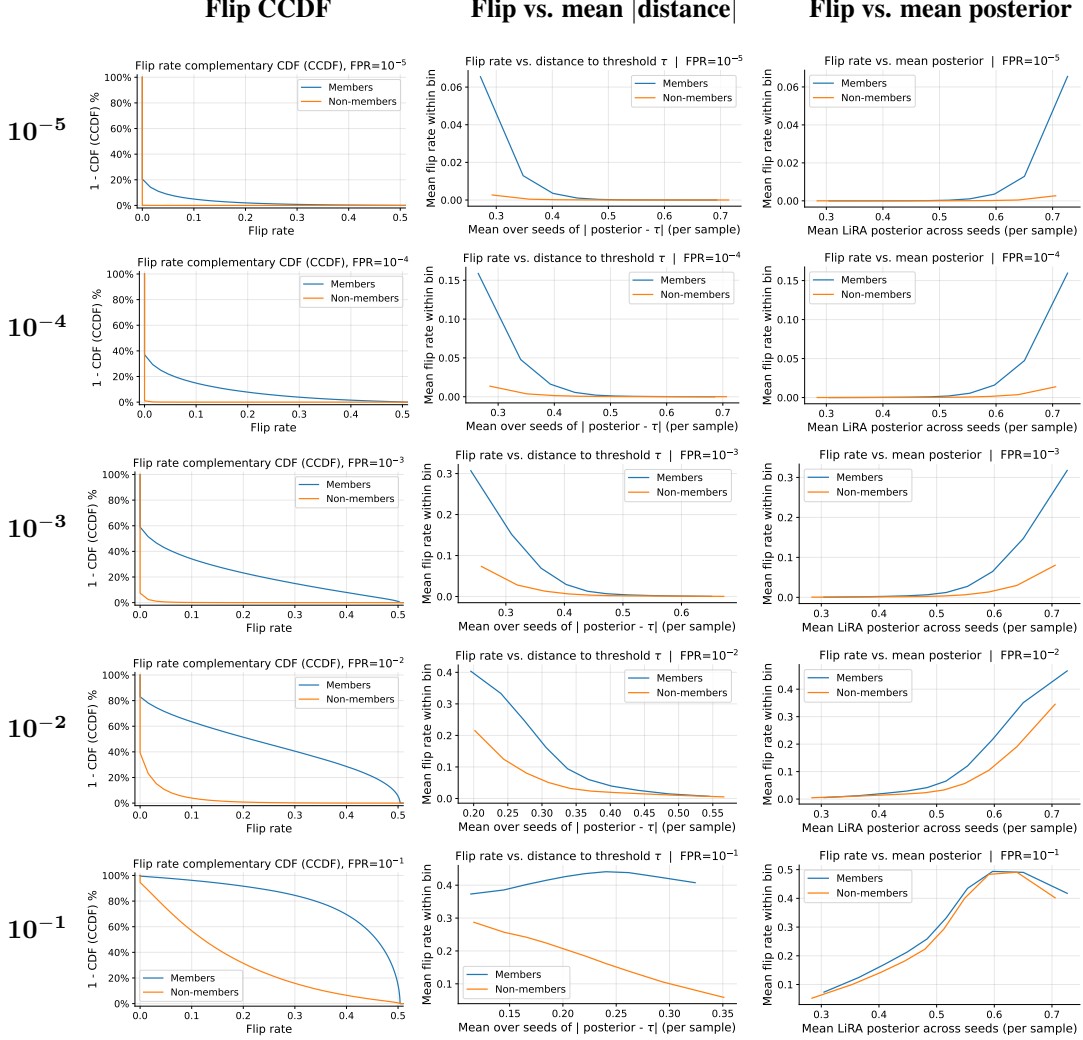

Figure 27: **Detailed flip rate results for the 302M model.** For the 302M model, the number of target replicas $B=127$. (**rows**) For different fixed FPR $\in \{10^{-5}, 10^{-4}, 10^{-3}, 10^{-2}, 10^{-1}\}$, we provide (**columns**) three different views on flip rate across member and non-member samples. (**left**) We plot the empirical complementary CDF (CCDF) for flip rate, conditioned on membership status. Higher curves indicate more instability. As FPR increases, the differences in flip rate across classes become more pronounced. While flip rate is minimal at low FPR, it is substantial—particularly for members—at higher FPR. For example, at FPR$=10^{-1}$ approximately 70% of member samples exhibit $\widehat{\text{flip}}_{10^{-1},127} \geq 0.4$, compared to approximately 10% of non-members. (**middle**) We plot flip rate as a function of the mean of the magnitude of the distance (per sample) from the posterior to the decision threshold $\tau$. Further to the left means closer to $\tau$. This shows instability in terms of the distance to the threshold (regardless of direction of that distance). For higher FPR, the flip rate for members is much higher than for non-members for the same mean absolute distance to $\tau$. (**right**) We plot flip rate as a function of the mean posterior. For higher FPR, the member and non-member flip rates are more similar as a function of the mean LiRA posterior. For the last two columns, we use quantile (i.e., equal-count) bucketing on the $x$-axis so that each plotted point is based on (essentially) the same number of samples. That way, points on the curves are directly comparable.

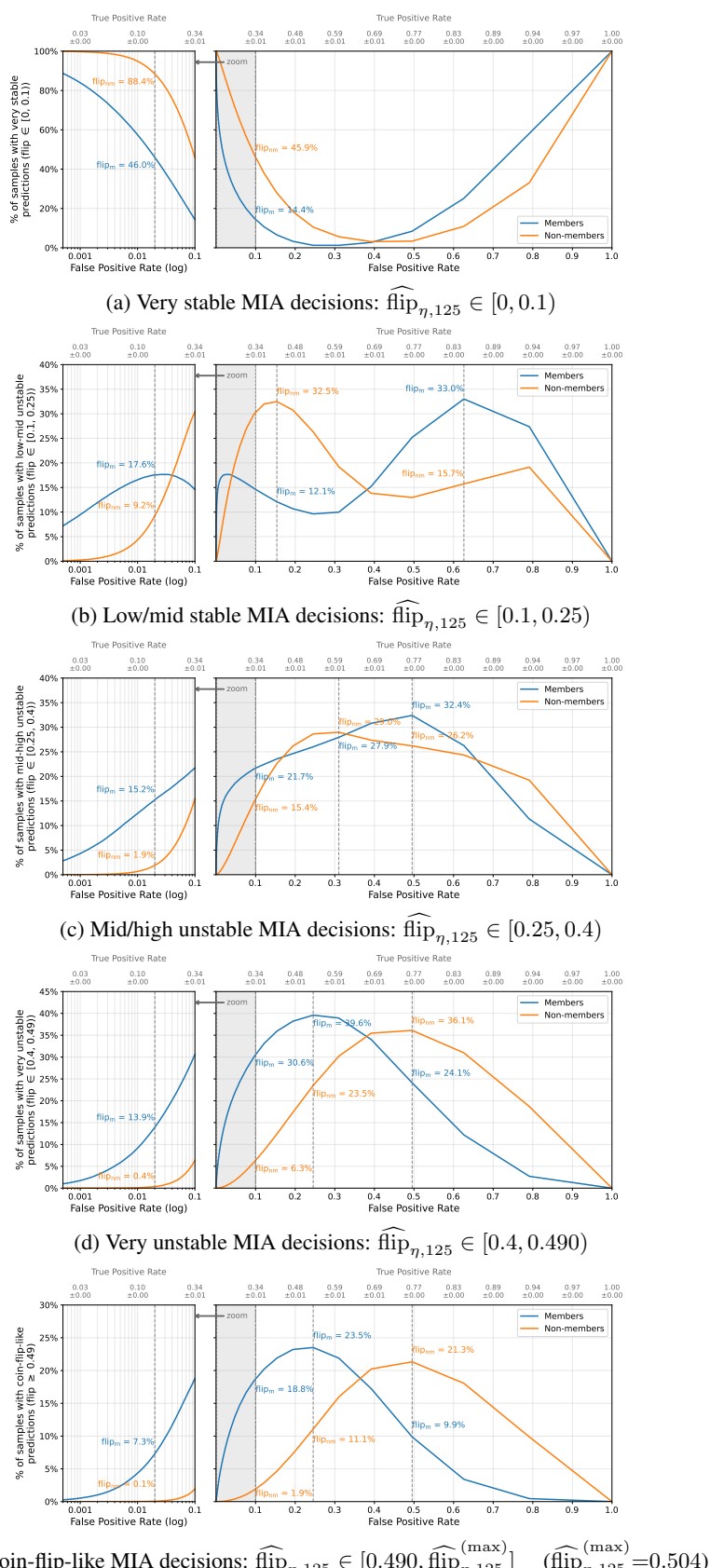

(a) Very stable MIA decisions: $\widehat{\mathrm{flip}}_{\eta,125} \in [0, 0.1)$

(b) Low/mid stable MIA decisions: $\widehat{\mathrm{flip}}_{\eta,125} \in [0.1, 0.25)$

(c) Mid/high unstable MIA decisions: $\widehat{\mathrm{flip}}_{\eta,125} \in [0.25, 0.4)$

(d) Very unstable MIA decisions: $\widehat{\mathrm{flip}}_{\eta,125} \in [0.4, 0.490)$

(e) Coin-flip-like MIA decisions: $\widehat{\mathrm{flip}}_{\eta,125} \in [0.490, \widehat{\mathrm{flip}}_{\eta,125}^{(\max)}]$     ($\widehat{\mathrm{flip}}_{\eta,125}^{(\max)} = 0.504$)

Figure 28: **Flip rate variation by fixed FPR for the 140M model**. For different ranges of $\widehat{\mathrm{flip}}_{\eta,125}$, we plot how class-conditional flip rate varies by FPR. We annotate plots with corresponding mean $\pm$ standard deviation for the corresponding TPR. See main text for additional discussion.

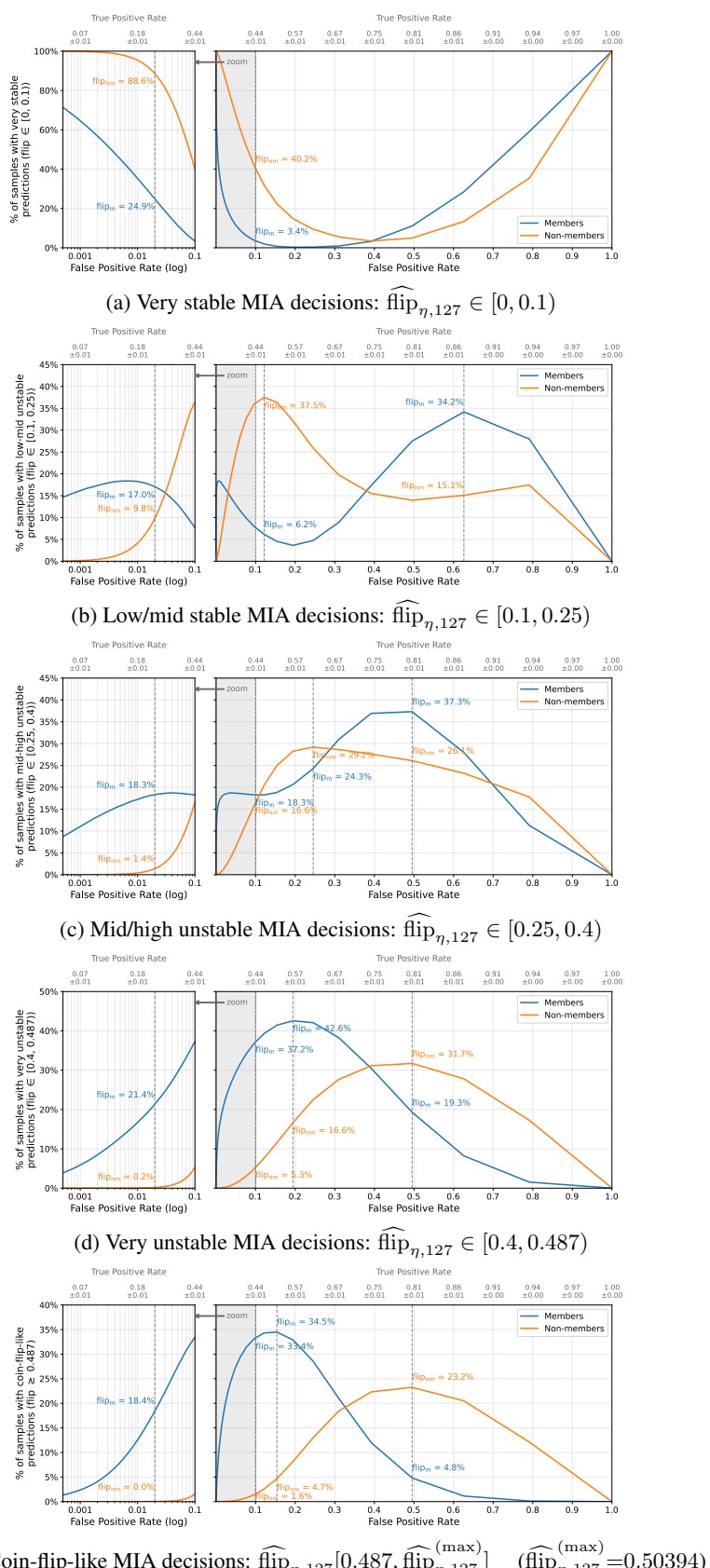

(a) Very stable MIA decisions: $\widehat{\mathrm{flip}}_{\eta,127} \in [0, 0.1)$

(b) Low/mid stable MIA decisions: $\widehat{\mathrm{flip}}_{\eta,127} \in [0.1, 0.25)$

(c) Mid/high unstable MIA decisions: $\widehat{\mathrm{flip}}_{\eta,127} \in [0.25, 0.4)$

(d) Very unstable MIA decisions: $\widehat{\mathrm{flip}}_{\eta,127} \in [0.4, 0.487)$

(e) Coin-flip-like MIA decisions: $\widehat{\mathrm{flip}}_{\eta,127}[0.487, \widehat{\mathrm{flip}}_{\eta,127}^{(\mathrm{max})}]$    $(\widehat{\mathrm{flip}}_{\eta,127}^{(\mathrm{max})} = 0.50394)$

Figure 29: **Flip rate variation by fixed FPR for the 302M model**. For different ranges of $\widehat{\mathrm{flip}}_{\eta,B}$, we plot how class-conditional flip rate varies by FPR. We annotate plots with corresponding mean $\pm$ standard deviation for the corresponding TPR. See main text for additional discussion.

Table 3: **140M-parameter model flip proportions for different FPR.** For different settings of FPR, we show the percentage of samples (by class) whose empirical flip lies in each range. Lower flip corresponds to more stable MIA decisions; values near $0.5$ indicate coin-flip-like behavior. We split the high-instability region into $[0.4, 0.490)$ and $[0.490, 0.504]$ to isolate cases that are statistically indistinguishable from a coin flip. The $\max$ population flip is $0.5$; with 125 seeds, the empirical $\max$ is $\approx 0.504$. ($t_{0.05}(125) \approx 0.490$ is the minimum value at level $\alpha=0.05$ that our hypothesis test yields; see Appendix E.2.4.) See Appendix E.2.1. Typically reported log-scale FPR rows are highlighted in gray.

| FPR | very stable flip $\in [0, 0.1)$ | | low/mid unstable flip $\in [0.1, 0.25)$ | | mid/high unstable flip $\in [0.25, 0.4)$ | | very unstable flip $\in [0.4, 0.49)$ | | coin flip flip $\geq 0.49$ | |
|---|---|---|---|---|---|---|---|---|---|---|
| | Mem. | Non-Mem. | Mem. | Non-Mem. | Mem. | Non-Mem. | Mem. | Non-Mem. | Mem. | Non-Mem. |
| $10^{-5}$ | 98.93% | 100.00% | 0.86% | 0.00% | 0.17% | 0.00% | 0.03% | 0.00% | 0.01% | 0.00% |
| $10^{-4}$ | 95.45% | 99.99% | 3.31% | 0.01% | 0.93% | 0.00% | 0.24% | 0.00% | 0.07% | 0.00% |
| $10^{-3}$ | 83.94% | 99.74% | 9.47% | 0.25% | 4.31% | 0.01% | 1.63% | 0.00% | 0.65% | 0.00% |
| $10^{-2}$ | 57.48% | 94.88% | 16.56% | 4.34% | 12.46% | 0.66% | 8.52% | 0.12% | 5.00% | 0.00% |
| 0.02 | 45.97% | 88.42% | 17.55% | 9.24% | 15.23% | 1.91% | 12.95% | 0.42% | 8.29% | 0.01% |
| 0.05 | 28.46% | 70.12% | 17.10% | 20.63% | 18.72% | 6.88% | 21.56% | 2.37% | 14.16% | 0.01% |
| $10^{-1}$ | 14.37% | 45.94% | 14.60% | 30.39% | 21.69% | 15.40% | 30.34% | 8.27% | 19.00% | 0.00% |
| 0.2 | 2.93% | 17.29% | 10.51% | 30.34% | 24.80% | 26.59% | 37.46% | 17.83% | 24.30% | 8.95% |
| 0.5 | 8.85% | 3.52% | 25.61% | 13.00% | 32.29% | 26.17% | 24.92% | 31.79% | 8.34% | 25.53% |
| 0.75 | 49.02% | 26.08% | 30.66% | 18.59% | 15.17% | 20.91% | 4.32% | 22.21% | 0.84% | 12.21% |
| $10^{0}$ | 100.00% | 100.00% | 0.00% | 0.00% | 0.00% | 0.00% | 0.00% | 0.00% | 0.00% | 0.00% |

Table 4: **302M-parameter model flip proportions for different FPR.** For different settings of FPR, we show the percentage of samples (by class) whose empirical flip lies in each range. Lower flip corresponds to more stable MIA decisions; values near $0.5$ indicate coin-flip-like behavior. We split the high-instability region into $[0.4, 0.490)$ and $[0.490, 0.50394]$ to isolate case that are statistically indistinguishable from a coin flip. The $\max$ population flip is $0.5$; with 127 seeds, the empirical $\max$ is $\approx 0.50394$. ($t_{0.05}(127) \approx 0.487$ is the minimum value at level $\alpha=0.05$ that our hypothesis test yields; see Appendix E.2.4.) See Appendix E.2.1. Typically reported log-scale FPR rows are highlighted in gray.

| FPR | very stable flip $\in [0, 0.1)$ | | low/mid unstable flip $\in [0.1, 0.25)$ | | mid/high unstable flip $\in [0.25, 0.4)$ | | very unstable flip $\in [0.4, 0.487)$ | | coin flip flip $\geq 0.487$ | |
|---|---|---|---|---|---|---|---|---|---|---|
| | Mem. | Non-Mem. | Mem. | Non-Mem. | Mem. | Non-Mem. | Mem. | Non-Mem. | Mem. | Non-Mem. |
| $10^{-5}$ | 94.46% | 100.00% | 4.17% | 0.00% | 1.09% | 0.00% | 0.23% | 0.00% | 0.05% | 0.00% |
| $10^{-4}$ | 84.15% | 100.00% | 10.06% | 0.00% | 4.15% | 0.00% | 1.31% | 0.00% | 0.33% | 0.00% |
| $10^{-3}$ | 64.59% | 99.85% | 16.15% | 0.14% | 11.01% | 0.00% | 5.87% | 0.00% | 2.38% | 0.00% |
| $10^{-2}$ | 35.27% | 95.41% | 18.24% | 4.14% | 17.25% | 0.40% | 16.69% | 0.04% | 12.55% | 0.00% |
| 0.02 | 24.87% | 88.61% | 17.04% | 9.81% | 18.29% | 1.39% | 21.38% | 0.17% | 18.41% | 0.03% |
| 0.05 | 11.42% | 67.44% | 12.96% | 24.64% | 18.67% | 6.40% | 29.49% | 1.27% | 27.46% | 0.24% |
| $10^{-1}$ | 3.45% | 40.16% | 7.67% | 36.41% | 18.28% | 16.59% | 37.22% | 5.28% | 33.39% | 1.56% |
| 0.2 | 0.26% | 14.07% | 3.62% | 31.40% | 20.97% | 28.54% | 42.65% | 17.33% | 32.49% | 8.66% |
| 0.5 | 11.57% | 5.13% | 27.88% | 13.98% | 37.14% | 26.02% | 18.81% | 31.66% | 4.60% | 23.21% |
| 0.75 | 50.77% | 28.47% | 31.22% | 17.07% | 15.27% | 19.34% | 2.53% | 20.53% | 0.22% | 14.59% |
| $10^{0}$ | 100.00% | 100.00% | 0.00% | 0.00% | 0.00% | 0.00% | 0.00% | 0.00% | 0.00% | 0.00% |

### E.2.6 How many MIA true positives are statistically indistinguishable from a coin flip?

From the above analysis, a natural follow-on is to attempt to estimate how many attack true positives are statistically indistinguishable from a coin flip. That is, when we report ROC-AUC at fixed FPR, how much of the corresponding TPR is composed of positive MIA decisions that are essentially a coin flip, rather than reflecting reliable inference signal? In this appendix, we use the results from our experiments in Appendix E.2.5 to estimate an answer to this question. We split members into different bins that correspond to ranges for flip rate, and estimate the count (and rate) of true positives for each bin.

This decomposition is a post-hoc audit tool that uses target replicas. An attacker facing a single target cannot know which of its true positives are indistinguishable from a coin flip, when making per-sample membership claims. Therefore, the takeaway here is about reliability of per-sample claims that an attacker makes, not about an attacker's observable signal. Our aim is to surface the extent of unreliability that may affect the attacker's claims (regardless of the attacker's knowledge of reliability of those claims).

We perform analysis that aggregates across attacks on multiple target models, so the numbers and figures we report are *not* comparable with single-attack results in the typical MIA setup. These results serve as diagnostics to assess attack reliability. Individual attacks on specific targets of course vary in their performance, and are not directly comparable to what we present here that aggregates over many such attacks to try to better understand overall properties of attack behavior.

**Decomposing attack TPR into contributions from flip rate bins.** Fix a dataset $\mathbb{D}_{\text{IN}}$ with $|\mathbb{D}_{\text{IN}}|=M$ member examples and $B$ i.i.d. target replicas $r_1, \ldots, r_B \sim \mu$. At a fixed FPR $\eta$, each target replica $r$ sets its threshold $\tau_r(\eta)$, calibrated on non-members (Section 2 & Appendix A).

For a member $\boldsymbol{x}$, define the per-seed decision

$$Y_r(\boldsymbol{x}) = \mathbf{1}\{\Lambda_r(\boldsymbol{x}) \geq \tau_r(\eta)\} \in \{0, 1\},$$

(This is just the binary membership decision rule $b_r^{(\eta)}(\boldsymbol{x}) = \mathbf{1}\{\Lambda_r(\boldsymbol{x}) \geq \tau_r(\eta)\} \in \{0, 1\}$, calibrated on non-members, but defined here specifically only on members for to reflect this analysis.)

So the define the **seed-wise true positive count** and **seed-wise true** TPR are

$$\text{TP}_r = \sum_{\boldsymbol{x} \in \mathbb{D}_{\text{IN}}} Y_r(\boldsymbol{x}), \qquad \text{TPR}_r = \frac{\text{TP}_r}{M}.$$

**Flip bins.** Using the unbiased flip U-statistic $\widehat{\text{flip}}_{\eta,B}(\boldsymbol{x})$ (Equation 9, computed once from all $B$ seeds), we partition members into disjoint flip bins $\{\mathbb{B}_j\}_j$, e.g., $[0, 0.1)$, $[0.1, 0.25)$, $[0.25, 0.4)$, $[0.4, t_\alpha(B))$, and $[t_\alpha(B), \widehat{\text{flip}}_{\eta,B}^{(\max)}]$ with $t_\alpha(B)$ from Appendix E.2.4. For example, $t_{0.05}(127) \approx 0.487$ and $t_{0.05}(125) \approx 0.490$.

For bin $j$ and seed $r$ we define the **bin TP count** and its **TPR mass**:

$$\text{TP}_{r,j} = \sum_{\boldsymbol{x} \in \mathbb{B}_j} Y_r(\boldsymbol{x}), \qquad \text{TPR}_{r,j} = \frac{\text{TP}_{r,j}}{M}.$$

By construction, $\text{TPR}_r = \sum_j \text{TPR}_{r,j}$ for every seed $r$.

**Across-seed means and standard deviations.** All means and STDs are computed across seeds. (We report STD across seeds, i.e., with denominator $B$, to match the rest of the paper.) For totals,

$$\overline{\text{TPR}} = \frac{1}{B}\sum_{r=1}^{B} \text{TPR}_r, \quad \text{STD(TPR)} = \sqrt{\frac{1}{B}\sum_{r=1}^{B}(\text{TPR}_r - \overline{\text{TPR}})^2},$$

and the corresponding counts follow from the variance scaling law ($\text{Var}[aX] = a^2\text{Var}[X]$):

$$\overline{\#\text{TP}} = M\,\overline{\text{TPR}}, \qquad \text{STD}(\#\text{TP}) = M\,\text{STD(TPR)}.$$

For each bin $j$,

$$\overline{\mathrm{TPR}_j} = \frac{1}{B}\sum_r \mathrm{TPR}_{r,j}, \qquad \mathrm{STD}(\mathrm{TPR}_j) = \sqrt{\frac{1}{B}\sum_r\left(\mathrm{TPR}_{r,j} - \overline{\mathrm{TPR}_j}\right)^2},$$

and analogously for bin TP counts: $\overline{\#\mathrm{TP}_j} = M\,\overline{\mathrm{TPR}_j}$ and $\mathrm{STD}(\#\mathrm{TP}_j) = M\,\mathrm{STD}(\mathrm{TPR}_j)$.

**Share of TPR from a bin: mean-of-ratios.** For seed $r$, define the **per-seed share** of TPR attributable to bin $j$:

$$S_{r,j} \;=\; \frac{\mathrm{TP}_{r,j}}{\mathrm{TP}_r} \quad (\text{set } S_{r,j}{=}0 \text{ if } \mathrm{TP}_r{=}0).$$

In our Tables 5 and 6, we report the across-seed mean and STD of these shares:

$$\overline{S}_j = \frac{1}{B}\sum_r S_{r,j}, \qquad \mathrm{STD}(S_j) = \sqrt{\frac{1}{B}\sum_r\left(S_{r,j} - \overline{S}_j\right)^2}.$$

This is a *mean of ratios*, i.e., the average fraction of each seed's TPR that comes from bin $j$.

**Alternative (ratio-of-means) and why it differs.** A different but also reasonable summary is the *ratio of means*,

$$R_j \;=\; \frac{\overline{\mathrm{TPR}_j}}{\overline{\mathrm{TPR}}} \;=\; \frac{\frac{1}{B}\sum_r \mathrm{TP}_{r,j}/M}{\frac{1}{B}\sum_r \mathrm{TP}_r/M} \;=\; \frac{\frac{1}{B}\sum_r \mathrm{TP}_{r,j}}{\frac{1}{B}\sum_r \mathrm{TP}_r}.$$

$R_j$ answers "what fraction of the *expected* true positives lie in bin $j$?," whereas $\overline{S}_j$ answers "for a *typical seed*, what fraction of that seed's true positives lie in bin $j$?" Because of seed-to-seed variability and the correlation between numerator and denominator, $R_j \neq \overline{S}_j$ in general. Our tables use $\overline{S}_j$ (mean of per-seed shares) and its STD across seeds, as we are trying to estimate the fraction of the average/typical seed's true positives that lie in each bin (in particular, the statistically indistinguishable-from-a-coin-flip bin).

**Combining two bins.** Let the two high-instability bins be $U{=}[0.4, t_\alpha(B))$ and $A{=}[t_\alpha(B), \widehat{\mathrm{flip}}_{\eta,B}^{(\max)}]$ and define the combined bin $C{=}U \cup A = [0.4, \widehat{\mathrm{flip}}_{\eta,B}^{(\max)}]$.
For each seed $r$,

$$S_{r,C} = S_{r,U} + S_{r,A}, \quad \mathrm{TP}_{r,C} = \mathrm{TP}_{r,U} + \mathrm{TP}_{r,A}.$$

Therefore, means add by linearity of expectation: $\overline{S}_C = \overline{S}_U + \overline{S}_A$ and $\overline{\#\mathrm{TP}_C} = \overline{\#\mathrm{TP}_U} + \overline{\#\mathrm{TP}_A}$. However, for standard deviations,

$$\mathrm{STD}(S_C) = \sqrt{\mathrm{Var}(S_U) + \mathrm{Var}(S_A) + 2\,\mathrm{Cov}(S_U, S_A)},$$

Since we compute $C$ directly from per-seed $C$ values, we can easily combine bins in a way that is statistically correct. We report this combined bin in our tables.

**What has no STD.** Bin membership counts $|\mathbb{B}_j|$ and their percentages $|\mathbb{B}_j|/M$ have no across-seed STD because bins are defined once from $\widehat{\mathrm{flip}}_{\eta,B}$ computed using all $B$ seeds. True positives, though, do have across-seed STD, as these are estimated for each target.

**Assumptions and caveats for interpreting these numbers**

1. **Single-target vs. many-seed view.** These numbers diagnose instability that is hidden to an attacker with access to only a single target. A sample counted as a "TP from coin-flip-like MIA decisions" is not "incorrect"—it is a member that this seed calls positive, but whose decision flips frequently across equally plausible seeds. Even though it is not incorrect, it does *not* reflect reliable knowledge about *inferring* membership for that sample, since it is effectively a coin-flip decision. We quantify how much of $\overline{\mathrm{TPR}}$ is borne by such decisions. However, any single target (that could reasonably be attacked by a real-world attacker) may deviate from this mean.

Table 5: **140M model: Contribution of high-flip members to TPR at fixed FPR.** Shares are computed per seed (TPs in bin divided by total TPs for that seed) and then averaged; their STDs are across seeds. We additionally report the combined bin $\widehat{\text{flip}}_{\eta,125} \in [0.4, 0.504]$ ("very unstable + coin flip"). For the 140M model, the coin-flip cutoff is $t_\alpha(125) \approx 0.490$, and $\widehat{\text{flip}}_{\eta,125}^{\max} = 0.504$. Typically reported $\log$-scale FPR rows are highlighted in gray.

| FPR | TPR (mean±STD) | TP (mean±STD) | Very unstable [0.4, 0.490) | | Coin flip [0.490, 0.504] | | Very unstable + coin flip [0.4, 0.504] | |
|---|---|---|---|---|---|---|---|---|
| | | | TPs (mean±STD) | Share of TPR (mean±STD) | TPs (mean±STD) | Share of TPR (mean±STD) | TPs (mean±STD) | TPR (mean±STD) |
| $10^{-5}$ | $0.002 \pm 0.001$ | $1{,}114 \pm 305$ | $61 \pm 9$ | $5.8\% \pm 1.4\%$ | $24 \pm 4$ | $2.3\% \pm 0.7\%$ | $84 \pm 11$ | $8.0\% \pm 2.0\%$ |
| $10^{-4}$ | $0.008 \pm 0.001$ | $4{,}247 \pm 563$ | $432 \pm 46$ | $10.2\% \pm 0.7\%$ | $160 \pm 13$ | $3.8\% \pm 0.4\%$ | $592 \pm 55$ | $14.1\% \pm 1.0\%$ |
| $10^{-3}$ | $0.030 \pm 0.002$ | $15{,}826 \pm 1{,}239$ | $3{,}048 \pm 304$ | $13.3\% \pm 0.4\%$ | $1{,}493 \pm 142$ | $6.8\% \pm 0.3\%$ | $4{,}541 \pm 413$ | $19.9\% \pm 0.8\%$ |
| $10^{-2}$ | $0.104 \pm 0.005$ | $54{,}373 \pm 2{,}414$ | $17{,}240 \pm 1{,}464$ | $31.2\% \pm 1.0\%$ | $12{,}164 \pm 1{,}070$ | $22.8\% \pm 1.0\%$ | $29{,}404 \pm 2{,}461$ | $54.0\% \pm 2.5\%$ |
| 0.02 | $0.148 \pm 0.005$ | $77{,}667 \pm 2{,}718$ | $27{,}455 \pm 2{,}031$ | $34.7\% \pm 0.8\%$ | $20{,}609 \pm 1{,}765$ | $27.1\% \pm 1.1\%$ | $48{,}064 \pm 3{,}698$ | $61.8\% \pm 3.0\%$ |
| 0.05 | $0.236 \pm 0.006$ | $123{,}864 \pm 2{,}978$ | $47{,}831 \pm 2{,}742$ | $38.7\% \pm 0.7\%$ | $37{,}910 \pm 3{,}223$ | $30.5\% \pm 1.1\%$ | $85{,}742 \pm 5{,}492$ | $69.2\% \pm 3.2\%$ |
| 0.1 | $0.336 \pm 0.006$ | $176{,}346 \pm 3{,}180$ | $70{,}670 \pm 3{,}050$ | $39.2\% \pm 0.6\%$ | $53{,}769 \pm 3{,}443$ | $31.4\% \pm 0.9\%$ | $124{,}439 \pm 6{,}231$ | $70.5\% \pm 2.7\%$ |
| 0.2 | $0.480 \pm 0.006$ | $251{,}888 \pm 3{,}256$ | $97{,}926 \pm 2{,}386$ | $39.1\% \pm 0.3\%$ | $67{,}384 \pm 1{,}688$ | $26.4\% \pm 0.3\%$ | $165{,}310 \pm 4{,}489$ | $65.6\% \pm 1.2\%$ |
| 0.5 | $0.766 \pm 0.004$ | $401{,}354 \pm 2{,}242$ | $74{,}851 \pm 610$ | $18.7\% \pm 0.2\%$ | $28{,}804 \pm 485$ | $7.2\% \pm 0.1\%$ | $103{,}655 \pm 807$ | $25.8\% \pm 0.3\%$ |
| 0.75 | $0.915 \pm 0.002$ | $479{,}540 \pm 882$ | $14{,}674 \pm 339$ | $3.1\% \pm 0.1\%$ | $2{,}644 \pm 75$ | $0.6\% \pm 0.0\%$ | $17{,}318 \pm 403$ | $3.6\% \pm 0.1\%$ |
| $10^0$ | $1.000 \pm 0.000$ | $524{,}288 \pm 0$ | $0 \pm 0$ | $0.0\% \pm 0.0\%$ | $0 \pm 0$ | $0.0\% \pm 0.0\%$ | $0 \pm 0$ | $0.0\% \pm 0.0\%$ |

2. **Calibration asymmetry.** Again, we note that thresholds $\tau_r(\eta)$ are calibrated on non-members for each seed, anchoring to non-member behavior by construction. Member decisions are therefore more exposed to seed-induced variation. This explains large member/non-member flip gaps and is consistent with our seed-to-seed $\tau$ dispersion at higher $\eta$. The true positive decomposition analysis we perform here is consistent with these other results.

3. **Finite-$B$ effects and hypothesis testing.** The acceptance region cutoff $t_\alpha(B)$ is derived from an exact two-sided binomial test at level $\alpha$ and is slightly conservative because of discreteness. Borderline cases (exactly at the tails) are included (fail-to-reject rule $p \geq \alpha$). This is important because all claims that we make about MIA decisions that are indistinguishable from a coin flip— including the decompositions here—hinge on the assumptions and results of this hypothesis test.

4. **Extremely low FPR $\eta$.** When $\overline{\text{TPR}}$ is very small, the share $S_{r,j}$ can be numerically unstable; we suppress shares when $\overline{\text{TPR}} = 0$ and note this where appropriate.

**Results of the decomposition.** We show results for both model sizes in Tables 5 and 6, respectively. They indicate very large numbers of member decisions are indistinguishable from a coin flip or highly unstable as FPR increases and moves the decision boundary into denser parts of the score distribution. Even at just FPR=$10^{-3}$, $15.4\% \pm 0.6\%$ of true positives for the 302M model are indistinguishable from a coin flip—reflecting thousands of sample MIA decisions. If we consider both very unstable and coin-flip-like MIA decisions, they are responsible for $42.2\% \pm 0.9\%$ of true positives at this FPR.

**Estimating the contribution of coin-flip-like MIA decisions to ROC-AUC.** We similarly can use this type of analysis to estimate how much of ROC-AUC can be attributed to MIA decisions that are indistinguishable from a coin flip. Similarly, the point of this analysis is not to suggest that these MIA decisions are "incorrect;" the point is to attempt to distinguish the degree to which coin-flip-like MIA decisions are impacting overall claims about successful membership *inference*. To do so, we distinguish between the whole (averaged) ROC curve and associated mean AUC, and the mean ROC curve (and mean AUC) computed for non-coin-flip-like decisions.

For a ROC curve written as TPR vs. FPR, fix an FPR interval $[a, b] \subset (0, 1)$ and write $w = b - a$. For a given target $r$, denote $\text{TPR}_r(\eta)$ as the true positive rate at operating point FPR=$\eta$, and let $\overline{\text{TPR}}(\eta)$ be the mean across seeds.

We distinguish positives that come from coin-flip-like decisions as follows: at each FPR, we remove the portion of the $\overline{\text{TPR}}$ that is supported by decisions that are indistinguishable from a coin flip. That is, fix $B$ targets and significance $\alpha$; let $t_\alpha(B)$ be the exact two–sided Binomial$(B, 0.5)$

Table 6: **302M model: Contribution of high-flip members to TPR at fixed FPR.** Shares are computed per seed (TPs in bin divided by total TPs for that seed) and then averaged; their STDs are across seeds. We additionally report the combined bin $\widehat{\mathrm{flip}}_{\eta,127} \in [0.4, 0.50394]$ ("very unstable + coin flip"). For the 302M model, the coin-flip cutoff is $t_\alpha(127)\approx0.487$, and $\widehat{\mathrm{flip}}_{\eta,127}^{\max}=0.50394$. Typically reported $\log$-scale FPR rows are highlighted in gray.

| FPR | TPR (mean±STD) | TP (mean±STD) | Very unstable [0.4, 0.487) | | Coin flip [0.487, 0.50394] | | Very unstable + coin flip [0.4, 0.50394] | |
| --- | --- | --- | --- | --- | --- | --- | --- | --- |
| | | | TPs (mean±STD) | Share of TPR (mean±STD) | TPs (mean±STD) | Share of TPR (mean±STD) | TPs (mean±STD) | TPR (mean±STD) |
| $10^{-5}$ | $0.009 \pm 0.003$ | $4{,}878 \pm 1{,}496$ | $371 \pm 66$ | $8.1\% \pm 2.1\%$ | $122 \pm 16$ | $2.8\% \pm 1.5\%$ | $493 \pm 80$ | $10.9\% \pm 3.6\%$ |
| $10^{-4}$ | $0.027 \pm 0.005$ | $14{,}254 \pm 2{,}631$ | $2{,}233 \pm 349$ | $15.8\% \pm 1.0\%$ | $805 \pm 87$ | $5.8\% \pm 0.9\%$ | $3{,}038 \pm 432$ | $21.6\% \pm 1.8\%$ |
| $10^{-3}$ | $0.073 \pm 0.009$ | $38{,}414 \pm 4{,}886$ | $10{,}295 \pm 1{,}548$ | $26.9\% \pm 0.9\%$ | $5{,}883 \pm 651$ | $15.4\% \pm 0.6\%$ | $16{,}231 \pm 2{,}194$ | $42.2\% \pm 0.9\%$ |
| $10^{-2}$ | $0.181 \pm 0.013$ | $94{,}823 \pm 6{,}939$ | $32{,}549 \pm 3{,}549$ | $34.3\% \pm 1.5\%$ | $31{,}866 \pm 3{,}429$ | $33.5\% \pm 1.5\%$ | $64{,}477 \pm 6{,}973$ | $67.8\% \pm 3.0\%$ |
| 0.02 | $0.235 \pm 0.014$ | $123{,}384 \pm 7{,}183$ | $44{,}612 \pm 3{,}894$ | $36.1\% \pm 1.3\%$ | $47{,}301 \pm 4{,}805$ | $38.2\% \pm 2.0\%$ | $91{,}913 \pm 8{,}687$ | $74.3\% \pm 3.3\%$ |
| 0.05 | $0.333 \pm 0.014$ | $174{,}687 \pm 7{,}312$ | $69{,}431 \pm 4{,}149$ | $39.7\% \pm 0.9\%$ | $71{,}632 \pm 6{,}160$ | $40.9\% \pm 2.2\%$ | $141{,}062 \pm 10{,}270$ | $80.6\% \pm 3.1\%$ |
| 0.1 | $0.435 \pm 0.014$ | $228{,}106 \pm 7{,}209$ | $98{,}285 \pm 4{,}353$ | $42.9\% \pm 0.6\%$ | $88{,}061 \pm 5{,}522$ | $38.6\% \pm 1.4\%$ | $186{,}021 \pm 9{,}760$ | $81.5\% \pm 2.0\%$ |
| 0.2 | $0.570 \pm 0.013$ | $299{,}000 \pm 6{,}649$ | $128{,}904 \pm 3{,}649$ | $43.1\% \pm 0.3\%$ | $87{,}119 \pm 2{,}415$ | $29.1\% \pm 0.2\%$ | $216{,}023 \pm 5{,}982$ | $72.2\% \pm 0.4\%$ |
| 0.5 | $0.809 \pm 0.008$ | $424{,}228 \pm 3{,}952$ | $66{,}051 \pm 1{,}888$ | $15.6\% \pm 0.4\%$ | $12{,}913 \pm 241$ | $3.0\% \pm 0.1\%$ | $78{,}964 \pm 2{,}093$ | $18.6\% \pm 0.4\%$ |
| 0.75 | $0.925 \pm 0.004$ | $484{,}974 \pm 1{,}908$ | $9{,}094 \pm 372$ | $1.9\% \pm 0.1\%$ | $613 \pm 28$ | $0.1\% \pm 0.0\%$ | $9{,}708 \pm 394$ | $2.0\% \pm 0.1\%$ |
| $10^{0}$ | $1.000 \pm 0.000$ | $524{,}288 \pm 0$ | $0 \pm 0$ | $0.0\% \pm 0.0\%$ | $0 \pm 0$ | $0.0\% \pm 0.0\%$ | $0 \pm 0$ | $0.0\% \pm 0.0\%$ |

acceptance–band flip cutoff (Appendix E.2.4). Let $M = |\mathbb{D}_{\mathrm{IN}}|$ denote the number of members. For seed $r$ at FPR $\eta$, write $Y_r(\boldsymbol{x}) \in \{0, 1\}$ for the membership decision on member $\boldsymbol{x}$, and let

$$\mathrm{TPR}_{\mathrm{raw},r}(\eta) \;=\; \frac{1}{M} \sum_{\boldsymbol{x} \in \mathbb{D}_{\mathrm{IN}}} Y_r(\boldsymbol{x}), \qquad \mathrm{TPR}_{\mathrm{coin},r}(\eta) \;=\; \frac{1}{M} \sum_{\boldsymbol{x} \in \mathbb{D}_{\mathrm{IN}}} Y_r(\boldsymbol{x}) \, \mathbf{1}\{\widehat{\mathrm{flip}}_{\eta,B}(\boldsymbol{x}) \geq t_\alpha(B)\}.$$

We define the non-coin-flip-like-induced TPR per seed by subtraction,

$$\mathrm{TPR}_{\mathrm{noncoin},r}(\eta) \;=\; \mathrm{TPR}_{\mathrm{raw},r}(\eta) - \mathrm{TPR}_{\mathrm{coin},r}(\eta).$$

We compute the averages across $B$ seeds as

$$\overline{\mathrm{TPR}}_{\mathrm{raw}}(\eta) \;=\; \frac{1}{B} \sum_{r=1}^{B} \mathrm{TPR}_{\mathrm{raw},r}(\eta), \qquad \overline{\mathrm{TPR}}_{\mathrm{noncoin}}(\eta) \;=\; \frac{1}{B} \sum_{r=1}^{B} \mathrm{TPR}_{\mathrm{noncoin},r}(\eta).$$

By linearity of expectation, across seeds,

$$\overline{\mathrm{TPR}}_{\mathrm{noncoin}}(\eta) \;=\; \overline{\mathrm{TPR}}_{\mathrm{raw}}(\eta) - \overline{\mathrm{TPR}}_{\mathrm{coin}}(\eta). \tag{24}$$

Note that $\overline{\mathrm{TPR}}_{\mathrm{noncoin}}$ is a diagnostic curve: it subtracts the statistically indistinguishable-from-a-coin-flip component and is *not* itself the ROC of a single threshold rule that could be produced by an attack.

We can then compare the overall ROC to the non-coin-flip-like ROC. It may be useful to do so for a specific range of FPR, rather than for the entire ROC curve. To do so, note that the unnormalized partial AUC (pAUC) over $[a, b]$ is

$$\mathrm{pAUC}[a, b] \;:=\; \int_a^b \overline{\mathrm{TPR}}(\eta) \, d\eta.$$

The normalized pAUC is the mean TPR averaged over FPR in $[a, b]$; it rescales to $[0, 1]$ by dividing by the band width:

$$\mathrm{pAUC}_{\mathrm{norm}}[a, b] \;:=\; \frac{1}{b - a} \int_a^b \overline{\mathrm{TPR}}(\eta) \, d\eta, \quad \text{which equals the mean TPR over the band.}$$

To quantify improvement over a random classifier, we subtract the area under the by-chance curve. For the random classifier $\overline{\mathrm{TPR}}(\eta) = \mathrm{FPR}(\eta)$, so

$$\int_a^b \eta \, d\eta \;=\; \frac{\eta^2}{2} \Big|_{\eta=a}^{b} \;=\; \frac{b^2 - a^2}{2}.$$

Dividing by $w = b - a$ gives the normalized random baseline $\frac{1}{b-a} \cdot \frac{b^2 - a^2}{2} = \frac{a+b}{2}$. We can similarly report the **lift** above these random baselines:

$$\text{Lift}[a,b] \coloneqq \int_a^b \left(\overline{\text{TPR}}(\eta) - \eta\right) d\eta = \text{pAUC}[a,b] - \frac{b^2 - a^2}{2},$$

$$\text{Lift}_{\text{norm}}[a,b] \coloneqq \frac{1}{b-a} \int_a^b \left(\overline{\text{TPR}}(\eta) - \eta\right) d\eta = \text{pAUC}_{\text{norm}}[a,b] - \frac{a+b}{2}.$$

Note that $\text{Lift}_{\text{norm}}[a,b] = 0$ for a random classifier and equals the average vertical gap from chance in $[a,b]$.

**Worked example.** We apply this procedure to the ROC for the 302M model. We do so for the entire curve in Figure 30b, which shows that $\approx 0.059$ of ROC-AUC can be attributed to coin-flip-like MIA decisions. Of course, the region where this has the most impact is the range of $\eta$ that moves the decision boundary into the denser part of the score distribution where MIA calls more positives, but IN/OUT distribution overlap is more extensive. TPR rises, but so does the share of coin-flip-like true positives.

We can therefore also isolate the effect on this part of ROC-AUC by setting $[a,b] = [10^{-4}, 10^{-1}]$. Here $w = 0.0999$, the normalized random baseline is $\frac{a+b}{2} = 0.05005$, and the unnormalized random area is $\frac{b^2 - a^2}{2} = 0.004999995$. With our measured areas,

$$\text{pAUC}_{\text{norm}}^{\text{raw}} = 0.314748, \quad \text{pAUC}_{\text{norm}}^{\text{noncoin}} = 0.190810,$$

so

$$\text{Lift}_{\text{norm}}^{\text{raw}} = 0.264698, \qquad \text{Lift}_{\text{norm}}^{\text{noncoin}} = 0.140760.$$

As a result, the non-coin-flip-like ROC retains about $\frac{0.140760}{0.264698} \times 100 \approx 53\%$ of the raw lift in this FPR band, and the average TPR for the band drops by $1 - \frac{0.190810}{0.314748} \times 100 \approx 39.4\%$ relative to raw. In other words, filtering out coin-flip-like positives yields a substantially lower pAUC in this band. A sizable part of the apparent attack advantage in this FPR range comes from samples whose per-seed decisions are statistically indistinguishable from coin flips (Appendix E.2.4). This weakens the value/reliability of per-sample positives in this FPR regime.

It is important to note that this is a conservative estimate of the parts of the attack that are reliable, as we only filter out positives that pass the coin-flip-like flip threshold, $t_\alpha(B)$. As a result, these numbers still include highly unstable cases (that are arguably also not reflective of meaningfully reliable membership inference, e.g., flip in $[0.4, t_\alpha(B))$). In this respect, our non-coin-flip-like ROC is a conservative upper bound on reliable membership inference. Given the extent of highly unstable MIA decisions in this band, we would expect larger decreases in partial AUC and Lift if we filtered those decisions in our analysis.

**Caveats.** Because bins are defined using $\widehat{\text{flip}}_{\eta,B}$ pooled over $B$ seeds, per-seed variability in bin membership is not propagated. This makes our estimates conservative for variability, but keeps the decomposition identity exact (i.e., the sum of the bin masses equals $\overline{\text{TPR}}$). In our implementation, we also clip $\text{TPR}_{\text{noncoin},r}$ at 0 to guard against finite-sample noise, i.e., $\text{TPR}_{\text{noncoin},r}(\eta) \leftarrow \max\{0, \text{TPR}_{\text{raw},r}(\eta) - \text{TPR}_{\text{coin},r}(\eta)\}$.

### E.2.7 Additional notes on interpreting flip rate measurements

Strong membership inference claims are *conditional on a training pipeline*: given a fixed recipe (model class, optimizer, etc.), does the model's behavior on $x$ indicate that $x$ was in the training set? Strong MIAs such as LiRA estimate the sample $x$'s score given that $x \in \text{IN}$ vs. the score given that $x \in \text{OUT}$, computed from many reference models by varying the data, i.e., they probe *data counterfactuals* under the same pipeline.

Our flip rate analysis probes the complementary axis: *model counterfactuals*. We draw equally plausible targets $r \sim \mu$ from the same pipeline using the same training dataset (changing only the seed that controls batch order), hold the set of references and calibration procedure fixed (i.e., thresholds calibrated per target on non-members), and ask whether the binary decision for a sample $x$ is stable across those targets. References address *data* variability; target flip rate quantifies *model* variability. From a statistical reliability perspective, per-sample inference should be robust with respect to both.

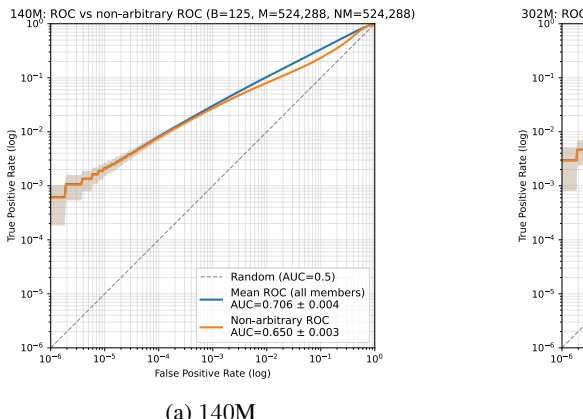
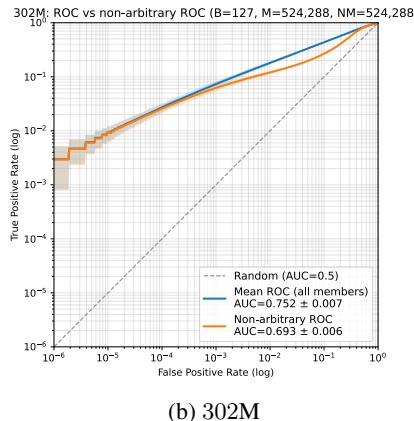

(a) 140M                                    (b) 302M

Figure 30: **Decoupling overall attack success from success based on coin-flip like sample decisions.** We produce the same mean ROC curves and mean AUC as in Figure 24 for both the (**a**) 140M and (**b**) 302M models ($B{=}125$ and $B{=}127$, respectively). At each TPR for fixed FPR, we estimate how many true positives are attributable to MIA decisions that are statistically indistinguishable from a coin flip ($t_\alpha(125){\approx}0.490$ and $t_\alpha(127){\approx}0.487$, respectively; see Appendix E.2.4). By Equation 24, at each $\eta$, we can then estimate how many true positives are from non-coin-flip-like decisions. Consistent with our other results, As both curves enter ranges for FPR with nontrivial TPR, coin-flip-like MIA decisions make up a significant proportion of ROC-AUC, with a greater effect for the 302M architecture.

**Descriptive vs. inferential claims.** Put another way, both axes may matter depending on the type of claim we want to make. We can make a *descriptive* claim [9]: "On this specific model, we called $x$ a 'member' and were right." In contrast, we can also make an *inferential* claim: "If we retrained the same pipeline, we would still call $x$ a 'member' most of the time." Flip rate distinguishes between these. In the absence of flip rate analysis, we can make the first type of claim relatively easily, as in the prior work in the literature [e.g., 5]. But without measuring something like flip rate, we cannot assess the second type of claim. When we do measure flip rate, observing high values (i.e., $\approx 0.5$) means that the second claim fails: a positive call on $x$ carries about the same evidential weight as a coin toss; it may be correct, but not because of stable signal. This type of evidential weight is weak because this call is not reproducible—even if it is correct on the particular target that the attacker happens to be attacking. (Conversely, think of a sample $x$ that has low flip rate: the attack is able to produce reliable signal about membership for that $x$.)

**Why ML security should care about inferential (not just descriptive) claims.** On a first read of our flip rate results, a security-focused reader may not necessarily be interested in making an inferential claim. An attacker only gets one model, and so one might think, "our positive call was correct: why care about reliability with respect to the training pipeline that produced that one model?" From this perspective, the descriptive claim suffices.

However, in machine learning, focusing only on descriptive claims is arguably insufficient; for a statistical procedure (like a strong MIA), the standard for success is higher. That is, to develop *scientific knowledge*, we require the *entire attack procedure* to be reliable for making claims about $x$. From this perspective, a statistically coin-flip-like true positive is a lucky hit, not reliable evidence produced by this specific attack procedure. (Again, conversely, a stable true positive is reliable evidence produced by this procedure.)

This is why we perform flip rate analysis: it is a useful diagnostic for making scientifically sound claims about the entire attack procedure, which is precisely what we are trying to address in this paper: the extent to which strong MIAs (as an attack procedure) can succeed on LLMs. For an attacker that attacks a single target, traditional MIA metrics (TPR at fixed FPR, ROC-AUC) are still useful; but our results surface what fraction of decisions are fragile. For reliable scientific knowledge, if that fraction is large, then claims about the MIA procedure's effectiveness should be couched as descriptive, not inferential.

### E.2.8 An alternative instability diagnostic: fixing the target, resampling reference sets

The instability analysis in the main paper and the rest of this appendix (Section 5 & Appendices E.2.1–E.2.7) fixes the IN/OUT reference set and varies the target seed, probing *model counterfactuals*. A natural complementary diagnostic, closer in spirit to the typical threat model, fixes the target and varies the IN/OUT references, probing *reference counterfactuals*. Here we formalize that diagnostic and explain why we did not run it at scale.

**Setup and notation.** Fix a trained target $h$ on dataset $\mathbb{D}$. Let $\mathbb{S}^{(1)}, \dots, \mathbb{S}^{(B)}$ be $B$ independent reference sets; each set $\mathbb{S}^{(j)}$ contains $|\Phi_{\text{IN}}| = R_{\text{IN}}$ IN references and $|\Phi_{\text{OUT}}| = R_{\text{OUT}}$ OUT references (in our experiments, $R_{\text{IN}} + R_{\text{OUT}} = 128$). Given $\mathbb{S}^{(j)}$, define the LiRA score for a sample $\boldsymbol{x}$ as $\Lambda_h^{(j)}(\boldsymbol{x}) \in \mathbb{R}$, computed exactly as in the main setup (Appendix A) but using only the models in $\mathbb{S}^{(j)}$ as the references.

**Calibration to non-members at fixed** FPR **(per reference set).** Let the non-member dataset be $\mathbb{D}_{\text{OUT}}$ with size $N_{\text{OUT}}$. Form the empirical CDF of OUT scores using reference set $j$:

$$\widehat{F}_{\text{OUT}}^{(j)}(t) = \frac{1}{N_{\text{OUT}}} \sum_{\boldsymbol{x} \in \mathbb{D}_{\text{OUT}}} \mathbf{1}\{\Lambda_h^{(j)}(\boldsymbol{x}) \le t\}.$$

For each reference set $j$ and FPR $\eta$, we calibrate a decision rule for the single target $h$, choosing the right-open (left-limit CDF) cutoff

$$\tau_h^{(j)}(\eta) = \inf\{t : \widehat{F}_{\text{OUT}}^{(j)}(t^-) \ge 1 - \eta\}, \qquad b_h^{(\eta,j)}(\boldsymbol{x}) = \mathbf{1}\{\Lambda_h^{(j)}(\boldsymbol{x}) \ge \tau_h^{(j)}(\eta)\}.$$

Then the realized FPR on the non-member (OUT) dataset for the target $h$ using references $j$ satisfies

$$\widehat{\text{FPR}}^{(j)}(\eta) = \frac{1}{N_{\text{OUT}}} \sum_{\boldsymbol{x} \in \mathbb{D}_{\text{OUT}}} \mathbf{1}\{\Lambda_h^{(j)}(\boldsymbol{x}) \ge \tau_h^{(j)}(\eta)\} = 1 - \widehat{F}_{\text{OUT}}^{(j)}\big(\tau_h^{(j)}(\eta)^-\big) \le \eta,$$

i.e., finite-sample, conservative with ties. This mirrors how an attacker would calibrate a threshold for the target using the specific reference set they happened to train. (In this case, we train $B$ such equally plausible reference sets.)

**Reference-resampling flip rate.** Let $J$ denote an index drawn uniformly from $\{1, \dots, B\}$, representing an i.i.d. draw of a reference set. For this reference-resampling setup, define the population flip rate across possible reference sets for the fixed target $h$ as

$$\text{flip}_\eta^{\text{ref}}(\boldsymbol{x}) := \Pr_{J, J' \overset{\text{i.i.d.}}{\sim} \{1, \dots, B\}} \big[b_h^{(\eta,J)}(\boldsymbol{x}) \ne b_h^{(\eta,J')}(\boldsymbol{x})\big] = 2\,\theta^{\text{ref}}\big(1 - \theta^{\text{ref}}\big),$$

where $\theta^{\text{ref}} := \Pr_J \big[b_h^{(\eta,J)}(\boldsymbol{x}) = 1\big]$. Given $B \ge 2$ reference sets, the unbiased estimator is

$$\widehat{\text{flip}}_{\eta,B}^{\text{ref}}(\boldsymbol{x}) = \binom{B}{2}^{-1} \sum_{1 \le j < \ell \le B} \mathbf{1}\big\{b_h^{(\eta,j)}(\boldsymbol{x}) \ne b_h^{(\eta,\ell)}(\boldsymbol{x})\big\} = \frac{2\,B_0^{\text{ref}}(\boldsymbol{x})\,B_1^{\text{ref}}(\boldsymbol{x})}{B\,(B-1)},$$

with $B_1^{\text{ref}}(\boldsymbol{x}) = \sum_{j=1}^B b_h^{(\eta,j)}(\boldsymbol{x})$ and $B_0^{\text{ref}}(\boldsymbol{x}) = B - B_1^{\text{ref}}(\boldsymbol{x})$. Just as in our target-resampling analysis, we label a decision "coin-flip-like at level $\alpha$" by testing $H_0 : \theta^{\text{ref}} = 0.5$ with the exact two-sided binomial test on $B$ votes; equivalently, $\widehat{\text{flip}}_{\eta,B}^{\text{ref}}(\boldsymbol{x}) \ge t_\alpha(B)$, where $t_\alpha(B) = \frac{2\,k_L(B - k_L)}{B(B-1)}$ and $k_L$ is the equal-tail cutoff (Appendix E.2.4).

**Cost and feasibility.** In the main paper target-resampling flip analyis, we trained one reference set of size $R = 128$ and $B_{\text{target}} = 127$ targets ($R + B_{\text{target}} = 255$ total trained models for the 302M setting). For this reference-resampling diagnostic, to obtain comparable statistical fidelity, one would fix the target and train $B_{\text{ref}}$ independent reference sets, each of size $R$, i.e., train $R \times B_{\text{ref}}$ reference models. Taking $B_{\text{ref}} \approx 128$ yields $128 \times 128 = 16{,}384$ references (plus one target), requiring about $4\times$ more models than all of the other experiments in this paper combined. To approximate this diagnostic, we could perhaps subsample smaller reference sets from a single larger pool (e.g., subsample the 128 references from the target flip rate experiments). However, such subsampling would introduce dependence across reference sets, violating the i.i.d. assumption and complicating the exact binomial test we use to determine the coin-flip cutoff.

**Why we expect qualitatively similar instability to our target-resampling experiments.** Empirically, our cross-target flip rate analysis shows that many members have small margins relative to the calibrated threshold: for these $\boldsymbol{x}$, $\Lambda_r(\boldsymbol{x}) - \tau_r(\eta)$ is frequently near 0 (Appendix E.2.5).

For the reference-resampling diagnostic, write the per-reference-set margin for the fixed target $h$ as

$$\Delta^{(j)}(\boldsymbol{x}) \;:=\; \Lambda_h^{(j)}(\boldsymbol{x}) - \tau_h^{(j)}(\eta)$$

$$= \underbrace{m_h(\boldsymbol{x})}_{\text{target + data fixed}} + \underbrace{\varepsilon_\Lambda^{(j)}(\boldsymbol{x})}_{\text{score variability from ref. sets}} - \underbrace{\varepsilon_\tau^{(j)}}_{\text{quantile variability from per-set calibration}},$$

where $\varepsilon_\Lambda^{(j)}$ captures sampling variability from fitting IN/OUT reference statistics and $\varepsilon_\tau^{(j)}$ captures quantile-estimation variability in $\tau_h^{(j)}(\eta)$. Our cross-target results show many members with margins near 0 (poor separability of reference distributions and proximity to the threshold). For such samples $\boldsymbol{x}$, any nonzero variability in $\varepsilon_\Lambda^{(j)}(\boldsymbol{x}) - \varepsilon_\tau^{(j)}$ will, with nontrivial probability, change the sign of $\Delta^{(j)}(\boldsymbol{x})$ and thus flip the decision. And so, given the sensitivity that we observe when training counterfactuals, we expect this reference-resampling diagnostic to surface qualitatively similar per-sample instability as the target-resampling experiments we run—especially for members—without running the much more expensive reference-resampling procedure.

## F  Additional per-sample MIA vulnerability results

Figure 6b indicates that it is often the case that vulnerable sequences tend to be longer. Beyond sequence length, we observe that samples more vulnerable to MIA tend to have higher mean TF-IDF scores (Figure 31a), suggesting that texts with distinctive, uncommon terms may exhibit stronger signals for membership inference. We compute these TF-IDF scores without normalization, collecting document frequency statistics over a random subsample of the original dataset and then taking the mean across all tokens in each sample. Similarly, samples containing unknown tokens (`<unk>`) appear more vulnerable to MIA (Figure 31b).

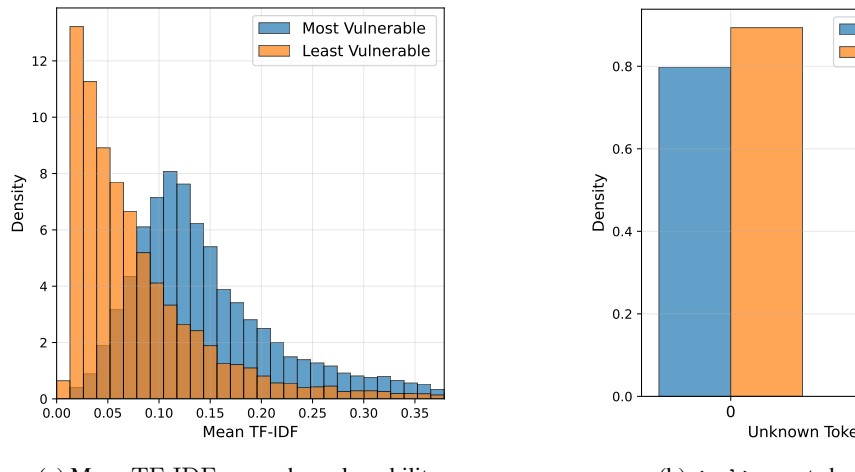

(a) Mean TF-IDF scores by vulnerability      (b) `<unk>` counts by vulnerability

Figure 31: **Text property distributions by MIA vulnerability.** The most vulnerable samples tend to (**a**) have higher TF-IDF scores compared to least vulnerable samples, and (**b**) are more likely to contain at least one unknown token (`<unk>`).

### F.1  Does memorization imply strong membership inference attacks?

While memorization is a key factor that can make a model susceptible to membership inference attacks, it does not automatically guarantee that strong MIAs will always be successful. Memorization refers to a model learning specific details about its training data, rather than just general patterns.

When a model heavily memorizes training samples, it often exhibits distinct behaviours for these samples, which MIA attackers, in principle, can exploit. Indeed, studies have shown that the risk of

membership inference is often highest for those samples that are highly memorized [4]. However, our results show that the practical success and strength of a particular MIA can also depend on other factors, such as the model architecture, the type of data, the specifics of the attack method, and whether the memorization leads to clearly distinguishable outputs or behaviors for member versus non-member samples. Some models might memorize data in ways that are not easily exploitable by current MIA techniques, or the signals of memorization might be subtle for well-generalizing models, making strong attacks more challenging despite the presence of memorization.

### F.2 Evolution of losses over different model sizes

In Figure 32, for three samples, we plot loss (target) and the per-sample reference distributions $p_{\mathrm{IN}}(\boldsymbol{x})$ and $p_{\mathrm{OUT}}(\boldsymbol{x})$ over different model sizes. Each of these models is trained for 1 epoch on $2^{23} \approx 8.3\mathrm{M}$ samples. This is a sanity check that the losses decrease (on the same sample) as the model size increases. Note that, for these samples, the distance between member and non-member reference distributions does not significantly shift as the model size grows.

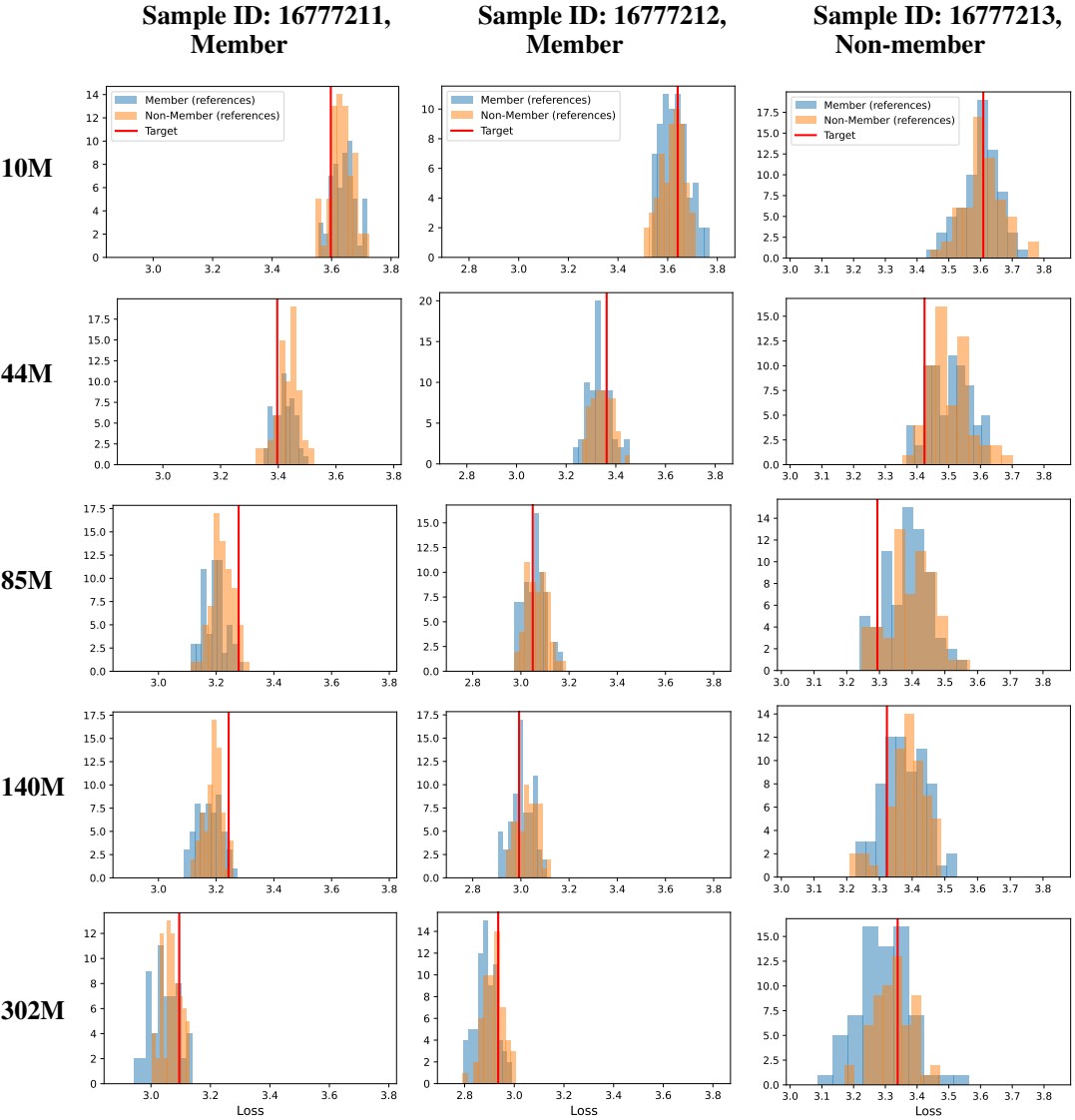

Figure 32: **Target loss and reference loss distributions for three samples.** For three different samples (referenced by their IDs in the C4 dataset), we plot the reference distributions and the loss of the sample for the target model (as a vertical red line). Each row shows results for a different model size.

# G  Experiment configuration details

In Table 7, we provide pecific experimental settings. Unless otherwise stated, we used the AdamW optimizer [32] with a cosine scheduler. The initial learning rate is set to $10^{-7}$ and increases linearly over 750 warm up steps to a peak learning rate of $3 \cdot 10^{-4}$, after which it decreases according to the cosine schedule to a final value of $3 \cdot 10^{-5}$. We typically use 128 reference models and a single target model to measure MIA vulnerability, drawing a dataset of size $2N$ from C4 from which we subsample training datasets of size $N$. For each reference and target model, the training set is subsampled from the same larger dataset of size $2N$. This means each sample in this larger dataset is a member for $\approx$64 reference models. The batch size is fixed to 128 and sequence length to 1024; if an sample has fewer tokens, we pad to 1024. The weight decay is set to 0.1, and a global clipping norm is set to 1.0. Note that we can approximately convert the training set size to total number of training tokens by multiplying the training set size by 400, as this the approximate average number of tokens within a C4 sample. For example, this means the 1018M model was trained on 20.4B tokens in Figure 2.

Table 7: **Experimental details.** Experiment (figure), training set size (approximate number of samples), model size, and specific details that diverge from default settings.

| Experiment | Training set size | Model size | Other information (which diverges from default experimental settings) |
|---|---|---|---|
| Figure 1 | 7M | 140M | Max. 512 references |
| Figures 2, 15a | 500K
2.2M
4.25M
7M
15.1M
24.4M
30.2M
50.9M | 10M
44M
85M
140M
302M
489M
604M
1018M | 128 references |
| Figure 3a | 2.2M
1.1M | 44M
44M | 2 different variations; 1 epoch and 2 epochs (on the same 2.2M samples, but split in different ways across epochs) |
| Figures 3b, 17a | 7M | 140M | 10 epochs |
| Figure 4a | 50K
100K
500K
1M
5M
10M | 140M
140M
140M
140M
140M
140M | 80 warm up steps |
| Figure 4b | $2^{23}$ | 10M
44M
85M
140M
302M
425M
489M
509M
604M
1018M | |
| Figures 5, 23, 24b, 25, 27, 29, 30b | 7M | 302M | 127 target models (varying only in random seed; same training data); 128 references |
| Figures 6, 7, 8 | 7M | 140M | 128 references |
| Figure 9a | 7M | 140M | Max. 256 references |
| Figure 9b | 7M | 140M | Max. 64 references, 10K $\mathbb{Z}$ population size |
| Figure 10 | 7M | 140M | 256 references |
| Figure 11 | 500K | 10M | 10K $\mathbb{Z}$ population size |
| Figures 12, 13 | 500K | 10M | 10K-300K $\mathbb{Z}$ population size; 64 references |
| Figure 14 | $2^{19}$ | 10M | up to 128 references (testing online and offline variants) |
| Figure 15b | 50K | 140M | Learning rate schedules: cosine, cosine with 0 weight decay, cosine with no clipping, linear. We use 50 warm up steps. |
| Figure 16 | 7M | 140M | Comparing to de-duplicated training dataset |
| Figure 17b | $2^{19}$ | 140M | 20 epochs |
| Figure 18 | - | - | Identical to Figure 2, where we use 16 different target models |
| Figure 19 | - | - | Identical to Figure 4b, where we use 16 different target models |
| Figure 20 | | | Identical to Figure 18, except varying references (up to 128). |
| Figure 21 | $2^{23}$ | 140M | Up to 64 targets, 64 references |
| Figures 22, 24a, 25, 26, 28, 30a | 7M | 140M | 125 target models (varying only random seed; same training data); 128 references |
| Figure 31 | - | - | Identical to Figure 6b |
| Figure 32 | $2^{23}$ | - | 10M-302M model sizes |

