# OpenReview forum: "Exploring the limits of strong membership inference attacks on large language models"
_NeurIPS.cc/2025/Conference — NeurIPS 2025 poster_

### Official Review · Reviewer_mdKc · 2025-06-26

**Clarity:** 3
**Significance:** 2
**Originality:** 3
**Rating:** 4
**Confidence:** 3

**Summary:**

This paper studies the effectiveness of strong Membership Inference Attacks (MIAs) on pre-trained Large Language Models (LLMs), such as GPT-2-like models. While previous research has primarily focused on smaller models or weaker attacks, this work scales one of the strongest MIAs, the Likelihood Ratio Attack (LiRA), to large-scale LLMs trained on massive datasets.

**Questions:**

What potential improvements could be made in the MIA attack methodologies to push AUC values beyond 0.7, particularly for large models?

**Ethical Concerns:**

["NO or VERY MINOR ethics concerns only"]

**Final Justification:**

Updated rating after the rebuttal, as most concerns are resolved.

**Limitations:**

The experiments are based on GPT-2-like models and may not fully generalize to other types of LLMs (e.g., BERT-based or newer architectures like GPT-4). Further work would be needed to determine how these attacks scale across diverse architectures.

The technical contribution is limited, although this work provides some insights based on experiments.

Updated rating after the rebuttal.

**Quality:**

3

**Strengths And Weaknesses:**

Strengths:

1. The authors conduct experiments on a larger scale than most prior research by training over 4,000 models of varying sizes.

2. The paper covers multiple dimensions of MIAs, such as the number of reference models, model size, training data size, and the impact of different hyperparameters on MIA success.

Weaknesses:

1. The focus on only two MIA methods (LiRA and RMIA) may not provide a full picture of the various possible membership inference strategies. Including more attack methods might have been useful to give a broader view of potential vulnerabilities.

2. While the paper demonstrates that strong MIAs can succeed in a controlled environment, the practical impact remains limited due to the low AUC values (<0.7).

---

> ### Author Rebuttal · Authors · 2025-07-30
>
> Thanks for the review.
>
> > What potential improvements could be made in the MIA attack methodologies to push AUC values beyond 0.7, particularly for large models?
>
> Good question,  It is often assumed that adding more reference models should capture a sample’s member and non-member distributions accurately. However, we found that training more reference models doesn’t improve attack performance beyond a certain point, indicating that training more reference models will not help separate these distributions. Future work could focus on trying to find other signals (beyond the loss of an example) that can help separate these distributions, or focus on establishing if we are hitting an upper limit of attack capabilities. However, it is entirely possible that much higher AUC values are impossible to achieve. We believe that showcasing existing limitations of state-of-the-art strong MIAs (as our results do) is useful for advancing the field.
>
> > The focus on only two MIA methods (LiRA and RMIA) may not provide a full picture of the various possible membership inference strategies. Including more attack methods might have been useful to give a broader view of potential vulnerabilities.
>
> LiRA is widely regarded as one of, if not the strongest known MIA. (Please refer to Section 2 for references that support this claim.) We also compared against another very strong attack, RMIA, and found LiRA outperforms RMIA in almost all settings. We are not claiming any theoretical bounds in this work. Nevertheless, it remains important to understand the strengths and limitations of state-of-the-art strong attacks in this setting, in order to have a sense of how vulnerable pre-trained LLMs are to such attacks. To the best of our knowledge, we are the first to investigate this in detail–to instantiate these strong attacks and study them in depth in this setting. Altogether, we believe that our results shed light on what is empirically possible to achieve with an MIA in this common (and growing) setting. We believe that it is important knowledge for the field to understand the limits of these attacks (notably, as mentioned below in the reviewer’s next question, that AUC is perhaps lower than one might have hoped or expected).
>
> > While the paper demonstrates that strong MIAs can succeed in a controlled environment, the practical impact remains limited due to the low AUC values (<0.7).
>
> Yes, as noted above,  the significance of our work to the literature is primarily in investigating the strengths and limits of state of the art MIAs on LLMs. We don’t believe low AUC values detract from our findings, but rather reflect important knowledge that can help advance the field.

---

> > ### Comment · Reviewer_mdKc · 2025-08-04
> >
> > Thanks for the comments. Most of the concerns are resolved.

---

> > > ### Author Response · Authors · 2025-08-05
> > > **Response**
> > >
> > > Thank you for acknowledging our rebuttal!
> > >
> > > We’re noticing an odd (hopefully bug) on our end with OpenReview: with what we can see, it looks like this review no longer has a rating (somehow, we can just the confidence score). We can see that you’ve noted that you’ve adjusted your score based on the discussion period. If you get a chance, can we trouble you to make sure the rating is selected on your end or conveyed to the ACs?
> > >
> > > Thank you so much.

---

> > > > ### Comment · Reviewer_mdKc · 2025-08-05
> > > >
> > > > Seems the rating after the update is hidden from authors.

---

### Official Review · Reviewer_J6ua · 2025-07-01

**Clarity:** 3
**Significance:** 4
**Originality:** 3
**Rating:** 4
**Confidence:** 3

**Summary:**

This work aims to investigate whether the limitations observed in prior studies stem from the design choices of the attacks themselves or whether MIAs are fundamentally ineffective on large language models. To this end, the authors propose scaling LiRA to GPT-2 architectures ranging from 10 million to 1 billion parameters, training reference models on over 20 billion tokens from the C4 dataset.

**Questions:**

Please refer to Weaknesses.

**Ethical Concerns:**

["NO or VERY MINOR ethics concerns only"]

**Final Justification:**

I have read the rebuttal and it has addressed most of my concerns. I will keep my score.

**Limitations:**

Please refer to Weaknesses.

**Quality:**

3

**Strengths And Weaknesses:**

**Strengths**

1. The motivation of the manuscript is novel.

2. The manuscript offers a comprehensive experimental analysis.

**Weaknesses**

1. In principle, the main text should be self-contained; however, too many details placed in the Appendix.

2. The reason for setting 4,000 GPT-2-based reference models is not clearly explained.

3. The transitional claim regarding LLMs is somewhat misleading, as the models used in the manuscript are relatively small and do not truly qualify as large language models.

4. Can the manuscript provide feasible recommendations for MIAs that are limited to LLMs?

---

> ### Author Rebuttal · Authors · 2025-07-30
>
> Thanks for the helpful review!
>
> > In principle, the main text should be self-contained; however, too many details placed in the Appendix.
>
> We will try to fix this to make it more self-contained. Are there details in particular whose inclusion you believe would improve the main text? We ran a very large number of experiments, so unfortunately some details will need to remain in the Appendix. If there are sections where this stood out to you as resulting in unclear presentation, we would greatly appreciate knowing.
>
> > The reason for setting 4,000 GPT-2-based reference models is not clearly explained.
>
> This was the total number of models we trained accumulated over all of our experiments. The reason for this total number is because, for each experiment, the results (AUC) saturated for a sufficient number of reference models.  At this saturation point, there was no benefit in scaling up the number of models we trained, as AUC remained the same. We will make this clearer in the camera ready version.
>
> > The transitional claim regarding LLMs is somewhat misleading, as the models used in the manuscript are relatively small and do not truly qualify as large language models.
>
> We intended for  the qualifier in the title to help with this, but we can remove “Large” if desired. It is however worth noting that how “large” a model needs to be in order to be an LLM is not exactly standardized. Much of the literature still does consider models of this size to be LLMs, but, for example, models of this size of course are not “frontier LLMs.”
>
> > Can the manuscript provide feasible recommendations for MIAs that are limited to LLMs?
>
> We are not sure if the reviewer is asking for a recommendation for how to improve MIAs or defend against them, so we will provide answers to both possible interpretations.
>
> For improvement: Figure 1 shows the main difficulty in scaling to strong MIA attacks. A large number of reference  models is  required to reach peak attack performance (at this point, performance saturates, and more reference models do not help improve performance). Future designs should concentrate on reducing the number of reference models. We discuss this in Appendix A, where we compare LiRA with RMIA. In certain settings with fewer reference models, RMIA outperforms LiRA.. Fundamentally though, our results also indicate  that the performance of MIA relies on how well an attacker can capture per-example variance in vulnerability to MIA, when the sample is a member / non-member of training. The simplest way to measure per-example variance is through training a sufficiently large number of reference models. Future work should explore other, cheaper ways to capture this signal.
>
> For defending against MIA: Our insights in figure 5 suggest that certain properties of the training process and the data are responsible for excessive vulnerability. For instance, an example tends to be more vulnerable when it is included near the end of training; “atypical” examples are also more vulnerable.  Data curation and model training should take this into consideration when designing their training pipeline.

---

> > ### Comment · Reviewer_J6ua · 2025-08-01
> >
> > I have read the rebuttal and it has addressed most of my concerns. I will keep my score.

---

### Official Review · Reviewer_Lzho · 2025-07-02

**Clarity:** 3
**Significance:** 3
**Originality:** 2
**Rating:** 4
**Confidence:** 3

**Summary:**

The paper studies multiple phenomena for membership attacks that were little known before when it comes to larger models—with enormous numbers of parameters in the era of LLMs and huge training corpus. The paper states three claims: 1- The success rate does not exhibit any meaningful relationship with the model size. I.e., it is not true that larger models are easier to break. 2- The success rate is not related to the training size either. 3- Samples later seen at training are more susceptible to the membership attacks, yet the connection with data extraction is not clear.

**Questions:**

Address weaknesses above

**Ethical Concerns:**

["NO or VERY MINOR ethics concerns only"]

**Limitations:**

Yes

**Quality:**

3

**Strengths And Weaknesses:**

Strengths:
The paper is well written, with the messages clear in each section and sections well-partitioned which follow a natural flow. The paper has several important messages (mentioned above) that were not fully investigated before, and have a clear impact on the literature. This will motivate the community for further studies.

Weaknesses:
1- The novelty of the paper is limited as it is grounded on the previously known LiRA attack. The major contribution of the paper is scaling all the models, the number of reference models, and the training data.
2- One of the surprising results is that AUC never surpasses 0.7 when increasing the number of reference models. Even though the paper claims that they study the fundamental limits of membership attacks (for instance, lines 9 and 10), the low success that is reported only based on LiRA questions the generality and universality of their results.
3- Having said that, there is more room for explaining many of the results. E.g., why does the saturation for the number of reference models happen so quickly?
4- In Figure 4a and 4b, where the relation with more training epochs is studied, the authors resort to discussing AUC only. Whereas the plots keep toggling for different values of FPR and it looks like the mentioned message does not apply everywhere. In other words, the statement that membership attacks are more successful when increasing the number of training epochs seems to oversimplify what’s actually happening in the experiments.

---

> ### Author Rebuttal · Authors · 2025-07-30
>
> Thanks for the helpful review, answering your questions and responding to comments below:
>
> > The novelty of the paper is limited as it is grounded on the previously known LiRA attack. The major contribution of the paper is scaling all the models, the number of reference models, and the training data.
>
> Yes - the significance of our work to the literature is primarily in investigating the performance and limits of state of the art MIAs on LLMs across a wide range of settings. To the best of our knowledge, this has not been studied before, which (as we have discussed with respect to preliminaries and related work) has left important open questions in the field about the feasibility and reliability of MIAs in modern pre-trained LLM settings. So, we would like to emphasize that novelty can also be attributed to insightful findings; as noted in your review, our work has discovered several such findings that have helped resolve some of these important open questions and therefore will help advance the field.
>
> > One of the surprising results is that AUC never surpasses 0.7 when increasing the number of reference models. Even though the paper claims that they study the fundamental limits of membership attacks (for instance, lines 9 and 10), the low success that is reported only based on LiRA questions the generality and universality of their results.
>
> LiRA is widely regarded as one of, if not the strongest known MIA. (Please refer to Section 2 for references that support this claim.) We also compared against another very strong attack, RMIA, and found LiRA outperforms RMIA in almost all settings. We are not claiming any theoretical bounds in this work. Nevertheless, it remains important to understand the strengths and limitations of state-of-the-art strong attacks in this setting, in order to get a sense of how vulnerable pre-trained LLMs are to such attacks. To the best of our knowledge, we are the first to investigate this in detail by instantiating these strong attacks and studying them in depth in this setting. Altogether, we believe that our results shed light on what is empirically possible to achieve with an MIA attack in this common (and growing) setting. We contend that it is important knowledge for the field to understand the limits of these attacks (notably, as you’ve mentioned, that AUC is perhaps lower than one might have expected or hoped).
>
> > Having said that, there is more room for explaining many of the results. E.g., why does the saturation for the number of reference models happen so quickly?
>
> The variance of an individual example’s loss over member and non-member distributions is captured sufficiently well with a small number of reference models. As a result, at a certain point, there are diminishing returns to training more reference models. We are happy to update the manuscript to reflect this (e.g., by including a similar plot to Figure 19 spanning different numbers of reference models).
>
> > In Figure 4a and 4b, where the relation with more training epochs is studied, the authors resort to discussing AUC only. Whereas the plots keep toggling for different values of FPR and it looks like the mentioned message does not apply everywhere. In other words, the statement that membership attacks are more successful when increasing the number of training epochs seems to oversimplify what’s actually happening in the experiments.
>
> We assume the reviewer is referring to Figures 3a and 3b. Figure 4 measures MIA under different training and model sizes, and doesn’t vary the number of epochs. We choose to report AUC as our primary metric, as it is more challenging to visualize the TPR over a wide range of FPR in a streamlined way: (For comparison, see Figure 2b, which provides such an alternate visualization for only a limited range of FPR, at the cost of not surfacing overall AUC.). Every result in our work comes with an AUC-ROC curve (in log-log scale), allowing the reader to decide which FPR threshold is of interest, rather than a prescriptive choice from the authors. We are happy to add more discussion on FPR for this set of experiments if the reviewer thinks it will improve the paper, and can also mention this visualization choice at the beginning of Section 3 discussing our overall experimental setup.

---

> > ### Author Response · Authors · 2025-08-08
> >
> > We are very grateful for your review feedback. We kindly ask whether you might consider improving the score to reflect the strengthened manuscript. Please let us know if any further clarification is required.
> >
> > Sincerely yours,
> >
> > Authors

---

### Official Review · Reviewer_2cHX · 2025-07-03

**Clarity:** 4
**Significance:** 4
**Originality:** 2
**Rating:** 5
**Confidence:** 4

**Summary:**

This paper scales strong MIAs to large language models, addressing limitations of prior work that used weaker or small-scale attacks. By applying the LiRA attack to thousands of GPT-2–like models, the authors show that strong MIAs can succeed, but their effectiveness is limited in practical settings. They also find that vulnerability does not consistently increase with model size and that MIAs and data extraction capture different privacy risks.

**Questions:**

1. Why does the paper mostly focus on the AUC evaluation metric and not on TPR at low FPR? Can you provide more insight from this perspective?

2. Can we provide more insights into MIA attack designs or suggest directions for improving MIA attack design for LLMs?

3. This paper involves large-scale experiments that require significant computational resources. Can we also provide more insights from a cost perspective? Also, the possibilities of further scaling this to larger LLMs?

**Ethical Concerns:**

["NO or VERY MINOR ethics concerns only"]

**Final Justification:**

I will keep my original score 5. I think this is a good paper and should be accepted.

**Limitations:**

Yes.

**Quality:**

3

**Strengths And Weaknesses:**

Strengths:

1. Insightful results (e.g., data used in later training is more vulnerable; no clear relationship between MIA and extraction attacks) — these findings are meaningful and could influence future studies.

2. Large-scale experiments.

3. Demonstrates that strong MIAs on LLMs are feasible.

Weaknesses:

1. For the most part, this paper uses AUC as the primary evaluation metric. However, the LiRA paper highlights that TPR at low FPR may be a more appropriate measure for evaluating MIAs. It would be helpful if the authors could also present findings using these alternative metrics, which may yield additional insights.

2. The attack methods are not very novel, but applying existing SOTA MIAs in the LLM setting.

Thank you for your submission. I enjoyed reading this paper and learned new things from it—especially the insightful findings about which data is more vulnerable to MIAs and the relationship between MIA and extraction attacks (where many prior works suggested a connection, but this paper presents a different perspective). That said, I do have some concerns, as mentioned above.

---

> ### Author Rebuttal · Authors · 2025-07-30
>
> Thanks for your kind review. Answering your questions below:
>
> >Why does the paper mostly focus on the AUC evaluation metric and not on TPR at low FPR? Can you provide more insight from this perspective?
>
> This is partially explained in lines 153-159: We choose to report AUC as our primary metric, as it is more challenging to visualize the TPR over a wide range of FPR in a streamlined way: (For comparison, see Figure 2b, which provides such an alternate visualization for only a limited range of FPR, at the cost of not surfacing overall AUC.). Every result in our work comes with an AUC-ROC curve (in log-log scale), allowing the reader to decide which FPR threshold is of interest, rather than a prescriptive choice from the authors. For clarity, we can also sign post this/ reference this rationale at the beginning of section 3, where we describe our overall experimental setup.
>
> > Can we provide more insights into MIA attack designs or suggest directions for improving MIA attack design for LLMs?
>
> Figure 1 shows the main difficulty in scaling strong MIA attacks. A large number of reference models (each of which is also more expensive to train at scale) are required to attain peak attack performance. Future attack designs should concentrate on reducing the number of reference models. We discuss this in Appendix A, where we compare LiRA with RMIA. In certain (smaller scale) settings with fewer reference models, RMIA outperforms LiRA.. However, our results also show that peak attack performance (of roughly 0.7 AUC) is only part of the story. Fundamentally, our results also indicate that the performance of MIA relies on how well an attacker can capture per-example variance in MIA vulnerability when the sample is a member / non-member of training. The simplest way to measure per-example variance is through training a sufficiently large number of  reference models. Future work should explore other, cheaper ways to capture this signal. We can include these details in our conclusion/ takeaways section for future work.
>
> > This paper involves large-scale experiments that require significant computational resources. Can we also provide more insights from a cost perspective? Also, the possibilities of further scaling this to larger LLMs?
>
> We can certainly provide information on computation costs. Our maximum run used approximately 2e19 training flops using 4 TPUv3s (https://cloud.google.com/tpu/docs/v3), we will add a new compute cost column to Table 1 and discussion. Regarding scaling to large LLMs: since we wanted to measure MIA under “realistic” pre-training settings, where the number of training tokens is scaled proportional to the model size, the main difficulty is simply cost. We can reduce cost by reducing the number or fixing the number of training tokens. We plan to investigate the relationship between training tokens and model size in more depth in future work.

---

### Note · Authors · 2025-08-12

We extend our sincere gratitude to all reviewers for their thoughtful engagement and constructive feedback. We are encouraged that the reviewers recognized the significance of our core contribution: a large-scale empirical study that establishes benchmarks for strong Membership Inference Attacks on moderately large language models. The review process has been incredibly valuable.

---

### Decision · Program_Chairs · 2025-09-17

**Decision:**

Accept (poster)

**Comment:**

This paper provides a valuable large-scale empirical study of strong Membership Inference Attacks on moderately large language models. The reviewers praised its significant findings, such as the limited practical success of these attacks and the lack of a clear link between vulnerability and model size. The primary weaknesses noted were the limited methodological novelty, as it scales an existing attack, and the use of GPT-2 models, which may not represent today's larger architectures. Overall, the consensus is that the paper's strengths and insightful results from its comprehensive experiments outweigh these concerns, making it a solid contribution that merits acceptance.